# Optimizing differential expression analysis for proteomics data via high-performing rules and ensemble inference

Hui Peng [1,2], He Wang [1,2], Weijia Kong [1,2], Jinyan Li [3] ✉ &
Wilson Wen Bin Goh [1,2,4,5,6] ✉

Identification of differentially expressed proteins in a proteomics workflow typically encompasses five key steps: raw data quantification, expression matrix construction, matrix normalization, missing value imputation (MVI), and differential expression analysis. The plethora of options in each step makes it challenging to identify optimal workflows that maximize the identification of differentially expressed proteins. To identify optimal workflows and their common properties, we conduct an extensive study involving 34,576 combinatoric experiments on 24 gold standard spike-in datasets. Applying frequent pattern mining techniques to top-ranked workflows, we uncover high-performing rules that demonstrate optimality has conserved properties. Via machine learning, we confirm optimal workflows are indeed predictable, with average cross-validation F1 scores and Matthew's correlation coefficients surpassing 0.84. We introduce an ensemble inference to integrate results from individual top-performing workflows for expanding differential proteome coverage and resolve inconsistencies. Ensemble inference provides gains in pAUC (up to 4.61%) and G-mean (up to 11.14%) and facilitates effective aggregation of information across varied quantification approaches such as topN, directLFQ, MaxLFQ intensities, and spectral counts. However, further development and evaluation are needed to establish acceptable frameworks for conducting ensemble inference on multiple proteomics workflows.

Differential expression analysis (DEA) for proteomics data is crucial for accurate detection of phenotype-specific proteins, which can be useful in biomedical applications such as biomarker and drug target discovery[1,2]. DEA workflows usually comprise five key steps: raw data quantification, expression matrix construction, matrix normalization, missing value imputation (MVI), and finally, conducting differential expression analysis by means of a statistical method. In each step, multiple options in terms of methods/tools are available (Fig. 1).

Selecting different options in each step can result in varied outcomes in terms of differential protein reporting. Given the numerous combinations possible, identifying an optimal workflow suitable for one's data is challenging.

There have been prior efforts for identifying optimal workflows or, at least, identification of some high-performing options specific to certain steps in the DEA workflows. Each of these efforts have some limitations: Langley et al.[3] evaluated 7 DEA tools based on spectral

[1]Lee Kong Chian School of Medicine, Nanyang Technological University, Singapore, Singapore. [2]School of Biological Sciences, Nanyang Technological University, Singapore, Singapore. [3]Shenzhen Institute of Advanced Technology, Chinese Academy of Sciences, Shenzhen, China. [4]Center for Biomedical Informatics, Nanyang Technological University, Singapore, Singapore. [5]Center of AI in Medicine, Nanyang Technological University, Singapore, Singapore. [6]Division of Neurology, Department of Brain Sciences, Faculty of Medicine, Imperial College London, London, UK. ✉e-mail: Jinyan.li@siat.ac.cn; wilsongoh@ntu.edu.sg

**Fig. 1 | Workflow for performing differential expression analysis (DEA) on proteomics data. a** The five main steps in a typical DEA workflow. **b** Quantification covers peptide identification, protein assembling and quantitative analysis on an analysis platform, e.g., FragPipe[25] for DDA or DIA-NN[26] for DIA. **c** Selection of quantification results to be expressed as a matrix containing spectral counts or protein intensities. Available matrix types for each quantification platform are located in a colored dashed box. **d** Normalization of raw expression matrix to reduce systematic bias, e.g., by center.mean or by vsn[64]. **e** Imputation of missing values, which includes methods such as missForest[35]. **f** Selection of suitable DEA statistical tool, e.g., ROTS[6] to identify differentially expressed proteins. In **d**–**f**, the compared options in each step are grouped into different categories (shown in boxes with different colors). Detailed descriptions of compared options in each step are summarized in Supplementary Note 2.

count data, but did not consider the influence by other preprocessing steps such as normalization and missing value imputation (MVI) or intensity-based quantifications. Ramus et al.[4] benchmarked 8 DDA (data-dependent acquisition) workflows on their in-house yeast dataset (YUltq819_LFQ as shown in Table 1) integrating different tools for database search, protein assembly and validation, intensity-based quantitation, and with two tools for DEA; however, the impact of the data preprocessing methods was not fully considered. Välikangas et al.[5] integrated several proteomics data processing (identification and quantification) software tools and evaluated 9 MVI algorithms. However, only one DEA tool named ROTS[6] was considered, thus rendering their findings more suited for evaluating MVI methods only. Fröhlich et al.[7] concentrated on evaluating DIA (data-independent acquisition) processing tools, integrating various DIA processing tools, 4 normalization methods, and 7 DEA tools. However, their study did not consider MVI algorithms. Lin et al.[8] compared DEA performance using tools originally designed for gene expression data, and considered MVI and quantification tools in their analysis but overlooked normalization. Sticker et al.[9] compared 7 DEA tools on spike-in datasets but ignored preprocessing methods. Dowell et al.[10] generated spike-in datasets to benchmark tool combinations, including acquisition methods, replicate number, statistical approach, and FDR (False Discovery Rate) corrections[11]. However, the study was limited to only 2 DEA tools and one concentration contrast, while preprocessing was not considered. Suomi et al.[12] compared their DEA tool with three others but did not address preprocessing and other steps.

Although insightful, these prior studies provide an incomplete perspective, as they do not cover the gamut of steps in a workflow nor are they sufficiently representative of the various proteomics platforms available today. Hence, the impact of combinatorial synergies

between steps, e.g., normalization and MVI algorithms, remains poorly understood. It is thus unsurprising that we do not understand what differentiates a high performing workflow from a low performing one. Under such circumstances, researchers are relegated to use published ones from other studies. But this is also not a good solution as the published workflow may not have been benchmarked, and thus, optimality is not assured.

To identify optimal workflows for a wide gamut of proteomics platforms (including label-free and labeled), we implemented a brute-force approach to test all possible combinations of (readily available) options/methods. The performance of each workflow combination is attributed to what we define as a "quantification setting", i.e., pairings of a quantification platform and a data type, e.g., if an experiment involved quantification by Maxquant[13] and is a labeled Tandem Mass Tags (TMT) experiment[14], this setting will be encoded as "MQ_TMT". Unlike the other steps which can be easily changed, the quantification settings reflect experimental conditions which are less flexible (these settings were used for generating the raw data already). Workflows are benchmarked on a large assembly of spike-in datasets inclusive of label-free DDA and DIA modes, or labeled TMT.

Here, we report that optimality is predictable, where workflow performance levels can be classified accurately (average F1 score or MCC score > 0.84) and workflow ranks generalize well to unseen datasets (mean Spearman correlation coefficients > 0.56 between ranks based on unseen datasets against benchmark datasets).

Next, some steps in a workflow are more important in determining outcomes in a setting-specific manner. Normalization and DEA statistical methods exert greater influence than other steps for label-free DDA data and TMT data. Whereas for label-free DIA data, the matrix type is also important. We find that high-performing workflows

**Table 1 | Datasets used for workflow benchmarking**

| Dataset | ID | Technique | Contrasts[a] | Instrument | Mixture | PMID[b] |
|---|---|---|---|---|---|---|
| HYE5600735_LFQ | PXD028735[c] | DDA | 1 | SCIEX Triple TOF5600 | Human + yeast + E. coli | 35354825[31] |
| HYE6600735_LFQ | PXD028735 | DDA | 1 | SCIEX Triple TOF6600 | Human + yeast + E. coli | 35354825[31] |
| HYEqe735_LFQ | PXD028735 | DDA | 1 | Orbitrap QE-HFX | Human + yeast + E. coli | 35354825[31] |
| HYEtims735_LFQ | PXD028735 | DDA | 1 | TimsToF pro | Human + yeast + E. coli | 35354825[31] |
| HYtims134_LFQ | PXD036134 | DDA | 3 | TimsToF pro | Human + yeast | 36541440[83] |
| HEtims425_LFQ | PXD021425 | DDA | 3 | TimsToF pro | Human + E. coli | 34373457[84] |
| YUltq006_LFQ | PDC000006[d] | DDA | 2 | LTQ-Orbitrap | Yeast + UPS1 | 19858499[24] |
| YUltq099_LFQ | PXD002099 | DDA | 2 | LTQ Orbitrap Velos | Yeast + UPS1 | 26321463[85] |
| YUltq819_LFQ | PXD001819 | DDA | 3 | LTQ Orbitrap Velos | Yeast + UPS1 | 26862574[86] |
| HEqe408_LFQ | PXD018408 | DDA | 1 | Q Exactive | Human + E. coli | 33553868[10] |
| HYqfl683_LFQ | PXD007683 | DDA | 3 | Orbitrap Fusion Lumos | Human + yeast | 29635916[87] |
| HYEtims777_LFQ | PXD014777 | DDA | 1 | TimsToF pro | Human + yeast + E. coli | 32156793[13] |
| HYEtims735_DIA | PXD028735 | DIA | 1 | TimsToFpro | Human + yeast + E. coli | 35354825[31] |
| MYtims709_DIA | PXD034709 | DIA | 3 | TimsTOF Pro | Mouse + yeast | 36609502[88] |
| HEof_n600_DIA | PXD026600 | DIA | 3 | Orbitrap Fusion ETD | UPS1 + E. coli | 34472865[89] |
| HEof_w600_DIA | PXD026600 | DIA | 3 | Orbitrap Fusion ETD | UPS1 + E. coli | 34472865[89] |
| HYtims134_DIA | PXD036134 | DIA | 3 | TimsToF pro | human + yeast | 36541440[83] |
| HEqe777_DIA | PXD019777 | DIA | 3 | Q Exactive HF | Human + E. coli | 34373457[84] |
| HEqe408_DIA | PXD018408 | DIA | 1 | Q Exactive | Human + E. coli | 33553868[10] |
| HEqe277_TMT10 | PXD013277 | TMT | 3 | Q Exactive | Human + E. coli | 32205417[34] |
| HYqfl683_TMT11 | PXD007683 | TMT | 3 | Orbitrap Fusion Lumos | Human + yeast | 29635916[87] |
| HYms2faims815_TMT16 | PXD020815 | TMT | 3 | Orbitrap Fusion Lumos | Human+ yeast | 33175540[90] |
| HYsps2815_TMT16 | PXD020815 | TMT | 3 | Orbitrap Fusion Lumos | Human + yeast | 33175540[90] |
| HYms2815_TMT16 | PXD020815 | TMT | 3 | Orbitrap Fusion Lumos | Human + yeast | 33175540[90] |

[a]Number of contrasts used for benchmarking.
[b]The PubMed unique identifier of the publication reporting the dataset (https://pubmed.ncbi.nlm.nih.gov/).
[c]ProteomeXchange ID (http://proteomecentral.proteomexchange.org/cgi/GetDataset).
[d]Proteomic Data Commons Study Identifier (https://proteomic.datacommons.cancer.gov/pdc/).

of label-free data are enriched for the directLFQ intensity[15], no normalization (referring only to distribution correction methods that are not embedded with any particular settings), and incline SeqKNN[16,17], Impseq[17,18], or MinProb[19] (probabilistic minimum) for imputation while eschewing simple statistical tools (e.g., ANOVA[20], SAM[21], and t-test[22] are enriched in low-performing workflows).

Finally, we report that workflow integration is beneficial for expanding differential proteome coverage but is also a double-edged sword. Given the increased attention to machine learning approaches, we design an ensemble inference approach that integrates DEA results from individual top-performing workflows. The ensemble approach can increase true positives, leading to improvements of mean pAUC(0.01) by 1.17–4.61% and improvements of the mean G-mean scores by as high as 11.14% across six different quantification settings. In particular, the integration of top 1st workflows using top0 intensities (which incorporates all precursors) and intensities extracted with directLFQ and MaxLFQ[23] improves the DEA performance more than any of the best single workflows did, gaining a pAUC(0.01) of 4.61% under the MQ_DDA setting (using Maxquant to quantify label-free DDA data). This suggests that while top0 may not work as well as directLFQ intensity in DEA workflows, combining these multiple workflows provides complementary information that enhances DEA outcomes. However, the increase in true positives also comes with the risk of false positives. To mitigate such risks, further development on workflow integration approaches is needed.

For users to study the impact of choices at each step of a DEA workflow and facilitate the practical usage of our findings, we provide a unique resource, OpDEA, to guide workflow selection on new datasets. This tool is available at http://www.ai4pro.tech:3838/.

## Results

### Assembled benchmark proteomic datasets

We amassed 12 label-free DDA datasets, 5 TMT datasets, and 7 label-free DIA datasets from different proteomics projects for assessing the performance of DEA workflows (Table 1). This assemblage is, to the best of our knowledge, the most comprehensive and largest collection of benchmark data for workflow optimization across key proteomic platforms and has value for testing novel algorithms. We package these datasets under the OpDEA resource at http://www.ai4pro.tech:3838.

Table 1 presents the dataset names (Dataset), dataset depository identifier (ID), technique used for data generation (Technique), number of contrasts used for benchmarking (Contrasts), instrument type of the spectrometer (Instrument), the mixture type (Mixture) and the PubMed unique identifier of the publication reporting the dataset (PMID) for each dataset. A contrast refers to an expression-level comparison of two groups of proteins acquired from two different samples. Thus, the contrast number in a dataset is equivalent to the number of unique sample pairs contained therein.

For a specific spike-in dataset such as YUltq006_LFQ[24], a total of 48 UPS1 (Universal Proteomics Standard 1) proteins spanning five concentrations were compared against the yeast background proteins. In this scenario, by selecting samples given 2 out of the 5 distinct concentrations, 10 possible pairwise contrasts are generated (i.e., $C_5^2 = 10$). Each of these selected contrasts is useful for the identification of differentially expressed proteins during analysis.

To prevent the performance benchmark from being dominated by any single dataset, we evaluated all contrasts in those datasets with many contrasts, and then selected only a subset of these contrasts for comparison. E.g., only 2 out of the 10 contrasts in YUltq006_LFQ were

used. Altogether, we have 22, 15 and 17 contrasts representing DDA, TMT and DIA platforms.

Due to compatibility considerations, we used the following quantification software for specific proteomics platforms:

- Fragpipe[25] and Maxquant[13] for the DDA and TMT data.
- DIA-NN[26] and Spectronaut[27] (spt) for the DIA data.

For further details regarding workflows pertinent to each platform, please refer to the "Methods" section.

### Optimal workflows are predictable and are settings-specific

We evaluate workflows on five performance metrics (see the "Methods" section for details):

- Partial area under receiver operator characteristic curves (pAUC)[28] with false-positive rate (FPR) thresholds of 0.01, 0.05, or 0.1 (denoted as pAUC(0.01), pAUC(0.05), pAUC(0.1)).
- Normalized Matthew's correlation coefficient (nMCC)[29].
- Geometric mean of specificity and recall (G-mean)[30].

For each workflow, we calculate each performance metric and determine its corresponding mean value across benchmark datasets. Each mean performance metric (regarding a particular workflow) is then converted to a rank based on performance relative to other workflows. A workflow's final rank is determined by averaging the ranks over the 5 performance metrics (see the "Methods" section). The detailed rankings of the workflows are listed in Supplementary Data 2.

Figure 2a shows the distributions of performance metrics on FragPipe-based workflows across DDA datasets (the specific combination of the data type and a quantification platform are defined as a setting, e.g., FG_DDA refers to label-free DDA data quantified by FragPipe). We observed wide performance gaps between top- and bottom-ranked workflows (see our performance comparisons between top-5% workflows and bottom-50% in Supplementary Fig. 3 of Supplementary Information). The wide variability of performance metrics suggests that selecting an appropriate workflow is important.

To test whether our benchmarking results can support workflow recommendations for new datasets, we designed a leave-one-dataset-out cross-validation (LODOCV) procedure to take advantage of our assemblage of datasets in Table 1. LODOCV is a form of multiple validations, wherein each test round, one dataset is reserved for performance testing while the remaining datasets are used for model training (see the "Methods" section). Although LODOCV takes the form of a typical cross-validation procedure, there is an important distinction: In cross-validation, a dataset of a single origin is split into multiple components for the purpose of model tuning. In LODOCV, the datasets are of multiple origins. In this regard, the LODOCV procedure is akin to performing several rounds of independent validation and is more robust than typical procedures where only one independent validation is performed for a trained model. To evaluate consistency, we use the Spearman correlation coefficient ($R$) to compare the workflow ranks obtained from the training dataset against the validation dataset.

Figure 2b illustrates the results from one round of LODOCV. We plot the ranks of workflows obtained from the validation dataset HYEtims735_LFQ[31] (under the setting FG_DDA, Table 1) against the corresponding ranks from the remaining DDA datasets (training datasets). In this comparison, a mean $R$ of 0.72 was achieved, suggesting the conservation of information. To summarize across all our datasets, we present the overall LODOCV results tested under different settings in Fig. 2c (and Supplementary Data 1).

The workflow ranks on TMT are quite stable where we obtained ~0.8 mean $R$ regardless of whether FragPipe or Maxquant (FG_TMT or MQ_TMT) was used for quantification. A mean $R \sim 0.65$ was achieved on DEA workflows for FG_DDA and MQ_DDA (label-free DDA data

quantified by Maxquant). For workflows based on DIANN_DIA and spt_DIA (label-free DIA data quantified by DIA-NN and Spectronaut respectively), a mean $R$ of ~0.57 was obtained.

The mean ranks may be unstable if performance metrics are sensitive to the choice of instrument. To assay this, we used the Kruskal–Wallis test[32] (see the "Methods" section). We plot 30 top-performing workflows ranked by mean performance under different settings against the log-transformed $p$ values of the Kruskal–Wallis tests (Fig. 2d). We conclude that under most circumstances, the rankings of top workflows are not sensitive to instrument type (shown as gray markers in Fig. 2d). However, some workflows involving DIA-NN_DIA do show some sensitivity to instrument type, with 4 out of 30 workflows reporting $p$-values < 0.05, indicating significant ranking position differences between instrument types (Supplementary Data 2).

Based on the LODOCV and determining that the workflow ranks are not associated with instrument type, we conclude our benchmarking ranks are stable and are positively correlated to actual workflow performances. This generalizable performance suggests that optimal workflows are predictable and can be used to guide workflow recommendations on future (unseen) data.

We observed that top-ranked workflows for settings FG_DDA, MQ_DDA, FG_TMT, MQ_TMT, DIANN_DIA, and spt_DIA can substantially fall in performance rankings given changes in the four remaining workflow steps. Figure 2e presents the top 2 workflows for each matrix type under settings FG_DDA, MQ_DDA, DIANN_DIA, and spt_DIA by their mean pAUC(0.01) and G-mean scores (The top-ranked workflows under FG_TMT and MQ_TMT are shown in Supplementary Fig. 2 of Supplementary Information, see detailed descriptions of top-ranked workflows in Supplementary Tables 1 and 2 of Supplementary Information). The pAUC(0.01) and the G-mean are used as the representative AUC-based metric and confusion matrix-based metric for the convenient visualization, results based on the three remaining performance metrics can be found in Supplementary Tables 1 and 2 of Supplementary Information. The label and its corresponding color denote the DEA tool, normalization method, MVI algorithm and expression matrix type used by the workflow. The overall workflow rank position is shown in brackets.

### Frequent patterns extracted from high-performing workflows

To discover decision rules enriched in top-ranked workflows, we used machine learning. Under each setting, we encoded every workflow as a feature vector where every option in a step is considered as a categorical feature value. We assigned each workflow a performance level such as high ("H"), relatively high ("RH"), relatively low ("RL"), or low ("L"); if its rank falls within the top 5%, between top 5% and top 25%, between top 25% and 50%, and in the remaining 50%, respectively (see Supplementary Data 3). We used these workflow feature vectors and their labels to train a CatBoost classifier[33] followed by 10-fold cross-validation (we randomly split the workflows into 10 folds, and each fold is evaluated against a trained classifier from the other 9 folds of workflows, see the "Methods" section) for performance evaluation (Fig. 3a).

Workflow performance levels are predictable, with average F1 and MCC scores above 0.84. To understand which features are important for good prediction performance, we examined model feature importance and found that model performance depends more on the choice of normalization and DEA tool than on expression matrix type and MVI algorithms for label-free DDA and TMT data (Fig. 3b; see Supplementary Data 3 for more details). For label-free DIA data, normalization and DEA are again important but the matrix type also appears to be important.

As independent measures, feature importance (of each feature) is not always linked directly to workflow performance. Interactions between features can result in synergies or conflicts depending on

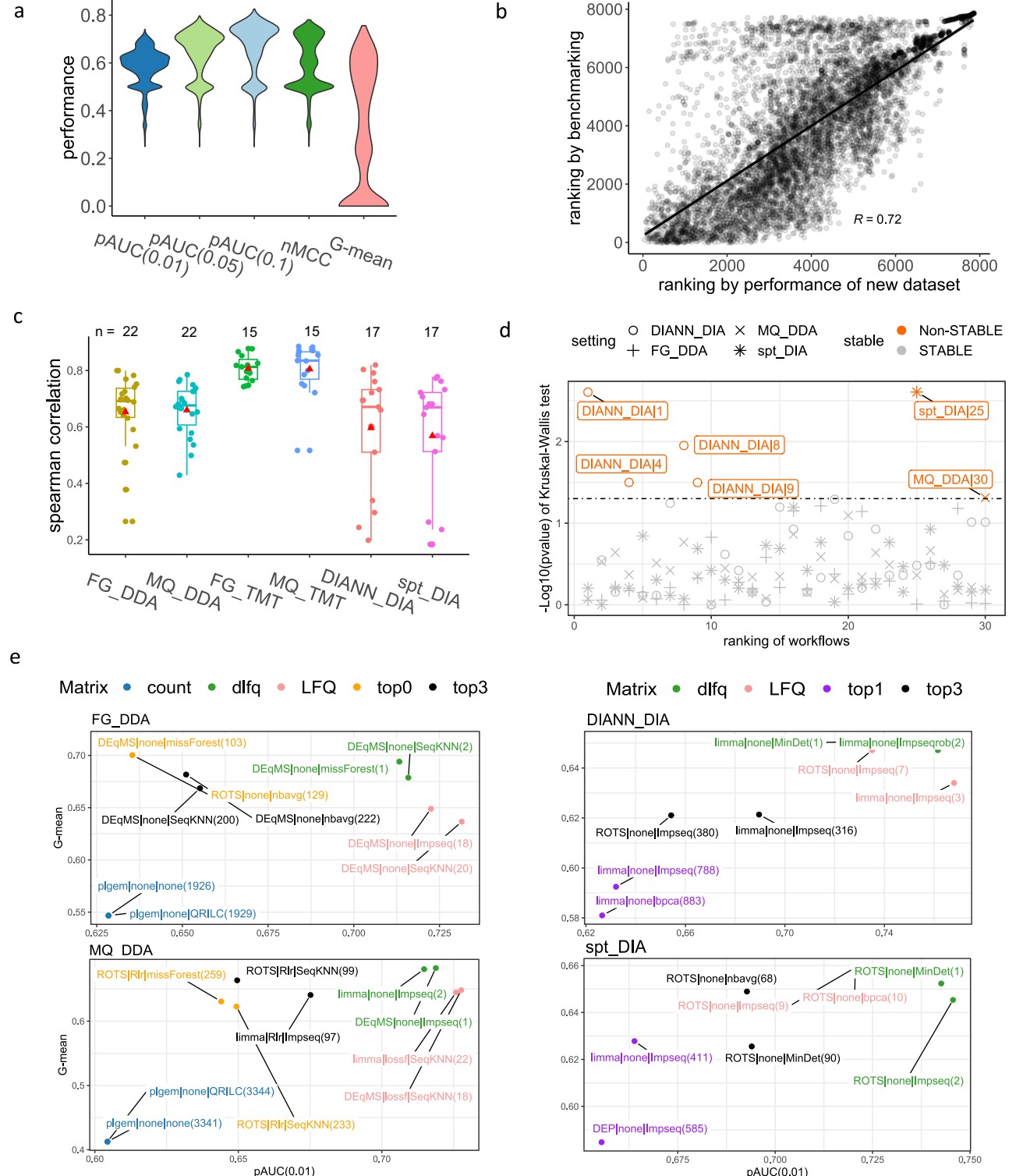

**Fig. 2 | Performance distributions of workflows, leave-one-dataset-out cross-validation (LODOCV) test results, and top-ranked workflows. a** The performance distribution of DDA data quantified by FragPipe-based (FG_DDA) workflows. **b** Presents an example to demonstrate the process of the leave-one-dataset-out cross-validation, where the x-axis shows the averaged ranks of FG_DDA workflows (across five metrics) obtained from dataset HYEtims735_LFQ; the y-axis shows corresponding ranks obtained via benchmarking with mean performance of the remaining datasets. A Spearman correlation of 0.72 with p-value < 2.2e−16 (two-sided t-test) is obtained (N = 7852). **c** The distributions of LODOCV results under different quantification settings. In the boxplots, the mean Spearman correlations are marked by red triangles, centerline indicates the median, box limits indicate

upper and lower quartiles, whiskers indicate the 1.5 interquartile range. The numbers of points for each boxplot (n) are shown above the boxplots. **d** Displays the Kruskal–Wallis (KW) test results checking whether workflow ranks are sensitive to instrument types. The x-axis lists the ranks of the top 30 workflows ranked by mean performances. The y-axis shows the log-transformed p-value of the KW tests. Most comparisons are non-significant, suggesting workflows are not sensitive to instrument types. **e** Shows the top 2 workflows for each matrix type under FG_DDA, MQ_DDA, DIANN_DIA, and spt_DIA settings. The color encodes the matrix type, and the labels encode selection details on DEA, normalization, and MVI. The overall ranks are shown in brackets. Source data of **a**–**e** are provided as a Source Data file.

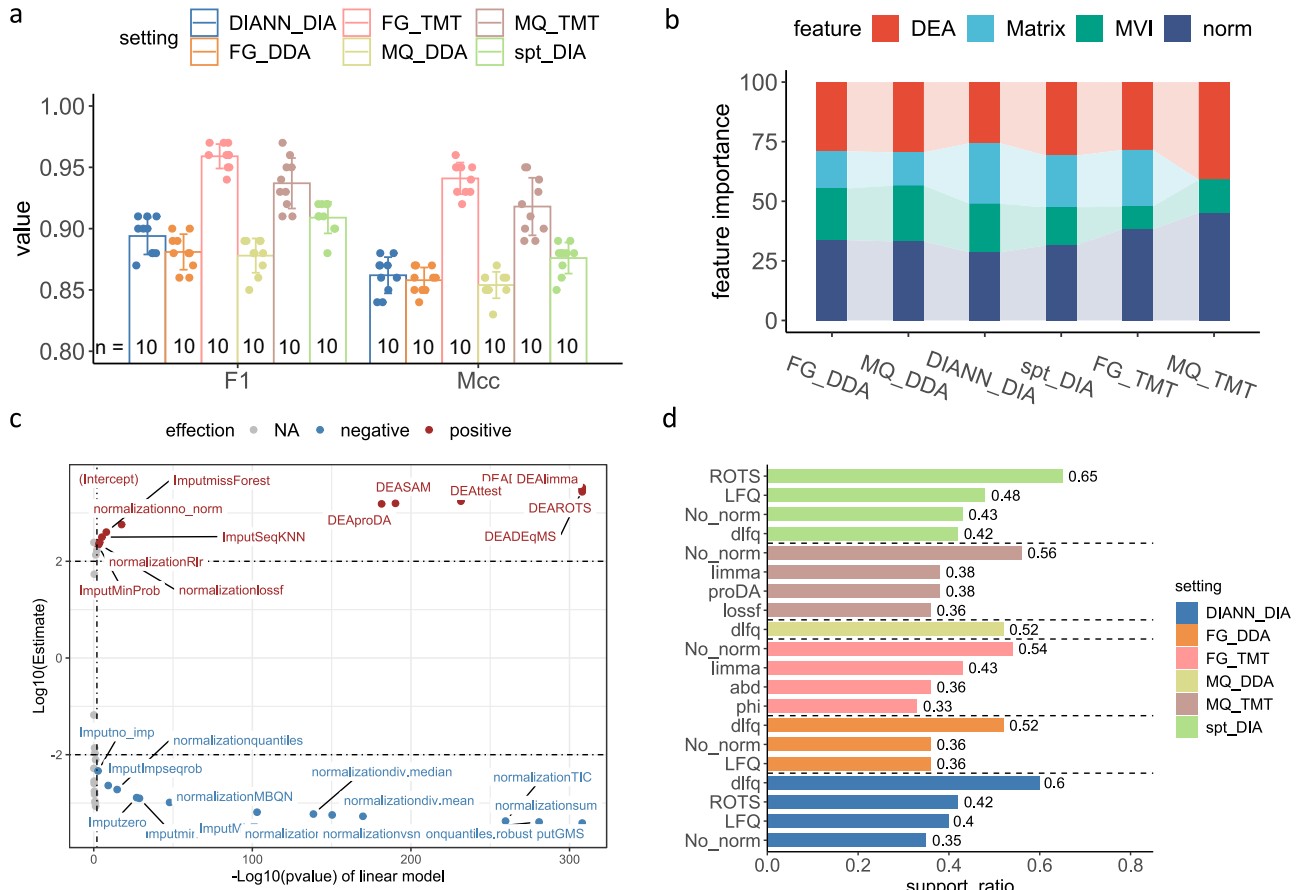

**Fig. 3 | Machine learning and frequent pattern mining provide insights into the traits of high-performing workflows. a** Shows the averaged 10-fold cross-validation F1 and MCC results for workflow classification using the CatBoost classifier. The data shown in the barplots correspond to the geometric mean ± geometric SD of the 10-fold cv. The points present the detailed performance in each round of the 10-fold cv ($n$ = 10, see numbers above the x-axis). **b** Shows the feature importance of CatBoost classifiers in workflow classification. The 0 importance of "Matrix" under MQ_TMT is due to only one Matrix type being available. **c** Shows how varying options in each step of an FG_DDA workflow affect its performance ranking via a linear regression model. The "Estimate" value provides the estimated coefficients for each predictor variable in the linear model. The $p$-value is calculated by a two-sided $t$-test. **d** Lists the discovered frequent patterns associated with high-performing workflows across different settings (with support ratios higher than 0.3). Source data of **a**–**d** are provided as a Source Data file.

their compatibility. We fitted linear models to investigate the interactions between features and workflow ranks. In our linear model, the options in each step of a workflow are set as predictor features, while the performance ranks of the workflows are set as response features (see the "Methods" section). Figure 3c displays the log-transformed $p$-value against the log-transformed estimated increase in ranking scores (calculated by $N$ minus ranking position; $N$ means the total number of workflows) for FG_DDA workflows. The choices of DEA tools such as limma, DEqMS[34], etc., and concomitant selection of MVI algorithms such as SeqKNN, missForest[35], no-normalization or lossf[36,37] normalization, etc., can improve workflow ranks. In contrast, most normalization methods coupled with MVI algorithms such as GMS[38] and no-imputation (no_imp) negatively impact rankings (linear model fitting results for other settings are discussed in Supplementary Note 1 and are listed in Supplementary Data 4).

To deep dive into the extent of the rank shifts, we extracted the ANOVA[20] tables from the linear models of the six settings to check the ranking score differentiation induced by changing the workflow step options (see the "Methods" section). Under setting FG_DDA, we observed that the statistical testing based on the mean differences of workflow groups formed from the options in each workflow step is significant ($p$-value < 0.05). The $F$ values (calculated as the ratio of the between-group variance to the within-group variance, which is used to indicate the impacts of options changes on workflow rankings) are

consistent with the feature importance scores obtained earlier (see Supplementary Table 3). We observed similar conclusions in other settings (Supplementary Fig. 4 and Supplementary Table 3 in Supplementary Information and Supplementary Data 4).

The performance level classification and linear model fitting demonstrated that optimal workflows were predictable and that certain options (and combinations of options) were associated with performance. This gives us confidence to use more targeted approaches to discover high-level rules or frequent patterns associated with high-performing "H" workflows. These can, in turn, be directly used in guiding workflow recommendations without being too explicit or narrow.

We employed the Frequent Patten Growth (FP-growth) algorithm[39] (we also repeated the analyses on low-performing "L" workflows for comparison) to study 393, 393, 236, 80, 315 and 315 "H" workflows for FG_DDA, MQ_DDA, FG_TMT, MQ_TMT, DIANN_DIA and spt_DIA respectively. Frequent patterns with a support ratio (SR, defined as the fraction of the total workflows containing the pattern) >0.30 are depicted in Fig. 3d (frequent patterns with SR ≥ 0.10 but SR ≤ 0.30 are listed in Supplementary Data 5). We identified some common frequent selection patterns for high-performing "H" workflows across various settings (cross-setting patterns). Normalization methods "no_norm" (no additional normalization to data extracted by quantification platforms with default settings, built-in normalization

may have been conducted, e.g., FragPipe enables "Normalize intensity across runs" by default, more details see our Supplementary Note 2) and "lossf"[36,37], and DEA tool "limma" are frequently selected for all the six settings with support ratio bigger than 0.10 (see Supplementary Data 5). Among these, "no_norm" is always the most popular normalization choice for "H" workflows of the various settings (MQ_DDA is the only exception where its corresponding "H" workflows prefer the inclusion of "lossf"). Normalization method center.median (in addition to "no_norm" and "lossf"), DEA tools ROTS and DEP (in addition to "limma"), and matrix types dlfq (directLFQ intensity) and LFQ (MaxLFQ intensity)[23] are also frequently favored by "H" workflows under all 4 label-free data settings. In particular, dlfq is the most enriched expression matrix type amongst "H" workflows under settings FG_DDA, MQ_DDA, and DIANN_DIA, having SRs of 0.52, 0.52, and 0.60, respectively (see Fig. 3d and Supplementary Data 5).

Frequent patterns related to MVI algorithms are more setting specific. "MinDet" has the highest SR (0.13) amongst DIANN_DIA "H" workflows. "SeqKNN", "missForest", and "Impseq" are favored MVI methods by "H" workflows in both settings FG_DDA and MQ_DDA. Limma is the most favored DEA tool amongst "H" workflows in the FG_TMT setting (SR = 0.43), in MQ_TMT (SR = 0.38), and in FG_DDA (SR = 0.25), while ROTS has high SRs in the "H" workflows under setting MQ_DDA (SR = 0.26), DIANN_DIA (SR = 0.42) or under setting spt_DIA (SR = 0.65).

The frequent pattern mining approach can also reveal interesting synergies or avoidances. For example, no-normalization tends to be associated with protein dlfq intensity in "H" workflows under all four label-free settings (SRs ranging from 0.15 to 0.18) and is coupled with LFQ in FG_DDA (SR = 0.11), DIANN_DIA (SR = 0.17) and in spt_DIA (SR = 0.19) "H" workflows. No-normalization is also associated with three available matrix types of FG_TMT (SR = 0.20 for TMT-Integrator abundance (abd), SR = 0.14 for TMT-Integrator ratio (ratio) and SR = 0.19 for Philosopher intensity (phi))[40]. dlfq is associated with the DEA tool ROTS in "H" workflows under all four label-free settings (SRs ranging from 0.11 to 0.29) and is associated with limma in FG_DDA, MQ_DDA, and DIANN_DIA "H" workflows. Similarly, LFQ is coupled with ROTS in DIANN_DIA and spt_DIA "H" workflows (more details presented in Supplementary Data 5).

When applying the FP-growth algorithm to "L" workflows (bottom 50%) across different settings, we found that top3, top0, and top1 (DIA only) intensities are enriched in label-free setting "L" workflows. We also found that normalization methods "div.mean", "div.median" are enriched in FG_TMT and MQ_TMT while "quantiles.robust" is enriched in label-free setting "L" workflows. Simple statistical tools, e.g., ANOVA, SAM, proDA, and t-test are enriched[41] across all six settings (Supplementary Data 5). While this does not mean these options are to be avoided completely, we think that the inclusion of these options in workflows should be dealt with more cautiously.

**General usefulness of options across different workflow settings**
Now that we have a sense of what optimal workflows look like based on associations with high DEA performance, we investigate the value of each option per workflow step. This helps establish the usefulness of an option over various workflow settings.

We opted to compare options in each workflow step via a pairwise comparison method (see the "Methods" section). The options are compared across 5 performance metrics. Figure 4a shows an example of a pairwise comparison of expression matrix types available for DIANN_DIA and spt_DIA based on pAUC(0.01) scores. The pairwise difference of pAUC(0.01) for "dlfq-LFQ" is calculated by subtracting the pAUC(0.01) score of a workflow using the expression matrix LFQ from the pAUC(0.01) score of the corresponding workflow that has replaced LFQ with dlfq. We can infer that incorporating the dlfq option is better than using LFQ under DIANN_DIA since the

mean pairwise difference is higher than 0 (red points in Fig. 4a). Similarly, both dlfq and LFQ are superior to top1 and top3. In addition, top3 is better than top1. Based on these reciprocal comparisons, we can derive the following rank order: 1:dlfq, 2:LFQ, 3:top3 and 4:top1. These matrix types can also be ranked by the remaining 4 performance metrics in the same way. Finally, the five performance metrics are averaged to finalize the order of options in each step (see the "Methods" section).

Expression matrix type: we find that dlfq always works the best on label-free settings (not applicable to TMT data, as indicated by "NA" in Fig. 4b), except for spt_DIA (LFQ works a little better than dlfq), followed by LFQ. Under the setting, FG_TMT, the TMT-Integrator abundance (abd) is better than the Philosopher intensity (phi) or TMT-Integrator ratio (ratio) (Fig. 4b).

Normalization: it is surprising to find the "no normalization" (or "none") option works consistently well on every setting except MQ_DDA (Fig. 4c). Does this suggest normalization methods are useless? We advise caution on this direct interpretation since the datasets used are artificial spike-in datasets. Moreover, certain quantification methods such as dlfq[15], LFQ[23] and TMT-Integrator[42] have built-in peptide-level or protein-level normalization steps. Thus, no additional normalization is usually required when these quantification methods have been used. The regression-based normalization method "lossf" and the simple approach "center.median" demonstrate superior performance compared to other methods. Thus, when normalization becomes necessary, especially in cases where substantial variances are observed among the samples within the same class, we recommend employing these methods.

MVI: there is less consistency here as the performance ranks of MVI algorithms vary widely across settings (Fig. 4d). We find that missForest works well with FG_DDA (ranked 1st) and with MQ_DDA (ranked 3rd). However, missForest is also the most time-consuming algorithm compared to other top-ranked algorithms (the left heatmap in Fig. 4d shows the running time, see Methods). MinProb works well with MQ_TMT (ranked 1st), with FG_DDA (ranked 2nd) and with FG_TMT (ranked 2nd). In addition, MinProb has the highest average ranking across the six different settings (averaging ranks based on the 6 settings shown in the last column of Fig. 4d, see the "Methods" section for more details). This is not surprising, as MinProb addresses missing-not-at-random (MNAR) missingness, which plagues proteomics data[43]. MinDet (ranked 1st for DIANN_DIA and 2nd for spt_DIA) and Impseq (ranked 3rd for DIANN_DIA and 1st for spt_DIA) are good candidates for imputing DIA data, and they have good overall ranks for various settings at the same time (average rankings are 3rd and 2nd, respectively, see last column of Fig. 4d). No imputation ("none" in Fig. 4d) works the best with FG_TMT and is the 3rd top ranked MVI algorithm with MQ_TMT though it works quite bad for the other 4 settings (ranked lower than 12th among the 16 algorithms). This may be due to the low missing rates of TMT data (average missing rate = 0.2%) compared to DIA data (average missing rate of 3%) and DDA data (average missing rate of 17%, see Supplementary Data 9).

Statistical tools for DEA: limma, ROTS, DEP, and proDA consistently rank amongst the top 4 options across all 6 settings. However, proDA runs much slower without appreciable performance gains (Fig. 4e). Hence, from a practical perspective, the other 3 options are preferable. For count-based DEA, plgem works best for both FG_DDA and MQ_DDA. Though we did not compare protein intensity and spectral count-based tools in DDA data DEA directly, we find that trend-wise, spectral count-based workflows perform worse than protein intensity-based workflows (no high-performance workflows involve spectral count). In Fig. 2e, the best count-based workflows of FG_DDA and MQ_DDA are ranked lower than 1900 (among a total of 7852 workflows for each setting, see Supplementary Data 2). More details are provided in Supplementary Data 6.

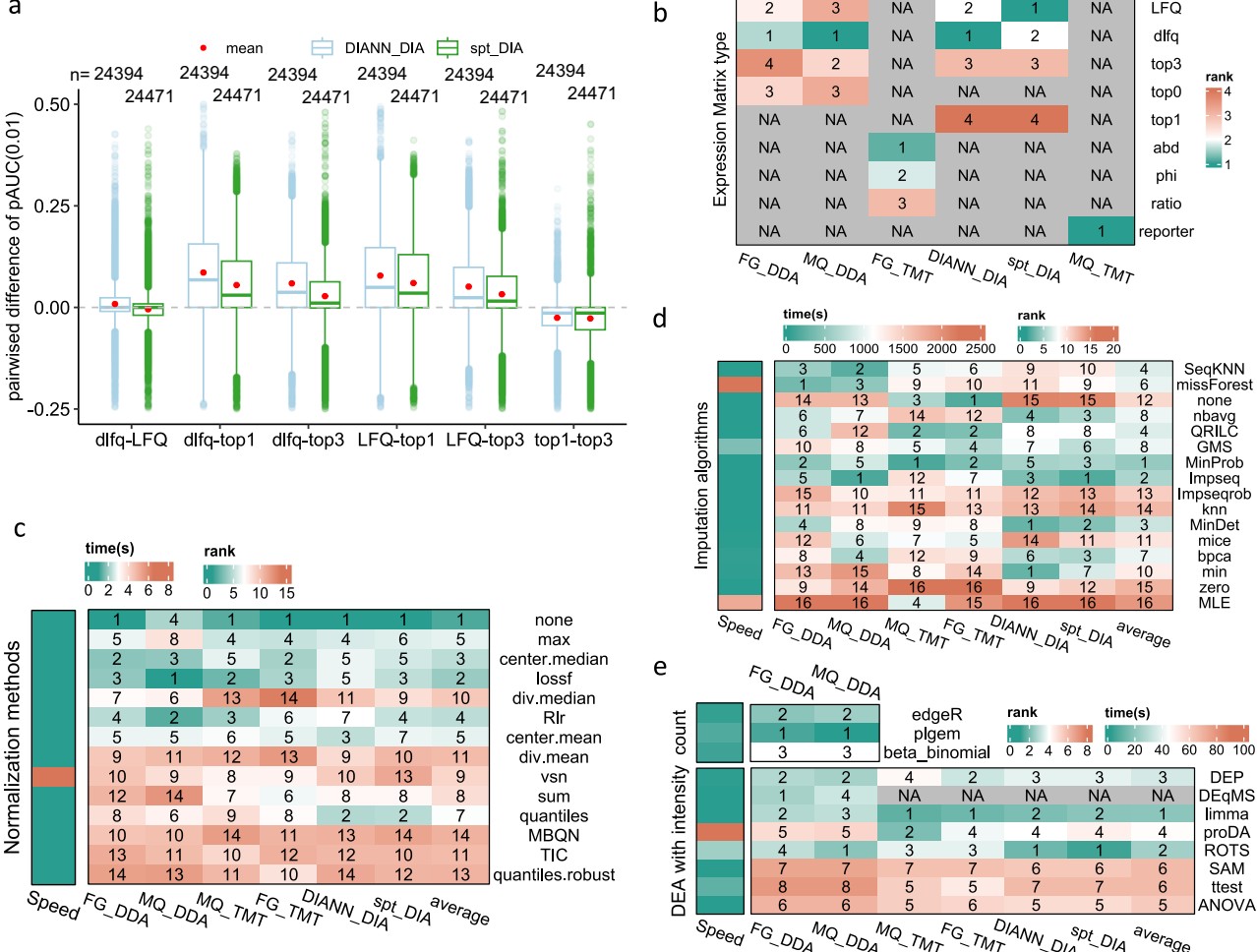

**Fig. 4 | Impact of expression matrix type, normalization, imputation, and DEA tool on differential expression analysis performance. a** Shows the distributions of pAUC(0.01) differences between a pair of matrices shown in the x-axis. The "dlfq-LFQ" in the x-axis indicates subtracting the pAUC(0.01) value of an LFQ-related workflow by the corresponding dlfq-related workflow where only matrix types are different between them. The colors of the boxplots show the settings of the compared matrix types. In the boxplots, the mean differences are marked by red points, centerline indicates the median, box limits indicate upper and lower quartiles, whiskers indicate the 1.5 interquartile range. The numbers of points for each boxplot (n) are shown above the boxplots. **b**–**e** Compare matrix types (**b**), normalization methods (**c**), MVI algorithms (**d**), and DEA tools (**e**) under different settings. The left side heatmaps in **c**–**e** present the running time (speed in second (s), in the left heatmap) of the options tested with the dataset HYEtims735_LFQ. The colors in the heatmaps show the average ranks of the options in each step based on five performance indicators. The labels in the heatmaps show the final rank positions of the options based on their average ranks. The last column of the heatmaps in **c**–**e** shows the cross-setting average ranks (averaging six ranks of different settings). Label "NA" means not applicable to the option under the specified setting below the heatmaps. Source data of (**a**–**e**) are provided as a Source Data file.

## Ensemble inference through integration of top-ranked workflows improves DEA coverage

Top workflows do not report the same differential proteins. This is especially true for workflows based on different expression matrices (Fig. 2e, see comparisons in Supplementary Fig. 5 in Supplementary Information). This led us to consider whether integration of multiple high-performing DEA workflows is synergistic, i.e., allows us to expand coverage on differentially expressed proteins. Data integration seems to be beneficial, as exemplified in earlier studies where merging intensity-based and spectral counts-based workflows produces improvements in DEA performance, even without optimization[44]. To explore the efficacy of amalgamating information from diverse workflows, we designed two ensemble inference approaches.

The first ensemble inference approach, "ens_multi-quant" integrates DEA results from top1 workflows when different expression matrix types are specified (this approach is not applicable to MQ_TMT as only one expression matrix type is available). The other "ens_topk" integrates DEA results from top-k workflows (see the "Methods" section). Detailed comparisons of TOP1, ens_multi-quant, and ens_topk

under different settings are listed in Supplementary Note 1 and Supplementary Data 7.

In general, ens_multi-quant improves performance (Fig. 5a). The performance gains based on mean pAUC(0.01) range from 1.17% to 4.61% across different settings. For mean pAUC(0.05) and mean pAUC(0.1), gains range from 0.95% to 4.46% and from 0.93% to 4.45% were obtained. We observed gains in mean nMCC ranging from 1.00% to 1.74% under settings MQ_DDA, FG_TMT, DIANN_DIA, and spt_DIA while decreases of 1.51% were seen under FG_DDA. The performance gain in terms of the mean G-mean score ranges from 5.79% to 11.14% and is even more appreciable at 11.14% (or 11.63% if in terms of the median performance) under DIANN_DIA. Although ens_topk improves DEA performances under most settings, it is inferior to ens_multi-quant. The improvement of the mean pAUC(0.01) score ranges from 0.04% to 2.93% (see more details in Supplementary Note 1 and Supplementary Data 8). The hurdle model[44] (which transforms p-values to z-values and then combines the z-values into a $\chi^2$ statistic) for p-value integration is more frequently applied than the Fisher's method (see Supplementary Data 8).

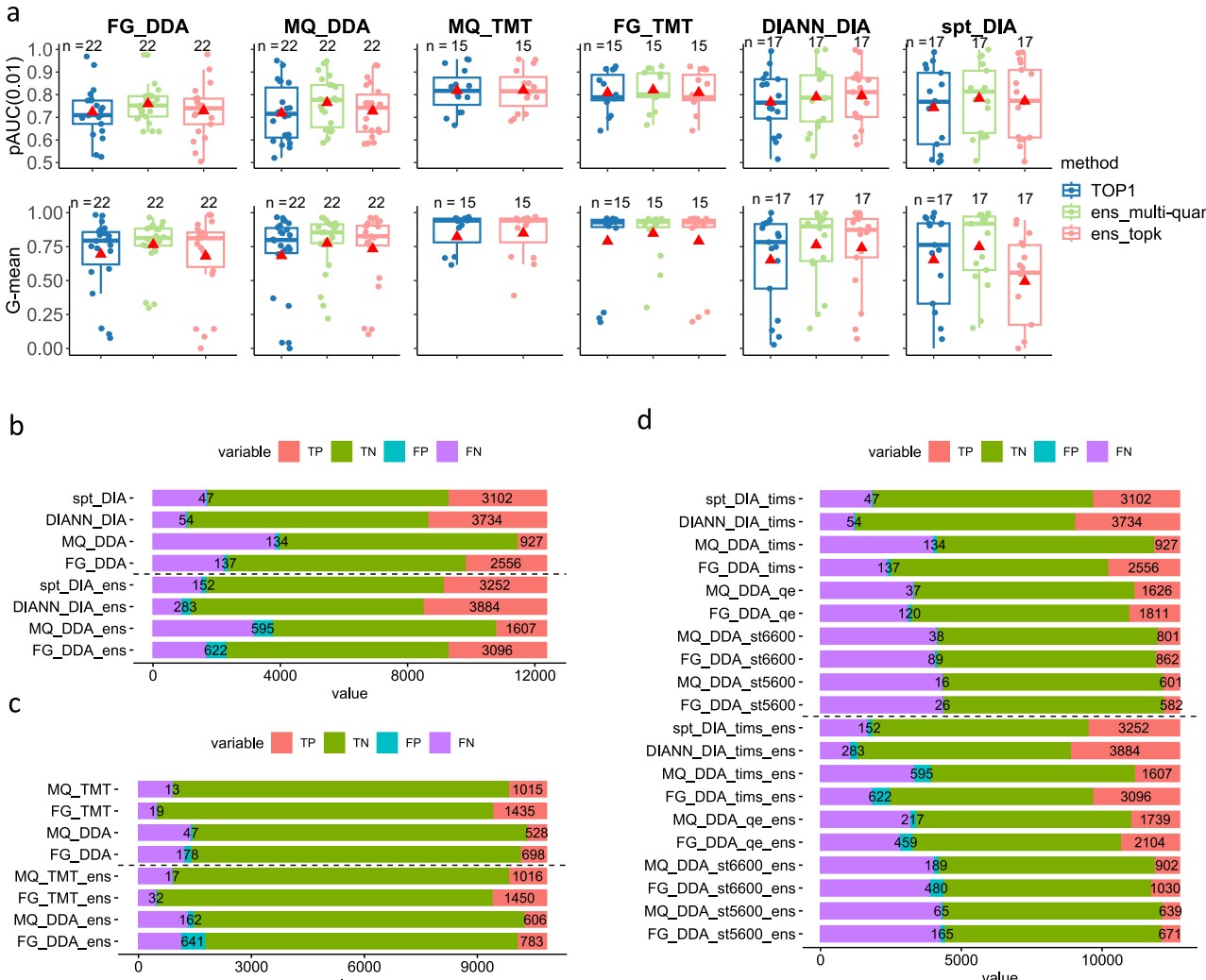

**Fig. 5 | Performance by ensemble inference is compared when settings and instruments are varied. a** Compares the performance by ensemble inferences and those by top 1 single workflows under different settings. The red triangle shows the mean performance. TOP1 refers to the best workflow specific to a given setting. Ens_multi-quant is an ensemble approach combining top1 workflows using different expression matrices. Ens_topk is an ensemble approach integrating the top-ranked k workflows. In the boxplots, the mean performances are marked by red triangles, centerline indicates the median, box limits indicate upper and lower quartiles, whiskers indicate the 1.5 interquartile range. The numbers of points for each boxplot (n) are shown above the boxplots. **b–d** Compares the performances by the top-ranked workflows and the ensemble inference workflows across different settings. The stacked bar plots in **b–d** show the absolute counts of reported true positive (TP), true negative (TN), false positive (FP), and false negative (FN) for differentially expressed proteins (DEPs). **b** Compares the best workflows of DDA and DIA settings and investigates the ensemble inference's improvement in DEP detection. The label of the bar gives the setting type, e.g., FG_DDA, or whether ens_multi-quant is used or not. Similarly, **c** compares the DDA and TMT and ensemble inference-enabled DDA and TMT best workflows. **d** shows the comparisons of instrument-specific (datasets generated by a specific machine) best single workflows (labeled using a setting as prefix and no suffix of "-ens", e.g., spt_DIA_tims) with instrument-specific best ens_multi-quant workflows (suffix as "-ens") in the detection of DEPs. Source data of **a–d** are provided as a Source Data file.

## Comparisons within quantification settings and within instrument types for differential expression analysis

Since we now have a means of predicting optimal pipelines across a variety of settings, we may ask when optimized to maximize performance if some settings are superior to others when it comes to DEA. We compared workflows under labeled (FG_TMT and MQ_TMT) and label-free settings, e.g., label-free DDA (FG_DDA and MQ_DDA) and label-free DIA (DIANN_DIA and spt_DIA).

We chose datasets that are compatible with the evaluated settings (see Table 1). The dataset HYEtims735_LFQ and HYEtims735_DIA (generated with the same samples and instrument but acquired under DDA and DIA mode, respectively (see Table 1 and Supplementary Data 9) were used to compare settings FG_DDA, MQ_DDA, DIANN_DIA, and spt_DIA. To facilitate comparisons across settings where

differential proteins are reportable, we pooled a reference protein list by merging proteins detected under different settings. Then, for each best workflow specific to a setting (prefixed top1), we calculated the coverage of this reference protein list (see Methods).

From the top 4 bars of Fig. 5b, we can see that the two DIA settings DIANN_DIA and spt_DIA work better than the DDA settings FG_DDA and MQ_DDA, with more true positives (TPs) and fewer false positives (FPs). For example, DIANN_DIA detected 1178 more TPs (3734−2556) than FG_DDA but found less than half the number of FPs (54 vs. 137). Compared to MQ_DDA, the TP number reported by DIANN_DIA is nearly quadrupled (3734 vs. 927), and the FP number is halved (54 vs. 134). Compared against DIANN_DIA, spt_DIA detected 632 less TPs (3734−3102) and 7 less FPs (54−47). Based on another two pairs of datasets (HEqe408_LFQ, HEqe408_DIA) and (HYtims134_LFQ,

HYtims134_DIA), we can observe similar performance that DIA works better than DDA and FG_DDA works better than MQ_DDA. In addition, DIANN_DIA works better than spt_DIA in 2 out of the 3 pair comparisons (see Supplementary Note 1 and Supplementary Data 10).

Only the dataset pair (HYqfl683_LFQ, HYqfl683_TMT11) is applicable to compare with DDA settings and TMT settings. As seen from the top 4 bars of Fig. 5c, the TP numbers found by the TMT settings (1435 TPs for FG_TMT and 1015 TPs for MQ_TMT) are nearly doubled that of each DDA setting (698 TPs for FG_DDA and 528 TPs for MQ_DDA). Again, much fewer FPs were found by the TMT settings than the DDA settings (19 and 13 for FG_TMT and MQ_TMT vs. 178 and 47 for FG_DDA and MQ_DDA).

Given the multiple choices for the instruments, e.g., SCIEX Triple TOF machines, Orbitrap QE-HFX, and TimsTOF pro, we ask which one can maximize DEA performance. We adopted the 5 datasets from the same project (ID PXD028735, see Table 1) to conduct such a cross-instrument-cross-setting comparison (settings are indicated by the prefixes, and instruments are indicated by the suffixes of the labels for the top bars in Fig. 5d). The TimsTOF pro (suffix as _tims) worked the best, discovering more proteins (higher proteome coverage) and more TPs no matter under DDA or DIA acquisition modes (see Supplementary Data 11). SCIEX Triple TOF5600 (suffix as _st5600) detected smaller numbers of TPs under both FG_DDA (582) and MQ_DDA (601) quantification settings, which account for only about 20% of the numbers of TimsTOF pro with DIA and 25% of the numbers of TimsTOF pro with FG_DDA. However, this machine also detected fewer FPs (16 with MQ_DDA and 26 with FG_DDA). SCIEX Triple TOF6600 (suffix as _st6600) worked better than SCIEX Triple TOF5600 with about 200 more TPs and also more FPs (22 increased by MQ_DDA and 63 by FG_DDA). Orbitrap QE-HFX (suffix as _qe) doubled the sizes of detected TP numbers compared to SCIEX Triple TOF6600 and kept the same or increased a little in FP numbers. It even detected more TPs than TimsTOF pro under the quantification setting MQ_DDA (1626 vs. 927). In conclusion, if possible, TimsTOF pro is recommended for proteomics data analysis, otherwise Orbitrap QE-HFX is suggested.

We are also interested in whether the use of ensemble inference can narrow the performance gaps caused by different settings or instruments. The bottom bars in Fig. 5b–d display the TP and FP numbers achieved by the best ensemble inference workflows (suffix as _ens), and these numbers are compared with the performances of the single best workflows (see Methods). We can see that the ensemble inference recovered more TPs than the single workflows except for the TMT data analysis (FG_TMT_ens and MQ_TMT_ens only find 1 and 15 more TPs than FG_TMT and MQ_TMT respectively, see Fig. 5c). For example, 150 (3884-3734) and 150 (3252-3102) more TPs were reported by DIANN_DIA_ens and spt_DIA_ens respectively than by DIANN_DIA and spt_DIA (Fig. 5b). Impressively, the TP number gap due to using Orbitrap QE-HFX and TimsTOF pro with setting MQ_DDA can be completely closed, where MQ_DDA_ens reached the TP number from 927 (MQ_DDA) to 1607 (Fig. 5b), which is almost the same as MQ_DDA_qe and FG_DDA_qe's 1626 and 1811 (Orbitrap QE-HFX, Fig. 5c). Similarly, ensemble inference has let Orbitrap QE-HFX (FG_DDA_qe_ens) to catch up with the performance of TimsTOF pro (FG_DDA_tims) in detecting TPs (2104 vs. 2556, Fig. 5d). In addition, FG_DDA_ens (3096) detected similar numbers of TPs compared to spt_DIA (3102) and spt_DIA_ens (3252). However, ensemble inference increases the false positive rate at the same time. For instance, although FG_DDA_ens found 540 (3096-2556) more TPs than FG_DDA, 485 (622-137) more FPs were also recommended (Fig. 5b). FG_DDA_st6600_ens found 168 (1030-862) more TPs than FG_DDA_st6600 with a sacrifice of detecting 391 (480-89) more FPs (Fig. 5c). We can see more successes in closing performance gap (setting-specific) albeit with smaller false positive rate increases when leveraging on ensemble inference, e.g., spt_DIA_ens recovered 214 more TPs but with only 26 more FPs comparing to spt_DIA testing on dataset HYtims134_DIA (see Supplementary Note 1). These evidences suggest that integrating different

quantification methods is a promising approach for improving DEA performance.

## Discussion

### Recommendations and resources for selecting optimal workflows

To the best of our knowledge, this is the most extensive benchmarking study to determine if optimal workflows in proteomics are predictable. Based on our findings, we make the following recommendations for specific quantification settings, considering average performance, running time, and generalizability:

- For label-free DDA data quantified by FragPipe (i.e., setting FG_DDA), we recommend a workflow combining protein directLFQ intensity, no normalization (refers to no additional normalization to data extracted with default quantification settings), SeqKNN for MVI, and DEqMS or ROTS (or limma, if running time is a concern) for DEA.
- For label-free DDA data quantified by Maxquant (i.e., setting MQ_DDA), we recommend a workflow combining protein directLFQ intensity, no normalization, Impseq for MVI, and DEqMS or limma for DEA.
- For label-free DIA data quantified by DIA-NN (i.e., setting DIANN_DIA), we recommend a workflow combining protein directLFQ intensity, no normalization, MinDet for MVI, and limma for the DEA (ROTS can be an alternative DEA method if running time is not considered).
- For label-free DIA data quantified by Spectronaut (i.e., setting spt_DIA), we recommend a workflow combining directLFQ intensity, no normalization, Impseq for MVI, and ROTS for DEA.
- For TMT data quantified by FragPipe (i.e., setting FG_TMT), we recommend a workflow combining TMT-Integrator abundance, no normalization, SeqKNN for MVI, and limma for DEA.
- For TMT data quantified by Maxquant, we recommend a workflow combining reporter intensity, no normalization, bpca for MVI, and proDA (or limma, if running time is a consideration) for DEA.
- For expression matrices without acquisition platform or quantification information, we recommend a workflow combining no normalization, lossf, Rlr[45], center.median for normalization options, MinProb, SeqKNN, Impseq or MinDet for MVI, and limma or ROTS for DEA.
- Label-free DIA and TMT are recommended for proteomics experiment design as they are more accurate, with lower missing rates, and have higher proteome coverages compared to label-free DDA. However, we don't have data to compare between label-free DIA and TMT directly. The choice of label-free DIA or TMT in experiment design should consider the level of multiplexing required, the desired dynamic range, and the budget.

For details on how we arrived at the recommendations above, please refer to previous sections discussing the benchmarking (Supplementary Note 1), the frequent pattern mining, options comparison in workflow steps, ensemble inference and cross-setting comparisons.

Next, based on our data assemblage and benchmarking results, we developed the following resources to help users identify optimal workflows:

- To choose the best workflows based on our benchmarking results, we implemented a webserver OpDEA (http://www.ai4pro.tech:3838), a standalone tool and companion R package (https://github.com/PennHui2016/OpDEA) to guide which workflow should be used after specifying the quantification platform and/or the preferred expression matrix types. Users can conduct single or multiple DEA analyses with our server, standalone tool or the R package directly. The results from multi-workflow-based DEA

can be easily compared and integrated through our tool. The datasets we used are also available through our webserver.

- Should users wish to try ensemble approaches, we recommend using ens_multi-quant with label-free DDA and DIA data. We also recommend combining the two advanced quantification methods, directLFQ and MaxLFQ, with top0 for DDA data or with top3 for DIA data via the hurdle approach.
- Should users require more flexibility and wish to inspect or access the data for themselves, they can access the benchmarking results directly in Supplementary Data 2. There is an additional benefit to using the data directly–users can benchmark their own self-defined workflows against our data's workflow ranking to evaluate and improve their in-house workflows.

### Ranked workflows provide a means for the identification of algorithms associated with high-performance

Our study highlights the significant impact of quantification methods (expression matrix type, especially for DIA data), normalization methods, and DEA tools on DEA performance and, thus, the importance of paying attention to the selections in these steps. Moreover, when a new method or algorithm is published, it is often tested within a narrow or specific context, which may be unduly favorable to itself. By comprehensively investigating the impact of tool selections in each important step of a workflow, we have a means to assay if an algorithm (or method) representing options for a workflow step is valuable. Aggregating our observations, we find:

- Expression matrix types from advanced quantification methods such as directLFQ and MaxLFQ are superior to spectral counts and topN, e.g, top1, top3 or top0 in DEA.
- Check data variance in advance to determine the need for data normalization. Normalization performed surprisingly badly and may even worsen the DEA performance. Some advanced quantification methods (directLFQ and MaxLFQ) have built-in peptide-level or protein level normalizations, thus usually no additional normalization is required. We recommend checking the distributions of the expression matrix in advance to determine whether normalization is required (and not just do it routinely, especially when it is not needed). If needed, regression-based normalization methods, e.g., lossf and Rlr, and center.median or center.mean are good options.
- Simple MVI algorithms such as MinProb and MinDet work well with DIA and TMT data. This is probably because most missingness in proteomics data is attributable to missing-not-at-random[43,46]. However, for bigger projects with many biological and technical replicates, removing high-missing-rate replicates or ignoring the missing values during the statistical analysis may be safer for avoiding bias introduced by MVI algorithms[47,48].
- DEA methods limma and ROTS are universally good, performing well in any quantification setting. proDA, DEqMS, and DEP are designed specifically for proteomics data and are superior to generic (albeit simple) statistical methods such as t-tests and ANOVA. However, proDA is very slow and may not be ready for big data projects comprising thousands of samples.

### Limitations

**Algorithm parameter tuning.** We maintained the default parameters for every algorithm/method to maintain consistency in the evaluation process. Moreover, these default parameters are supposed to be optimized. It is possible that some parameter optimizations may result in some rank changes, especially if the parameter changes are necessary to fit the raw data better.

Several MVI and DEA tools in our evaluations do have tuneable parameters, such as "k" for "knn" imputation[49] or the number of bootstrapping for ROTS[6]. While adjusting these parameters can affect the final DEA performance, it also increases the time cost and may

require additional data for validation. This is not feasible for analysing real-life data.

To avoid an explosion of simulations, it is useful to look into adaptive parameter tuning approaches in future work to consider a wider variety of options in our workflow simulations.

### Spike-in data

We used spike-in data to benchmark workflows. Spike-in datasets mainly simulate technical variation by producing technique replicates for each sample. In real-world proteomics data, biological variation always exists and may manifestly quite differently from technical variance. The inability to simulate biological variation exactly may introduce some degree of bias to our benchmarking results. However, there is currently no universally accepted way of simulating biological replicates. Attempting to do so at this time may invite skepticism. There are some promising new approaches based on deep generative models[50] for in silico simulation of biological variation, we will look at this closely for inclusion in our next round of benchmarking efforts.

### Fidelity and applicability to the real world

Although we try to test as many datasets and methods as possible, it does not mean the predictions and recommendations built on these will work on any data. We cannot guarantee it. Real-world data is complex (e.g., with more complex missing patterns, e.g., mixed by MNAR and MCAR, and bigger cross-run variances) and covers many situations (scenarios or sources or variation) that cannot exist in our relatively simpler simulations. Though the conclusions drawn here may not work universally, still, we think that OpDEA provides a useful starting point as a recommendation resource. Researchers will no longer need to simply copy a workflow from some other publication, with no knowledge of whether the workflow really works well for their own data. In this regard, OpDEA has the potential to positively enhance proteomics research.

### Ensemble inference as a prelude into the potential of multi-view learning in proteomics

The ensemble inference methods we propose suggest there is value in data maximization. The ensemble inference viewpoint works by integrating information across multiple perspectives (or "views") of proteomics data. This approach takes its inspiration from a branch of AI known as multi-view learning.

Our proposed ensemble inference method relies on simple p-value combining approaches, e.g., the hurdle approach, to improve DEA performance. While this approach, particularly ens_multi-quant, increases coverage of differentially expressed proteins (DEPs), it also increases false positives (Fig. 5b–d). Still, we think this is promising: As multiple quantification results show complementary evidence in recovering true positives, we think instead of relying merely on p-value combining, we should shift to and explore true multi-view learning approaches involving machine learning and AI.

Multi-view learning combines multiple views of the same object to improve generalization performance[51], which has been widely used in computer vision[52], recommendation system[53], bioinformatics[54], etc. In proteomics, the traditional approach is to only rely on one view of the data (depending on what is the favored workflow). We may regard the multiple quantification expression matrices as multiple views and learn via multi-view learning algorithms to improve DEA performance. With the advent and ready accessibility of advanced computational frameworks, we can easily execute various workflows processing different views in parallel and integrate these perspectives with multi-view learning AIs to improve the coverage, correctness, and interpretability of findings. We think there is value in pursuing novel AI approaches to maximize proteomics data by harnessing its multiple "views". Being able to predict the optimal workflows compatible with each view is a necessary step.

## Methods

### Quantitative analysis of mass spectrometry-based proteomics data

**DDA data quantification.** FragPipe v20.0 was used with its default quantification parameters. MSFragger-3.8[25] was adopted for database search (using default parameters), in which contaminants and decoy (sequence reverse) protein sequences were added to corresponding libraries by Philosopher v5.0.0[40], e.g., 48 human UPS1 proteins + reviewed UP000002311 Saccharomyces cerevisiae proteome from uniport database[55] for YUltq099_LFQ, YUltq819_LFQ and YUltq006_LFQ datasets. More details about the databases used for database search in our benchmarking are listed in Supplementary Table 1 of Supplementary Data 9. MSBooster-1.1.11[56], Percolator v3.06[57] and ProteinProphet[58] were used for peptide identification and protein inference. For quantification, IonQuant-1.9.8[59] with the match between runs (ion FDR of 0.01)[60], normalize intensity across runs, and MaxLFQ (minimum ions of 2)[23] were used. For label-free DDA data, we ran the quantification twice with the parameter "Top N ions" set as 0 and 3, respectively, to obtain the top0 (all precursors are considered) and top3 (only top3 most intense precursors are considered) quantification intensities. Details of the parameters are listed in Supplementary Note 2. The quantification results are stored in files containing identified peptides or proteins with their spectral counts or intensities and other annotation information. We extracted expression levels of identified proteins in the form of spectral counts or intensities and organized them as a matrix by setting proteins as rows and samples as columns. The cells hold the expression levels. This data structure is referred to as an expression matrix, and it holds varied expression information. In fact, four types of expression matrixes are obtainable from FragPipe, including spectral counts, top0 intensities, top3 intensities, and MaxLFQ intensities calculated by the MaxLFQ algorithm[23]. In addition, the directLFQ intensities were extracted with the directLFQ algorithm[15] (with the published directlfq tool: https://github.com/MannLabs/directlfq).

MaxQuant v2.1.0.0 was used as an alternative quantification platform for label-free DDA data[13]. Andromeda[61] was applied for database search. We kept most parameters default (see Supplementary Note 2 for more details) and used the same reference library as FragPipe. Similarly, match between runs and MaxLFQ label-free quantification algorithm were used. Again, spectral counts, protein top0 and top3 (set the Top N peptides as 0 and check the Top3 option of "Advanced site intensities") intensities, and MaxLFQ intensities can be acquired directly from Maxquant's "proteinGroups.txt" file for DEA. The directLFQ intensities were also extracted with the directLFQ algorithm[15]. We named the quantification analysis of DDA data with FragPipe the quantification setting of "FG_DDA" and the quantification analysis of DDA data with Maxquant the quantification setting of "MQ_DDA".

**DIA data quantification.** For DIA data, DIA-NN v1.8.1 was used for quantification under its default parameters (see details in Supplementary Note 2), and libraries were predicted from corresponding databases[26]. Match between runs was checked, and we ran DIA-NN under "Optimal results" mode. The top1 (most intense precursor-based quantification), top3, and MaxLFQ intensities were extracted by the "iq" R package[62] (https://github.com/tvpham/iq), and the directLFQ intensities were extracted by the directLFQ algorithm from the report file of DIA-NN.

Spectronaut 18[27] was used as an alternative platform for DIA data quantification. We used the same sequence databases as DIA-NN used and predicted libraries by Spectronaut 18 (used the directDIA[27] workflow for quantification). Similar to DIA-NN, we extracted top1, top3, MaxLFQ intensities with the "iq" package and extracted directLFQ intensities with the directlfq tool from the report file of Spectronaut 18. The detailed parameters are described in the Supplementary Note 2.

Similarly, the quantifications of DIA data with DIA-NN and Spectronaut were named quantification settings "DIANN_DIA" and "spt_DIA" for convenience.

**TMT data quantification.** Both FragPipe (v20.0) and Maxquant (v2.4.4.0) were adopted for TMT quantification. For FragPipe, similar to the DDA data quantification, we enabled the MSBooster, and per-colator and ProteinProphet were used for peptide identification and protein inference. The TMT-Integrator and the Philosopher were used for quantification, with most of the parameters as default[40]. Detail descriptions of the parameters are available in Supplementary Note 2. We extracted the TMT-Integrator abundance and TMT-Integrator ratio from the TMT-Integrator reports (in the tmt-report folder, https://github.com/Nesvilab/TMT-Integrator), and the Philosopher intensities were extracted from the "protein.tsv" file of the specified result folder. For Maxquant, we selected the quantification type according to the quantification level specified in the publication of the dataset, e.g., the "Reporter ion MS2" with "10plex TMT" was selected for analyzing dataset HEqe277_TMT10. Most of the other parameters were set as default (see Supplementary Note 2 for detailed description). Only the "Reporter intensity corrected" values were extracted from the "proteinGroups.txt" file. We used "FG_TMT" and "MQ_TMT" to indicate the quantification analysis of TMT data with FragPipe and Maxquant, respectively.

### Expression matrix type, normalization, imputation, and DEA statistical tools used in benchmarking

For label-free DDA data quantified with FragPipe and Maxquant, five types of expression matrices are obtainable, including spectral counts, protein top0 and top3 intensities, MaxLFQ intensities calculated by MaxLFQ algorithm[23], and the directLFQ intensities[15]. As for DIA quantification with DIA-NN or Spectronaut, four types of protein intensities are obtainable, including top1, top3, MaxLFQ (LFQ), and directLFQ (dlfq). For TMT data quantified by FragPipe, the TMT-Integrator abundance (abd), TMT-Integrator ratio (ratio), and Philosopher intensity (phi) are available, while for TMT data quantified by Max-quant, only the reporter intensity (reporter) was used (see above section and Figs. 1a, 2e and Supplementary Fig. 2).

Two preprocessing procedures (normalization and imputation) are conducted (Fig. 1a). Many methods have been proposed to normalize expression data and impute missing values. It's impossible to evaluate every normalization and imputation method. And so, only popular and readily accessible ones were used.

For normalization methods, we used those found in the MSnbase v2.22.0 R package[63] including "sum", "max", "center.mean", "center.median", "div.mean", "div.median", "quantiles", "quantiles.robust" and "vsn"(variance stabilization)[64]. The two regression-based normalization methods "loess"[36,37] and "Rlr"[45], the total ion current normalization (TIC) and the Mean/Median-balanced quantile (MBQN)[65] normalization methods were also evaluated (we conclude them in Fig. 1d). If no normalization is used, then we designate the normalization method as "none".

For MVI methods, we also used those found in the MSnbase v2.22.0 R package, such as "bpca" (Bayesian principal component analysis)[66], "knn" (k-nearest neighbors)[49], "QRILC" (quantile regression imputation of left-censored data)[47], "MLE" (maximum likelihood estimation)[67], "MinDet" (deterministic minimum)[19], "MinProb" (probabilistic minimum)[19], "min"[68], "zero"[19] and "nbavg" (neighbor averaging)[69], and another six popular imputation methods such as mice (Multivariate imputation by chained equations)[70], Impseq (Sequential imputation)[17,18], Impseqrob (Robust sequential imputation)[71], GMS (generalized mass spectrum missing peaks abundance imputation)[38], SeqKNN (sequential KNN imputation)[16,17], missForest (random forest)[35]. Again, if no imputation is performed, we designate the imputation method as "none".

We took DEA tools from those implementations in Bioconductor v3.15[72] and those we collected from published literature. From Bioconductor (Bioconductor−BiocViews: https://www.bioconductor.org/packages/release/BiocViews.html#__DifferentialExpression), we used DEA tools designed specifically for proteomics data. We chose them by popularity, given their download numbers (their ranks are shown in brackets; smaller means more popular). Our list includes DEP (272)[73], DEqMS (452)[34], plgem (769)[74], proDA (480)[75]. Popular and commonly used DEA tools limma (15)[76] and edgeR (25)[77] are not originally designed for proteomics data but are also downloaded quite frequently and made it into our list. Other general tools such as ANOVA[20], *t*-test[21], beta_binomial[78], MSstats[79], SAM[21], and ROTS[6] are collected from literature[7,44] or related websites. Each DEA tool has a preferred expression matrix type: edgeR, plgem, and beta_binomial are good with spectral counts; while other tools are either agnostic or work well directly with protein intensities. Some other DEA tools are more demanding: MSstats and DEqMS require additional information beyond protein intensities, whereas MSstats require feature-level data[79], and DEqMS needs peptide or spectral counts[34]. More descriptions of the normalization, imputation, and DEA methods are presented in Supplementary Note 2.

In summary, for label-free DDA data, based on quantification results from FragPipe or Maxquant, 7852 workflows are compared (see Supplementary Data 2 for detailed workflow lists), and for label-free DIA data quantified with DIA-NN or Spectronaut, 6284 workflows were compared (Supplementary Data 2). In the case of TMT data quantified by FragPipe, 4720 workflows were compared. In the case of Maxquant used for quantification, 1584 workflows were compared.

## Performance evaluation metrics

The prepared datasets were processed to produce corresponding expression matrices (Fig. 1). These expression matrices were then analyzed by workflows combining a particular expression matrix type, normalization method, MVI algorithm and DEA tool. The Benjamini−Hochberg method[80] is used for FDR control if no built-in method is applied.

Two types of performance measures were used, including partial area under receiver operator characteristic curves (pAUC) and confusion matrix-based metrics:

**Partial area under receiver operator characteristic curves (pAUC).** The receiver operator characteristic curve (ROC) is generated by plotting true positive rates (TPR) against false positive rates (FPR) under various thresholds to classify samples into two categories[28]. The area under the ROC (AUC) is a widely used performance indicator to evaluate the power of a classifier in classification tasks. The partial AUC is proposed for restricting the evaluation of given ROC curves in the range of FPRs that are considered interesting for diagnostic purposes[28]. In our performance evaluations, the confidence score of a given protein to be a true differentially expressed protein is calculated by 1-*q*-value, where *q*-value is its Benjamini−Hochberg adjusted *p*-value. The FPR is restricted to FPR ≤ 0.01, FPR ≤ 0.05, and FPR ≤ 0.1, respectively, to calculate three performance indicators, i.e., pAUC(0.01), pAUC(0.05), and pAUC(0.1) for a stricter performance evaluation.

**Confusion matrix-based evaluation.** The classification metrics calculated from the confusion matrix, such as the normalized Matthew's correlation coefficient (nMCC)[29] and the geometric mean of specificity and recall (G-mean)[30] were used as additional evaluation indicators where nMCC and G-mean are calculated as follows:

$$MCC = \frac{TP \times TN - FP \times FN}{\sqrt{(TP+FP)(TP+FN)(TN+FP)(TN+FN)}} \quad (1)$$

$$nMCC = \frac{1}{2}(MCC + 1) \quad (2)$$

$$Specificity = \frac{TN}{FP + TN} \quad (3)$$

$$Recall = \frac{TP}{TP + FN} \quad (4)$$

$$G - mean = \sqrt{Specificity \times Recall} \quad (5)$$

where True Positive (TP) is the truly differentially expressed protein has $|logFC| \ge log2(1.5)$ and *q*-value < 0.05, False Positive (FP) is the truly non-differentially expressed protein has $|logFC| \ge log2(1.5)$ and *q*-value < 0.05, True Negative (TN) is the truly non-differentially expressed protein that cannot pass the threshold of $|logFC| \ge log2(1.5)$ and *q*-value < 0.05 while False Negative (FN) is the truly differentially expressed protein that cannot pass the threshold of $|logFC| \ge log2(1.5)$ and *q*-value < 0.05. The *q*-value is the FDR-adjusted *p*-value.

The metric pAUC is a rank-based global performance indicator where its value is calculated by specifying an FPR range, e.g., ≤0.01. However, in a DEA task, we may always be interested in the reliability of detected differentially expressed proteins at a given logFC and *q*-value threshold but not an FPR range, which cannot be deciphered from the pAUC score. Thus, we combined the confusion matrix-based metrics nMCC and G-mean with the pAUC scores to conduct a more comprehensive performance evaluation.

## Workflow ranking

Workflows were ranked by five performance metrics, including pAUC(0.01), pAUC(0.05), pAUC(0.1), nMCC, and G-mean separately. A workflow's final rank is given by averaging its five ranks, as shown below.

$$rank_{final} = \frac{rank_{pAUC(0.01)} + rank_{pAUC(0.05)} + rank_{pAUC(0.1)} + rank_{nMCC} + rank_{G-mean}}{5}$$

$$(6)$$

## Leave-one-dataset-out cross-validation

To confirm whether our benchmarking results can be used for recommending optimal workflows for newcoming datasets, we conduct the leave-one-dataset-out cross-validation (LODOCV). Taking the label-free DDA data as an example, there are 12 datasets for benchmarking (see Table 1 for more details). In a LODOCV, each time we use 11 out of 12 datasets to rank workflows (benchmarking), the contrasts in the remaining dataset are regarded as newcoming datasets. We calculated the Spearman correlation coefficient between the workflow ranks based on benchmarking with the 11-datasets and the true workflow ranks of the newcoming data. The higher the correlation is, the more accurate recommendations could be made with our benchmarking.

## Kruskal−Wallis test

The biggest contrast number of the 24 gold standard datasets is 3. Thus it is impossible for us to check whether the benchmarking is sensitive to the dataset used for performance evaluation with statistical methods (the group size is no bigger than 3). The DDA, DIA, and TMT data were generated by different types of instruments (see Table 1), e.g., newer ones such as Tims TOF or older ones such as LTQ Orbitrap Velos. For the 20 contrasts (SCIEX Triple TOF5600 and SCIEX Triple TOF6600 were excluded) of the 12 DDA datasets, we group them into 3 categories, including Tims TOF pro ($N = 8$), LTQ-Orbitrap/Velos ($N = 7$) and other Orbitrap machines ($N = 5$). Each of the 20 contrasts was used to rank the workflows based on the 5 indicators (above benchmarking method). Then, we conducted the Kruskal−Wallis test[32]

to check whether a workflow's rankings based on the three groups of contrasts have the same median at a threshold of $p$-value $\leq 0.05$. Similarly, for the 17 contrasts of the DIA data, 2 instrument groups are available such as the Tims TOF pro ($N = 7$) and the other Orbitrap machines ($N = 10$). For the TMT data, we cannot split them into groups with the minimum size requirement of $N \geq 5$, so no Kruskal–Wallis test was conducted.

## CatBoost classification and the linear model

The CatBoost is a new gradient boosting toolkit supporting solving categorical features-based machine learning tasks[33]. To confirm that the performance level of a workflow is predictable with the information of its option in each step, 10-fold cross-validation was applied to validate the classification accuracy of a CatBoost classifier (with the python package of CatBoost-1.2.2) in classifying workflows. We first labeled the workflows as "H", "RH", "RL" and "L" if their ranking positions are located in the top 5%, 5–25%, 25–50%, and 50–100%, respectively. Then, the workflows were encoded as feature vectors with their options in each step as categorical features. At last, the workflows were split into 10-folds randomly, and in each of the 10 rounds, 9 out of the 10 folds data were used to train the CatBoost classifier and predict the performance levels of the workflows in the remaining fold. The hyperparameters were mostly left as default, e.g., "iterations" of 1000, "depth" of 6, "l2_leaf_reg", etc., while the "learning rate" was set as 0.3 for speeding up the training. The metrics MCC and F1-score were used as performance indicators and the average performance of the 10-round was recorded. The feature importance was evaluated by fitting the CatBoost classifier with all the workflows and their labels. The F1-score is calculated by:

$$\text{Precision} = \frac{\text{TP}}{\text{TP} + \text{FP}} \quad (7)$$

$$F1 = \frac{2 * \text{Recall} * \text{Precision}}{\text{Recall} + \text{Precision}} \quad (8)$$

We also fitted linear models to check the interactions and synergies of predictor variables, i.e., the options in each step (with dummy coding where the categorical variables, e.g., the normalization, are recoded into a set of separate binary variables) and the response variable, i.e., the ranking score of a workflow. The ranking score is calculated by:

$$\text{ranking score} = N - \text{ranking position} \quad (9)$$

where $N$ means the number of workflows considered, e.g., $N = 7852$ for workflows accepting DDA data analyzed by FragPipe. Ranking position means the number indicating the order of the performance of a workflow among all the workflows. Ranking position of 1 means the best, and smaller the better. The "Estimate" values (also known as coefficients) are used to indicate the average increase in the response variable associated with a one-unit increase in the predictor variable, assuming all other predictor variables are held constant. Bigger "Estimate" absolute value means higher impact of the predictor variable on the response variable. Positive "Estimate" value means positive impact; otherwise negative impact (see Supplementary Note 1). The $p$-value $< 0.05$ indicates the predictor variable is significant to the response variable. The interactions between categorical variables are discussed in Supplementary Note 1.

We also extracted the classic ANOVA (analysis of variance)[20] table from the above linear model to check the affections of categorical variables (i.e., expression matrix type, normalization, imputation, and DEA tool) on the response variable. The $F$-value (the ratio of the variation between sample means to the variation within the samples) calculated from the $F$-test[81] was used as the indicator of the impact of

category variable changes on the response variable. The $p$-value $< 0.05$ indicates the sample means are significantly different (see Supplementary Note 1).

## Frequent pattern growth algorithm

The frequent pattern growth algorithm (FP-growth)[39] is a frequent-pattern tree (FP-tree)-based mining method for mining the complete set of frequent patterns by pattern fragment growth. It is efficient and scalable for mining both long and short-frequent patterns. The Python package mlxtend-0.23.0 was used to implement the FP-growth algorithm. The support ratio (SR) is defined as the fraction of the total items containing the pattern and is used to measure the popularity of a pattern in all available items. We set the SR threshold as 0.1 (10%). The FP-growth algorithm is used to extract patterns from both high-performing workflows (with the label of "H" where they are ranked at top 5%) and low-performing workflows (bottom 50% workflows with labels of "L" workflows).

## Comparisons among choices in a single step of a workflow

After obtaining quantification results from a quantification setting, such as analysis of label-free DDA data with FragPipe (setting of FG_DDA), a comprehensive DEA workflow integrates several key selection steps, including:

(a) An expression matrix that contains the expression levels of identified proteins;

(b) A normalization method to reduce bias or noise;

(c) An algorithm for imputing missing values in the selected expression matrix;

(d) A DEA tool for conducting the final differential expression analysis.

Each step plays a crucial role. To examine the impact of a particular step, we simply maintained the options for other steps while varying the options of the step under investigation. To compare any two options for a given step, e.g., protein top1 intensity and protein MaxLFQ intensity in step (a), we calculated performance differences of workflow pairs where they are alike in every other way except the choice of option. Different options in each step can be ranked by their pairwise comparisons. For a given step, we first count the frequencies of the options winning in pairwise mean performance comparisons. Then, the option with a bigger frequency will be ranked higher. If two options have the same win frequencies, then their median performances will be compared for ranking. All five performance indicators were used to rank the options separately, the average rank of the 5 independent ranks was used as the final rank (similar to the above workflow ranking).

## Ensemble inference

We proposed two ensemble inference strategies: (1) integration of no less than 2 of top 1st workflows using spectral counts, protein top0, top3, MaxLFQ, and directLFQ intensities for label-free DDA data (we name it ens_multi-quant), and (2) integration of top K workflows in our overall rankings (ens_topk). For label-free DIA data, ens_multi-quant integrates at least two of the top 1st workflows using top1, top3, MaxLFQ, and directLFQ intensities. For TMT data, ens_multi-quant is only applicable when FragPipe is used for quantification where TMT-Integrator abundance, TMT-Integrator ratio, and Philosopher intensity are available for integration. Usually, the log2 fold change (log2FC) and FDR adjust $p$-value ($q$-value) are two key statistics to infer whether a protein is differentially expressed; our ensemble inference should integrate sub-workflows' log2FCs and $p$-values into a log2FC and a $p$-value (then calculate a $q$-value from the integrated $p$-value) as ensembled statistics for the visited protein. For the integration of log2FC, we choose the log2FC having the biggest absolute value among all the sub-log2FCs as the ensembled log2FC. For p-values, five methods were used.

**Hurdle model.** The first one is Goeminne et al.'s hurdle model[44] where the $p$-values were transformed to $z$-values and combine them in a $\chi^2$ statistic:

$$\chi^2_{\text{hurdle}} = \sum_{i=1:K} z_i^2 \qquad (10)$$

where $\chi^2$ statistic follows a $\chi^2$ distribution with $t$ degrees of freedom if $t$ out of $K$ sub-$p$-values exist, and if $t = 1$, then the integrated $p$-value equals to the existing one. $Q$-values will be obtained by an FDR-adjustment with Benjamini–Hochberg's method.

**Fisher's method.** The second method is Fisher's combined probability test (Fisher's method)[82]. Fisher's method firstly combines $K$ sub-$p$-values as a $\chi^2$ statistic in the way of:

$$\chi^2_{2K} = -2 \sum_{i=1:K} \log_{p_i} \qquad (11)$$

where $p_i$ is the $i$th $p$-value, and $\chi^2$ has a chi-squared distribution with 2K degrees of freedom, then an integrated $p$-value can be determined. Similarly, Benjamini–Hochberg adjusted $p$-values are calculated.

**Voting methods.** The remaining three methods are based on voting strategies where we regard the $K$ statistical tests as $K$ voters. Anyone of the $K$ $p$-values pass the threshold e.g., $p$-value < 0.05, then we say the integrated test is significant, so we use minimum $p$-value (min $p$) of these $K$ $p$-values as the integrated $p$-value. Similarly, if we adopt the strictest condition where we request all $K$ $p$-values should pass, then maximum $p$-value (max $p$) is used as integrated $p$-value. The last one is called majority win, where if more $p$-values pass then the integrated $p$-value pass, thus the median of $K$ $p$-values (median $p$) is used. At last, a $q$-value is obtained by the Benjamini–Hochberg method.

The ens_multi-quant workflows and ens_topk workflows are also ranked in the same way as ranking single workflows according to the five performance indicators.

**Cross-setting and cross-instrument comparison**
Workflows analyzing proteomics expression data obtained from six quantification settings, e.g., FG_DDA, DIANN_DIA, etc., were benchmarked separately since the protein lists from different settings are always quite different as various peptide identification and protein inference are adopted. In addition, the spike-in DDA, DIA, and TMT datasets are always obtained from different proteomics projects and generated by different instruments, making the direct comparisons of DDA, DIA, and TMT quantification difficult.

Among the 24 datasets, there are 4 pairs of datasets from the same projects that used the same instrument but applied different quantification techniques, namely, the pair of HYEtims735_LFQ and HYEtims735_DIA, pair HYtims134_LFQ and HYtims134_DIA, pair HEqe408_LFQ and HEqe408_DIA, and pair HYqfl683_LFQ and HYqfl683_TMT11. The first 3 dataset pairs have been used to compare settings of FG_DDA, MQ_DDA, DIANN_DIA, and spt_DIA, while the last dataset pair is only suitable for comparing FG_DDA, MQ_DDA, with FG_TMT, and MQ_TMT. To solve the problem of protein list inconsistency under distinct settings, we merged all proteins from different settings and removed those protein groups with more than one protein. The top 1st workflows under each setting were chosen for a fair comparison, and the missing proteins from a specific setting compared to the merged protein list were padded with logFC = 0 and $q$-value = 1 (all regarded as non-differentially expressed). The top 1st ens_multi-quant was also compared. Thus, different top 1st workflows for various settings can be compared by counting the numbers of true positives (TPs), true negatives (TNs), false positives (FPs), and false negatives (FNs).

The first 4 datasets HYE5600735_LFQ, HYE6600735_LFQ, HYEqe735_LFQ, and HYEtims735LFQ, together with HYEtims735_DIA were generated by the proteomics project of PXD028735[31] where the same protein mixtures were digested under the same experimental condition and then submitted to different instruments. We thus also conducted the cross-instrument comparison of the top 1st single workflows and top 1st ens_multi-quant workflows of settings FG_DDA, MQ_DDA, DIANN_DIA, and spt_DIA in the way like the above cross-setting comparison.

**Reporting summary**
Further information on research design is available in the Nature Portfolio Reporting Summary linked to this article.

## Data availability
The raw proteomics data used in this work can be downloaded under the ProteomeXchange IDs or Proteomic Data Commons Study Identifiers listed in Table 1. The datasets HYE5600735_DDA, HYE6600735_DDA, HYEqe735_DDA, HYEtims735_DDA, and HYEtims735_DIA used in this study are available in the PRIDE database under accession code PXD028735. The datasets HYtims134_DDA and HYtims134_DIA used in this study are available in the PRIDE database under accession code PXD036134. The dataset HEtims425_DDA used in this study is available in the PRIDE database under accession code PXD021425. The dataset YUltq006_DDA used in this study is available in the Proteomic Data Commons database under accession code PDC000006 (https://proteomic.datacommons.cancer.gov/pdc/TechnologyAdvancementStudies/). The dataset YUltq099_DDA used in this study is available in the PRIDE database under accession code PXD002099. The dataset YUltq819_DDA used in this study is available in the PRIDE database under accession code PXD001819. The datasets HEqe408_DDA and HEqe408_DIA used in this study are available in the PRIDE database under accession code PXD018408. The datasets HYqfl683_DDA and HYqfl683_TMT11 used in this study are available in the PRIDE database under accession code PXD007683. The dataset HYEtims777_DDA used in this study is available in the PRIDE database under accession code PXD014777. The dataset MYtims709_DIA used in this study is available in the PRIDE database under accession code PXD034709. The datasets HEof_n600_DIA and HEof_w600_DIA used in this study are available in the PRIDE database under accession code PXD026600. The dataset HEqe777_DIA used in this study is available in the PRIDE database under accession code PXD019777. The dataset HEqe277_TMT10 used in this study is available in the PRIDE database under accession code PXD013277. The datasets HYms2-faims815_TMT16, HYsps2815_TMT16, and HYms2815_TMT16 used in this study are available in the PRIDE database under accession code PXD020815. More details of these datasets can be found in Table 1 in Supplementary Data 9. All the quantification results, extracted expression matrices, and our benchmarking results are available at our website: http://www.ai4pro.tech:3838 or through Zenodo at https://doi.org/10.5281/zenodo.10482353 for raw quantification results, https://doi.org/10.5281/zenodo.10953347 for extracted expression matrices and https://doi.org/10.5281/zenodo.10953480 for benchmarking results. Source data for all the figures are provided with this paper. Source data are provided with this paper.

## Code availability
The Python and R codes used for benchmarking and data analysis are available at https://github.com/PennHui2016/OpDEA/tree/main/codes_DEA_benchmarking. The R package and its source codes of our tool OpDEA are available at: https://github.com/PennHui2016/OpDEA (or through Zenodo at https://doi.org/10.5281/zenodo.10867031). The webserver is available at http://www.ai4pro.tech:3838. The standalone tool is available at http://www.ai4pro.tech:3838 or through Zenodo at https://doi.org/10.5281/zenodo.10958381.

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

## Acknowledgements

This research/project is supported by the National Research Foundation, Singapore, under its Industry Alignment Fund-Prepositioning (IAF-PP) Funding Initiative (W.W.B.G.). Any opinions, findings, conclusions, or recommendations expressed in this material are those of the author(s) and do not reflect the views of the National Research Foundation, Singapore. This work was partly supported by the National Innovation Fellow Program of the MOST of China (J.L., Grant No. E327130001). W.W.B.G. also acknowledges the support from an MOE Tier 1 award (RS08/21). J.L. acknowledges the support from his start-up funding grant at Shenzhen Institute of Advanced Technology, Chinese Academy of Sciences.

## Author contributions

H.P. conceived and designed the experiments. H.P. collected the data, conducted the benchmarking data analysis, and deployed the webserver. J.L. collaborated on frequent patterns and ensemble inference. H.P. and W.W.B.G. wrote the manuscript. H.W., W.K. and J.L. helped improve the manuscript. W.W.B.G. supervised the study. W.W.B.G. and J.L. acquired funding. All authors reviewed the manuscript and approved the manuscript.

## Competing interests

The authors declare no competing interests.
