## [Peer Review File · Nature Communications]

Optimizing Differential Expression Analysis for Proteomics Data via High-Performing Rules and Ensemble InferenceREVIEWER COMMENTS

Reviewer #1 (Remarks to the Author):

The manuscript by Peng et al. benchmarks major post-acquisition data processing steps and workflows for DEP analysis. This study analyzes a number of benchmark data acquired by DDA or DIA methods. Three data analysis platforms (FragPipe, MaxQuant and DIA-NN) are evaluated with these benchmark data. This work is expected to provide useful guidance in constructing proteomic DEP analysis workflows optimized for specific data types. Considering the proteomic workflows predominantly employed in current biological studies, the scope of workflow evaluation by this study can be expanded to offer more precise recommendations depending on the experimental design and available instruments.

This study is restricted to the analysis of Orbitrap data and label-free quantification. The reviewer would suggest analyzing data acquired on TOF instruments as well, especially timsTOF data, since this instrument has become very popular in proteomics experiments. Available benchmark data from timsTOF Prof include the HYE124 data [10.1038/s41597-022-01216-6], and another mouse-yeast hybrid data (Nat Commun.2023,14(1):94). The TOF spectra are typically less stable than Orbitrap, which may result in different characteristics of acquired MS data, especially DIA data, and thus affect the expression matrix with different patterns. In addition to label-free DDA and DIA data, it would be very informative to analyze some TMT labeling-based benchmark data given the high popularity of TMT quantification in large-scale proteomics studies.

For DIA data analysis, the widely used software Spectronaut which implements many data processing steps different from DIA-NN should be also considered for comparison.

In DEP analysis, differential proteins are routinely selected based on protein-level intensity matrix instead of peptide intensity data. But this study evaluates several intensity matrices at the peptide level which are rarely used for inferring DEPs. In fact, the authors can focus on protein-intensity matrices and compare typical methods for deriving protein intensities from precursor intensities such as Top3 precursors, software specific algorithms and MaxLFQ. Thus the best peptide intensity-based workflow is not meaningful and cannot be selected in the ensemble inference approach.

At individual processing steps, a few commonly used methods are missing and could be included in revision: TIC-based normalization and the MaxLFQ normalization method; quite a few methods for missing value imputation have been strictly evaluated for DIA data analysis [10.1093/nar/gkaa498], and this study can include some top-ranking methods reported previously.

It seems DIA data analysis by DIA-NN is not performed in a conventional and optimal manner. DIA-NN was configured in a high-speed mode, which is not recommended by the developer. Data re-analysis in the standard mode is suggested to ensure sensitivity and quantification accuracy. Also the report files by DIA-NN seem not correctly filtered. Matrix exported by DIA-NN, "report.pg_matrix.tsv", should be the same as "MaxLFQ_raw". And "MaxLFQ" will also be same as the previous two, if normalized quantities are used for MaxLFQ estimation.

Many critical details are absent in Methods, including but not limited to the following:

- "(using default parameters)" is a very general description. Critical parameters that need to be defined clearly include: digestion rules; restriction of peptide length and mass; modifications; mass tolerance; other specific software settings like min number of matched fragments in MSFragger.
- Before Percolator, is MSBooster enabled? Since FragPipe 17.1 already has MSBooster included, this is an important step which may have considerable impact on sensitivity.
- For IonQuant, what the FDR thresholds set for MBR?
- For both MaxQuant and IonQuant in FragPipe, the number of min ions for protein quantification estimation should be mentioned, since they are different in default.

Moreover, the methods tested in this study should be described in more details (In Methods or SI) for better understanding of their major principles and fundamental differences. For example, in the

first section of supplementary notes, the difference of two "quantiles" methods used for normalization is not clearly described. And descriptions mainly used in array data processing should be replaced with proteomics-related terms, such as "protein" or "peptide" instead of "probe", "MS run" or "injection" instead of "array". Further, in both MVI and DEA sections, each selected method should be briefly described. The leave-one-project-out cross validation is well explained in the text, yet 10-fold cross validation is not clearly defined.

Table 1 should specify the instrument type (which Orbitrap model), and benchmark sample type (species mixture or standard protein spike-in). LTQ Orbitrap data acquired from out-of-date instruments should be avoided due to much lower sensitivity and accuracy than more current models.

The author use 5 metrics for workflow evaluation. While pAUC is commonly used for DEP sensitivity/specificity assessment, what are the additional benefits of using two other metrics nMCC and G-mean? This can be clarified in the main text.

In regard to Figures 2 and 3, apart from using these statistical metrics for comparison of different workflows, it would be more straightforward and informative to list the exact numbers of DEPs given by top1 and top2 workflows based on a specific platform. In addition, numbers of true and false positives (TP and FP at a regular FDR of 0.01) can be listed so that one can clearly see the difference of performance between top1 and top2 workflows, or between two different methods at one key step.

From a user point of view, one would most like to see the differences in overall performances for specific workflows in processing DDA and DIA data, yet such results are not clearly presented. For DDA data, the distribution of pAUC(0.01) for top3 workflows based on Fragpipe vs MaxQuant platforms can be graphed so as to reveal which platform performs better with the same datasets. Likewise, for DIA data, Spectronaut (v17) which is a widely used tool, should be included in this benchmark study to be compared with DIA-NN in DEP analysis, and the same pAUC graphs can be prepared for top3 workflows based on two platforms.

In Figure 4, when comparing the top-ranking workflows with ensemble inferences, it seems the differences are more subtle if using G-mean for evaluation than using pAUC (0.01), highlighting the impact of choosing different metrics. So it would be also helpful to show the numbers of DEPs (including TP and FP) given by top-ranking workflows and ensemble inferences. Since the ensemble inference method is more time-consuming especially when processing large datasets, the recommended ens_TK (K=20, min p) method for DIA data analysis lacks feasibility as one would need to run data analysis with 20 different workflows. The time consumption for the ensemble inference method need to be discussed and proved valid (i.e. whether significantly more TPs and fewer FPs can be obtained than single workflows).

In the section of 'Guidelines for selecting optimal workflows', the authors can discuss more about whether their recommendations depend on the MS instrument type, sample type (total cell lysate vs IP samples) or the acquisition or quantification method (DDA, DIA, label-free vs isobaric labeling). Moreover, the main conclusion in Abstract that 'These rules include avoiding normalization, utilizing MinProb for missing value imputation' seems contradictory with general recommendations by DIA-NN and FragPipe which prefer normalization for complex biological datasets and no imputation applied. The major difference in guidelines from different studies should be mentioned and discussed.

A supplementary table can be provided to list the file names for different benchmark data sets to facilitate easy access.

Reviewer #2 (Remarks to the Author):

The article describes how the parameters of a label-free quantification and differential expression analysis workflow influence its performance.

The authors examined how data quantification normalization, imputation, and DEA influence performance measures such as ROC, pAUC, or nMMC, given benchmarking datasets where the ground truth is known.

The authors also included a Shiny application to examine the benchmarking results. They furthermore made the source code to run the benchmarks available on Git Hub.

Reviewer comments

Major

1

The authors provide the source code to run the benchmarks https://github.com/PennHui2016/OpDEA/tree/master/source_codes_for_benchmarking. Although this is very welcome, the aim must be to allow readers and reviewers to replicate the analysis. However, providing detailed instructions in README.md and making the input protein, peptide, or design_files available is necessary. The links to the data in the Rshiny application are all broken (access is denied), for instance: https://drive.google.com/drive/folders/1qv-P_0Jhpe1ZevSVL84-ORLkG4B_vcuB?usp=sharing
Please make the data available at a public repository, not on google drive.

2

MIV - the Author examines various imputation methods. Although the option "None" (no imputation) is listed in Figure 2, there are no results for the no imputation option. Please include the None imputation option in the benchmark.

3

The R packages proDA and msqrob2-hurdle model do not require imputation before modeling but can handle missing data directly. However, the Authors did use these packages with MVI only, which disagrees with their intended usage scenario.

4

The study's problem is that the dataset selection could be more balanced. All the datasets are spiked in datasets capturing technical variability only. Results should be interpreted accordingly. These biases must be sufficiently discussed in the Manuscript, which must be addressed in the Section "Limitations of current work". Furthermore, conclusions about normalization or the modeling methods should be adjusted to accommodate information about typical biological datasets where variances and systematic differences among samples are more significant.

5.

What the publication attempts is a sensitivity analysis of the DEA workflows. The studied parameters (explanatory variables) are imputation, normalization, etc. The pACU MCC are the observed response variables. The Authors write: "Our findings indicate that the selection of a good MVI algorithm, normalization method and DEA tool is more crucial for DEA performance compared to the choice of expression matrix type.". However, the authors need to explain how they arrived at these results.

A sensitivity analysis must summarize and quantify the variance of the responses to parameter changes and consistently consider the sample sizes.

The results obtained by the parameter scans can be better summarized using an ANOVA analysis or a linear model, where the parameters of the model (explanatory variables) are the dataset, imputation, or DEA model, and the response is the performance score.

ANOVA and Linear models would also permit the authors to reveal and asses interactions among the input variables.

6

The authors use leave-on-out cross-validation to predict workflow performance to out-of-bag datasets, which suggests that such a prediction is possible. Although technically, it can be implemented, since the selection of datasets used for benchmarking is necessarily highly biased, it is not viable to build a prediction model.

Therefore, remove the leave-one-out cross-validation and instead use linear models and ANOVA to make inferences about the obtained results.

7

The Figure captions need to be revised, and the figure description extended. Please provide reports of the y and x axis.

Specifically, Figure 3, caption

"Panel A: shows the pairwise comparison of expression matrix types of FragPipe and maxquant."
unclear

Furthermore, Panel B states, "B compares five expression matrix types from DIA-NN." But what is the mean-pairwise difference on the Y-axis?

Reviewer #3 (Remarks to the Author):

This is a very technical manuscript describing the process of evaluating over 10,000 various 'workflows' (combinations of tools and parameter settings) for three commonly used pipelines, FragPipe, MaxQuant, and DIA-NN. The authors have clearly put a lot of effort in this work. Unfortunately, this reviewer had a very hard time following the manuscript and understanding the significance of this work. The authors consider multiple different datasets, DIA and DDA data, tools, evaluation metrics etc. The plots are highly technical, the manuscript is full of terminology and numbers. The conclusions they arrive at are somewhat reasonable and expected, that is, that the results vary widely depending on various decisions made at each step in the data analysis process. However, after reading the manuscript, I do not have a clear 'take home' message.

First, I think the challenge here is that each biological study is different, and it is simply unrealistic to come up with universal guidelines based on the analysis of somewhat artificial benchmark datasets. For example, biological samples such as affinity purified protein complexes require completely different analysis strategies compared to say organellar protein profiling, and different again for large scale plasma protein profiling experiments. In certain cases, normalization of the data is absolutely necessary, in other cases it is not advisable. I do not think what we learn from spike-in benchmarking datasets can be easily generalized to most of the biological studies.

Second, most users of pipelines such as FragPipe, DIA-NN, or MaxQuant will have a hard time connecting this manuscript with how they use these pipelines in practice. This is so in part because these pipelines generate quantification matrices that are then used as input into other downstream tools. Most commonly, the users would use Perseus, or MSstats, or one of the several python or R-based pipelines (e.g., LFQ-Analysis) etc. The authors, however, are not evaluating various settings and parameters available in those popular downstream data analysis pipelines. So the reader is left wondering what they should actually do if they are using say MSstats or Perseus. I believe to be useful, the downstream evaluation performed in this work should closely link to what options are actually available in the commonly used downstream tools for proteomics data.

Third, a significant part of the manuscript is devoted to ensemble approaches, that is, application of multiple workflows followed by some integration of the results. Unfortunately, for an average user such a strategy is not practical as it requires significant bioinformatics expertise. Perhaps even more importantly, it complicates the analysis and thus makes it harder for people to describe it properly and get their papers through peer review. I would personally not advise anyone to employ more complicated, ensemble-based analysis strategies, especially given very modest improvement reported by the authors (just a few percent in most cases).

Author point-by-point response letter

Manuscript: NCOMMS-23-27297 (Optimizing Differential Expression Analysis for Proteomics Data via High-Performing Rules and Ensemble Inference)

January 30, 2024

We would like to thank the reviewers for their valuable comments. We have revised our manuscript accordingly. Our point-by-point responses are as follows.

Below, we have highlighted changes made in the main revised text (in red) in the main_R1_with_revision_highlighted.docx file. Please note the page numbers cited here correspond to the main_R1_clean.docx file without revision highlights. To help reviewers locate key information, we copy these changes here (colored in red).

Comments from Reviewer 1

The manuscript by Peng et al. benchmarks major post-acquisition data processing steps and workflows for DEP analysis. This study analyzes a number of benchmark data acquired by DDA or DIA methods. Three data analysis platforms (FragPipe, MaxQuant and DIA-NN) are evaluated with these benchmark data. This work is expected to provide useful guidance in constructing proteomic DEP analysis workflows optimized for specific data types. Considering the proteomic workflows predominantly employed in current biological studies, the scope of workflow evaluation by this study can be expanded to offer more precise recommendations depending on the experimental design and available instruments.

Reviewer-1-Comment-1: This study is restricted to the analysis of Orbitrap data and label-free quantification. The reviewer would suggest analyzing data acquired on TOF instruments as well, especially timsTOF data, since this instrument has become very popular in proteomics experiments. Available benchmark data from timsTOF Prof include the HYE124 data [10.1038/s41597-022-01216-6], and another mouse-yeast hybrid data (Nat Commun.2023,14(1):94). The TOF spectra are typically less stable than Orbitrap, which may result in different characteristics of acquired MS data, especially DIA data, and thus affect the expression matrix with different patterns. In addition to label-free DDA and DIA data, it would be very informative to analyze some TMT labeling-based benchmark data given the high popularity of TMT quantification in large-scale proteomics studies.

Response to Reviewer-1-Comment-1: We appreciate the reviewer's kind suggestion to compare workflows based on data acquired by timsTOF, and furthermore, are grateful for the advice to explore two related datasets (we used the HYE124 dataset suggested by the reviewer to generate datasets HYE5600735_LFQ, HYE6600735_LFQ, HYEeq735_LFQ, HYEtims735 and HYEtims735_DIA. We used

the mouse-yeast hybrid dataset (Nat Commun.2023,14(1):94) as suggested by the reviewer to generate the dataset MYtims709_DIA). In addition, we have added several new datasets generated by timsTOF under both DDA and DIA acquisition modes. We also used several publicly available TMT datasets for benchmarking workflows designed for analyzing TMT data. More detailed information about the datasets used for workflow benchmarking is described in **Table 1** (see below) and **supp9. Tab1** in our **Supplementary file 9**.

Table 1 Datasets used for workflow benchmarking.

Dataset	ID	Technique	Contrasts [#]	Instrument	Mixture	PMID [§]
HYE5600735_LFQ	PXD028735*	DDA	1	SCIEX Triple TOF5600	human+yeast+ecoli	35354825 ²⁷
HYE6600735_LFQ	PXD028735	DDA	1	SCIEX Triple TOF6600	human+yeast+ecoli	35354825 ²⁷
HYEqe735_LFQ	PXD028735	DDA	1	Orbitrap QE-HFX	human+yeast+ecoli	35354825 ²⁷
HYEtims735_LFQ	PXD028735	DDA	1	TimsToF pro	human+yeast+E.coli	35354825 ²⁷
HYtims134_LFQ	PXD036134	DDA	3	TimsToF pro	human+yeast	36541440 ²⁸
HEtims425_LFQ	PXD021425	DDA	3	TimsToF pro	human+E.coli	34373457 ²⁹
YUltq006_LFQ	PDC000006**	DDA	2	LTQ-Orbitrap	yeast+UPS1	19858499 ³⁰
YUltq099_LFQ	PXD002099	DDA	2	LTQ Orbitrap Velos	yeast+UPS1	26321463 ³¹
YUltq819_LFQ	PXD001819	DDA	3	LTQ Orbitrap Velos	yeast+UPS1	26862574 ³²
HEqe408_LFQ	PXD018408	DDA	1	Q Exactive	human+E.coli	33553868 ¹⁰
HYqfl683_LFQ	PXD007683	DDA	3	Orbitrap Fusion Lumos	human+yeast	29635916 ³³
HYEtims777_LFQ	PXD014777	DDA	1	TimsToF pro	human+yeast+E.coli	32156793 ¹⁷
HYEtims735_DIA	PXD028735	DIA	1	TimsToF pro	human+yeast+E.coli	35354825 ²⁷
MYtims709_DIA	PXD034709	DIA	3	TimsTOF Pro	mouse+yeast	36609502 ³⁴
HEof_n600_DIA	PXD026600	DIA	3	Orbitrap Fusion ETD	UPS1+E.coli	34472865 ³⁵
HEof_w600_DIA	PXD026600	DIA	3	Orbitrap Fusion ETD	UPS1+E.coli	34472865 ³⁵
HYtims134_DIA	PXD036134	DIA	3	TimsToF pro	human+yeast	36541440 ²⁸
HEqe777_DIA	PXD019777	DIA	3	Q Exactive HF	human+E.coli	34373457 ²⁹
HEqe408_DIA	PXD018408	DIA	1	Q Exactive	human+E.coli	33553868 ¹⁰
HEqe277_TMT10	PXD013277	TMT	3	Q Exactive	human+E.coli	32205417 ³⁶
HYqfl683_TMT11	PXD007683	TMT	3	Orbitrap Fusion Lumos	human+yeast	29635916 ³³
HYms2faims815_TMT16	PXD020815	TMT	3	Orbitrap Fusion Lumos	human+yeast	33175540 ³⁷
HYsps2815_TMT16	PXD020815	TMT	3	Orbitrap Fusion Lumos	human+yeast	33175540 ³⁷
HYms2815_TMT16	PXD020815	TMT	3	Orbitrap Fusion Lumos	human+yeast	33175540 ³⁷

* ProteomeXchange ID (<http://proteomecentral.proteomexchange.org/cgi/GetDataset>);

** [Proteomic Data Commons Study Identifier \(https://proteomic.datacommons.cancer.gov/pdc/\)](https://proteomic.datacommons.cancer.gov/pdc/); # number of contrasts used for benchmarking; § the PubMed unique identifier of the publication reporting the dataset (<https://pubmed.ncbi.nlm.nih.gov/>).

The description details about the datasets used for benchmarking have been revised accordingly:

Revised contents in **Abstract** at page 2 of main_R1_clean.docx:

To identify optimal workflows and their common properties, we conducted an extensive study involving 34,576 combinatoric experiments on 24 gold standard spike-in datasets covering four key quantification platforms (FragPipe, MaxQuant, DIA-NN, and Spectronaut).

Revised contents in **Introduction** at page 4 of main_R1_clean.docx:

To identify optimal workflows for a wide gamut of proteomics platforms (including label-free and labelled), we implemented a brute-force approach to test all possible combinations of (readily available) options/methods. The performance of each workflow combination is attributed to what we define as a “quantification setting”, i.e., pairings of a quantification platform and a data type, e.g., if an experiment involved quantification by Maxquant¹⁷ and is a labelled Tandem Mass Tags (TMT) experiment¹⁸, this setting will be encoded as “MQ_TMT”. Unlike the other steps which can be easily changed, the quantification settings reflect experimental conditions which are less flexible (these settings were used for generating the raw data already). Workflows are benchmarked on a large assembly of spike-in datasets inclusive of label-free DDA and DIA modes, or labelled TMT.

Revised contents in **Results** at page 5-6 of main_R1_clean.docx:

We amassed 12 DDA label free (DDA) datasets, 5 TMT datasets and 7 DIA label free (DIA) datasets from different proteomics projects for assessing DEA workflows (**Table 1**). This assemblage is, to the best of our knowledge, the most comprehensive and largest collection of benchmark data for workflow optimization across key proteomic platforms, and also has value for testing novel algorithms. We package these datasets under the OpDEA resource at <http://www.ai4pro.tech:3838>.

Table 1 presents the dataset names (Dataset), dataset depository identifier (ID), technique used for data generation (Technique), number of contrasts used for benchmarking (Contrasts), instrument type of the spectrometer (Instrument), the mixture type (Mixture) and the PubMed unique identifier of the publication reporting the dataset (PMID) for each dataset. A contrast refers to an expression level comparison of two groups of proteins acquired from two different samples. Thus, the contrast number in a dataset is equivalent to the number of unique sample pairs contained therein.

For a specific spike-in dataset such as YUItq006_LFQ³⁰, a total of 48 UPS1 (Universal Proteomics Standard 1) proteins spanning 5 concentrations were compared against the yeast background proteins. In this scenario, by selecting samples given 2 out of the 5 distinct concentrations, 10 possible pairwise contrasts are generated (i.e., $C_5^2 = 10$). Each of these selected contrasts is useful for the identification of differentially expressed proteins during analysis.

To prevent the performance benchmark from being denominated by any single dataset, we evaluated all contrasts in those datasets with many contrasts, and then selected only

a subset of these contrasts for comparison. E.g., only 2 out of the 10 contrasts in YUItq006_LFQ were used. Altogether, we have 22, 15 and 17 contrasts representing DDA, TMT and DIA platforms.

Due to compatibilities, we used the following quantification software for specific proteomics platforms:

- Fragpipe¹³ and Maxquant¹⁷ for the DDA and TMT data
- DIA-NN¹⁴ and Spectronaut³⁸ (spt) for the DIA data

For further details regarding workflows pertinent to each platform, please refer to **Methods**.

Revised contents in **Conclusion** at page 21 of main_R1_clean.docx:

We benchmarked 34,576 DEA workflows on 5 performance metrics across 24 datasets (12 DDA datasets, 7 DIA datasets and 5 TMT datasets).

Reviewer-1-Comment-2: For DIA data analysis, the widely used software Spectronaut which implements many data processing steps different from DIA-NN should be also considered for comparison.

Response to Reviewer-1-Comment-2: We thank the reviewer's suggestion for us to include software Spectronaut for comparison. We have now conducted tests on the workflows based on the quantification platform of Spectronaut (Spectronaut 18 was used). The label-free DIA data quantified by Spectronaut is termed as "spt_DIA" in our manuscript. The revised version of the description about the platforms are quoted as follows:

Revised contents in **Abstract** at page 2 of main_R1_clean.docx:

To identify optimal workflows and their common properties, we conducted an extensive study involving 34,576 combinatoric experiments on 24 gold standard spike-in datasets covering four key quantification platforms (FragPipe, MaxQuant, DIA-NN, and Spectronaut).

Revised contents in **Results** at page 6 of main_R1_clean.docx:

Due to compatibilities, we used the following quantification software for specific proteomics platforms:

- Fragpipe¹³ and Maxquant¹⁷ for the DDA and TMT data
- DIA-NN¹⁴ and Spectronaut³⁸ (spt) for the DIA data

In the subsection "**Quantitative analysis of mass spectrometry-based proteomics data**" of **Methods** at page 22 of main_R1_clean.docx, the descriptions of Spectronaut-based quantification are as follows:

Spectronaut 18³⁸ was used as an alternative platform for DIA data quantification. We

used the same sequence databases as DIA-NN used and predicted libraries by Spectronaut 18 (used the directDIA³⁸ workflow for quantification). Similar to DIA-NN, we extracted top1, top3, MaxLFQ intensities with the “iq” package and extracted directLFQ intensities with the directlfq tool from the report file of Spectronaut 18. The detailed parameters are described in the **Supplementary results** file. Similarly, the quantifications of DIA data with DIA-NN and Spectronaut were named quantification settings “DIANN_DIA” and “spt_DIA” for convenience.

We used the term “setting spt_DIA” to specify that Spectronaut was used to quantify the DIA data in most part of our manuscript, detailed modifications won’t be quoted here.

Reviewer-1-Comment-3: In DEP analysis, differential proteins are routinely selected based on protein-level intensity matrix instead of peptide intensity data. But this study evaluates several intensity matrices at the peptide level which are rarely used for inferring DEPs. In fact, the authors can focus on protein-intensity matrices and compare typical methods for deriving protein intensities from precursor intensities such as Top3 precursors, software specific algorithms and MaxLFQ. Thus the best peptide intensity-based workflow is not meaningful and cannot be selected in the ensemble inference approach.

Response to Reviewer-1-Comment-3: We thank the reviewer’s suggestions for us to remove peptide intensity-based analysis and to test more types of protein quantifications. We now only consider protein-level differential expression analysis. On label-free DDA datasets, we compared quantification methods including spectral counts, “top0” (quantification based on all available precursors) intensities, “top3” (quantification based on the 3 most intense precursors) intensities, the MaxLFQ intensities, and the directLFQ intensities (a recently published method, <https://doi.org/10.1016/j.mcpro.2023.100581>). For DIA label-free data, we used iq R package (<https://github.com/typham/iq>) to extract protein intensities with methods “top 1(maxInt, refers to quantification with only the most intense precursor)”, “top 3” and “MaxLFQ”. The directLFQ algorithm was also tested. For the TMT data quantified by platform FragPipe, the TMT-Integrator abundance, TMT-Integrator ratio, and Philosopher intensity were acquired. If Maxquant is used for TMT data quantification, we extracted the reporter intensity as the expression matrix. We have removed the workflows using peptide-intensity data as input.

For the ensemble inference, only protein-level quantification-based DEA was considered in this revised version. We removed previous ens_2inp and ens_3inp methods where peptide level intensity-related top workflows may be involved. Instead, for each type of protein-level expression matrix (5 for DDA, 4 for DIA and 3 for TMT quantified by FragPipe are available), their top 1st workflows can be candidates for ensemble inference (at least 2 of them), we call this method “ens_multi-quant”. In this

new version, ens_multi-quant always works better than ens_topk (previously called ens_TK) and can improve pAUC(0.01) by 1%~4% and improve the G-mean score by 6%~11% in mean performance comparing to single top 1st workflows under different quantification settings, e.g., label-free DDA data quantified by FragPipe (FG_DDA).

We quote the revision related to expression matrices used in our benchmarking as follows:

Revised contents discussing the expression matrices from FragPipe, at page 22 of main_R1_clean.docx:

In fact, four types of expression matrixes are obtainable from FragPipe including spectral counts, top0 intensities, top3 intensities and MaxLFQ intensities calculated by MaxLFQ algorithm⁵¹. In addition, the directLFQ intensities were extracted with the directLFQ algorithm¹⁹ (with the published directlfq tool: <https://github.com/MannLabs/directlfq>).

Revised contents discussing the expression matrices from Maxquant at page 22 of main_R1_clean.docx:

Again, spectral count, protein top0 and top3 (set the Top N peptides as 0 and check the Top3 option of “Advanced site intensities”) intensities, and MaxLFQ intensities can be acquired directly from Maxquant’s “proteinGroups.txt” file for DEA. The directLFQ intensities were also extracted with the directLFQ algorithm¹⁹. We named the quantification analysis of DDA data with FragPipe the quantification setting of “FG_DDA” and quantification analysis of DDA data with Maxquant the quantification setting of “MQ_DDA”.

Revised contents discussing the expression matrices from DIA-NN at page 22 of main_R1_clean.docx:

The top1 (most intense precursor-based quantification), top3 and MaxLFQ intensities were extracted by the “iq” R package⁷² (<https://github.com/tvpham/iq>) and the directLFQ intensities were extracted by the directLFQ algorithm from the report file of DIA-NN.

The previous version didn’t discuss software Spectronaut. The revised version described the extracted expression matrices from Spectronaut’s report as follows (page 22 of main_R1_clean.docx):

Similar to DIA-NN, we extracted top1, top3, MaxLFQ intensities with the “iq” package and extracted directLFQ intensities with the directlfq tool from the report file of Spectronaut 18. The detailed parameters are described in the **Supplementary results** file.

In the revised version, we also evaluated workflows for analyzing TMT data. The available expression matrices for TMT data quantification are quoted as follows (at page 22-23 of main_R1_clean.docx):

TMT data quantification. Both FragPipe (v20.0) and Maxquant (v2.4.4.0) were adopted for TMT quantification. For FragPipe, similar to the DDA data quantification, we enabled the MSBooster, and percolator and ProteinProphet were used for peptide identification and protein inference. The TMT-Integrator and the Philosopher were used for quantification with most of the parameters as default ⁴⁹. Detail descriptions of the parameters are available in our **Supplementary results**. We extracted the TMT-Integrator abundance and TMT-Integrator ratio from the TMT-Integrator reports (in the tmt-report folder, <https://github.com/Nesvilab/TMT-Integrator>), and the Philosopher intensities were extracted from the “protein.tsv” file of the specified result folder. For Maxquant, we selected the quantification type according to the quantification level specified in the publication of the dataset, e.g., the “Reporter ion MS2” with “10plex TMT” were selected for analyzing dataset HEqe277_TMT10. Most of the other parameters were set as default (see **Supplementary results** for detailed description). Only the “Reporter intensity corrected” values were extracted from the “proteinGroups.txt” file. We used “FG_TMT” and “MQ_TMT” to indicate the quantification analysis of TMT data with FragPipe and Maxquant respectively.

Revised contents at page 23 of main_R1_clean.docx:

For label-free DDA data quantified with FragPipe and Maxquant, five types of expression matrices are obtainable including spectral counts, protein top0 and top3 intensities, MaxLFQ intensities calculated by MaxLFQ algorithm ⁵¹, and the directLFQ intensities ¹⁹. As for DIA quantification with DIA-NN or Spectronaut, four types of protein intensities are obtainable including top1, top3, MaxLFQ (LFQ) and directLFQ (dlfq). For TMT data quantified by FragPipe, the TMT-Integrator abundance (abd), TMT-Integrator ratio (ratio) and Philosopher intensity (phi) are available, while for TMT data quantified by Maxquant, only the reporter intensity (reporter) was used (see above section and **Figure 1A, Figure 2E and supp. Fig. 2**).

Revised description of ensemble inference methods in **Methods** at pages 28 of main_R1_clean.docx:

We proposed two ensemble inference strategies: (1) integration of no less than 2 of top 1st workflows using spectral counts, protein top0, top3, MaxLFQ and directLFQ intensities for label-free DDA data (we name it ens_multi-quant), and (2) integration of top K workflows in our overall rankings (ens_topk). For label-free DIA data, ens_multi-quant integrates no less than 2 of the top 1st workflows using top1, top3, MaxLFQ and directLFQ intensities. For TMT data, ens_multi-quant is only applicable when FragPipe is used for quantification where TMT-Integrator abundance, TMT-Integrator ratio and Philosopher intensity are available for integration.

The results related to the ensemble inference were revised per the using of additional datasets and the removal of peptide expression matrix-based workflows. The revised results related to ensemble inference are described in “**Ensemble inference through integration of top-ranked workflows improves DEA coverage**” at page 14-15 of

Reviewer-1-Comment-4: At individual processing steps, a few commonly used methods are missing and could be included in revision: TIC-based normalization and the MaxLFQ normalization method; quite a few methods for missing value imputation have been strictly evaluated for DIA data analysis [10.1093/nar/gkaa498], and this study can include some top-ranking methods reported previously.

Response to Reviewer-1-Comment-4: We thank the reviewer's suggestion for us to compare more commonly used normalization and MVI methods. We added four normalization methods such as "TIC", "lossf", "Rlr" and "MBQN". The top ranked imputation algorithms in [10.1093/nar/gkaa498] including "Impseq", "Impseqrob", "GMS" and "SeqKNN" were also included in our benchmarking. MaxLFQ implemented their inner normalization. As the inner normalization is not easy to be integrated into our workflows, e.g., used as function or package that can be called in our codes, it's hard for us to compare it with other normalization methods directly. We considered the MaxLFQ as a quantification method only.

Revised version description of normalization and imputation methods (at page 23 of main_R1_clean.docx):

For normalization methods, we used those found in the MSnbase v2.22.0 R package⁷³ including "sum", "max", "center.mean", "center.median", "div.mean", "div.median", "quantiles", "quantiles.robust" and "vsn"(variance stabilization)¹⁵. **The two regression-based normalization methods "lossf"^{47,48} and "Rlr"⁵⁵, the total ion current normalization (TIC) and the Mean/Median-balanced quantile (MBQN)⁷⁴ normalization methods were also evaluated (we conclude them in **Figure 1D**). If no normalization is used, then we designate the normalization method as "none".**

For MVI methods, we also used those found in the MSnbase v2.22.0 R package such as "bpca" (Bayesian principal component analysis)⁷⁵, "knn" (k-nearest neighbors)⁵⁹, "QRILC" (quantile regression imputation of left-censored data)⁵⁷, "MLE" (maximum likelihood estimation)⁷⁶, "MinDet" (deterministic minimum)²³, "MinProb" (probabilistic minimum)²³, "min"⁷⁷, "zero"²³ and "nbavg" (neighbour averaging)⁷⁸, **and another six popular imputation methods such as mice (Multivariate imputation by chained equations)⁷⁹, Impseq (Sequential imputation)^{21,22}, Impseqrob (Robust sequential imputation)⁸⁰, GMS (Generalized Mass Spectrum Missing Peaks Abundance Imputation)⁴⁵, SeqKNN (Sequential KNN imputation)^{20,21}, missForest (random forest)¹⁶.** Again, if no imputation is performed, we designate the imputation method as "none".

According to the revised version results, no normalization is always preferred. The two regression-based normalization methods lossf, Rlr are superior to other methods (except "none") like quantile etc. The algorithms SeqKNN and Impseq also work well

in missing value imputation. In our pairwise workflow comparison-based normalization method comparison, the method lossf and Rlr were ranked at 2nd and 4th among the competing normalization method across the 6 quantification settings (**Figure 4C**). For the imputation algorithm comparison, SeqKNN and Impseq were ranked at 5th and 2nd among different imputation algorithms across the 6 quantification settings (**Figure 4D**). We have revised the subsection of **“General usefulness of options across different workflow settings”**. We quote the revised version of comparison results as follows:

Revised version at page 12-14 of main_R1_clean.docx:

Normalization: it is surprising to find the “no normalization” (or “none”) option works consistently well on every setting except MQ_DDA (**Figure 4C**). Does this suggest normalization methods are useless? We advise caution on this direct interpretation since the datasets used are artificial spike-in datasets. Moreover, certain quantification methods such as dlq¹⁹, LFQ⁵¹ and TMT-Integrator⁵² have built-in peptide-level or protein-level normalization steps. Thus, no additional normalization is usually required when these quantification methods have been used. The regression-based normalization method "lossf" and the simple approach "center.median" demonstrate superior performance compared to other methods. Thus, when normalization becomes necessary, especially in cases where substantial variances are observed among the samples within the same class, we recommend employing these methods.

MVI: there is less consistency here as the performance ranks of MVI algorithms vary widely across settings (**Figure 4D**). We find that missForest works well with FG_DDA (ranked 1st) and with MQ_DDA (ranked 3rd). However, missForest is also the most time-consuming algorithm compared to other top-ranked algorithms (the left heatmap in **Figure 4D** shows the running time, see **Methods**). MinProb works well with MQ_TMT (ranked 1st), with FG_DDA (ranked 2nd) and with FG_TMT (ranked 2nd). In addition, MinProb has the highest average ranking across the six different settings (averaging ranks based on the 6 settings shown in the last column of **Figure 4D**, see **Methods** for more details). This is not surprising, as MinProb addresses missing-not-at-random (MNAR) missingness, which plagues proteomics data⁵³. MinDet (ranked 1st for DIANN_DIA and 3rd for spt_DIA) and Impseq (ranked 3rd for DIANN_DIA and 1st for spt_DIA) are good candidates for imputing DIA data, and they have good overall ranks for various settings at the same time (average rankings are 3rd and 2nd respectively, see last column of **Figure 4D**). No imputation (“none” in **Figure 4D**) works the best with FG_TMT and is the 3rd top-ranked workflow with MQ_TMT though it works quite bad for the other 4 settings (ranked lower than 12th among the 16 algorithms). This may be due to the low missing rates of TMT data (average missing rate = 0.2%) compared to DIA data (average missing rate of 3%) and DDA data (average missing rate of 17%, see **Supplementary file 9**).

Reviewer-1-Comment-5: It seems DIA data analysis by DIA-NN is not performed in a conventional and optimal manner. DIA-NN was configured in a high-speed mode, which is not recommended by the developer. Data re-analysis in the standard mode is

suggested to ensure sensitivity and quantification accuracy. Also the report files by DIA-NN seem not correctly filtered. Matrix exported by DIA-NN, “report.pg_matrix.tsv”, should be the same as “MaxLFQ_raw”. And “MaxLFQ” will also be same as the previous two, if normalized quantities are used for MaxLFQ estimation.

Response to Reviewer-1-Comment-5: We thank the reviewer for pointing out the bias setting of our usage of the software DIA-NN. We rerun the quantifications based on the DIA-NN with standard mode (set “the Speed and RAM usage” as default option of “Optimal results”). The results have been updated. We then extracted DIA-NN expression matrices from its output “report.tsv” file with the R package “iq” (<https://github.com/tvpham/iq>) by methods of “top1 (maxInt)”, “top3”, “MaxLFQ”. The latest quantification methods “directLFQ” (<https://doi.org/10.1016/j.mcpro.2023.100581>) was also extracted as an additional expression matrix. The revised contents are quoted as follows:

Revised contents at page 22 of main_R1_clean.docx:

DIA data quantification. For DIA data, DIA-NN v1.8.1 was used for quantification under its default parameters (see details in **Supplementary results**) and libraries were predicted from corresponding databases¹⁴. Match between runs was checked and we ran DIA-NN under “Optimal results” mode. The top1 (most intense precursor-based quantification), top3 and MaxLFQ intensities were extracted by the “iq” R package⁷² (<https://github.com/tvpham/iq>) and the directLFQ intensities were extracted by the directLFQ algorithm from the report file of DIA-NN.

Reviewer-1-Comment-6: Many critical details are absent in Methods, including but not limited to the following:

- “(using default parameters)” is a very general description. Critical parameters that need to be defined clearly include: digestion rules; restriction of peptide length and mass; modifications; mass tolerance; other specific software settings like min number of matched fragments in MSFragger.
- Before Percolator, is MSBooster enabled? Since FragPipe 17.1 already has MSBooster included, this is an important step which may have considerable impact on sensitivity.
- For IonQuant, what the FDR thresholds set for MBR?
- For both MaxQuant and IonQuant in FragPipe, the number of min ions for protein quantification estimation should be mentioned, since they are different in default.

Response to Reviewer-1-Comment-6: We thank the reviewer’s suggestion of adding more detailed descriptions of our **Methods**. We have added more details about the settings of our quantification analysis with different software in our **Methods** section. We described more details about the parameters left as default in the **Supplementary Methods** part or **Supplementary result** (in the Supplementary result.docx file, page 25-60). Here, we quote the revised contents in the main manuscript as below:

Descriptions of parameter settings in revised version at page 21-23 of main_R1_clean.docx:

Quantitative analysis of mass spectrometry-based proteomics data

DDA data quantification. FragPipe v20.0 was used with its default quantification parameters. MSFragger-3.8¹³ was adopted for database search (using default parameters), in which contaminants and decoy (sequence reverse) protein sequences were added to corresponding libraries by Philosopher v5.0.0, e.g.,⁴⁹ human UPS1 proteins + reviewed UP000002311 *Saccharomyces cerevisiae* proteome from uniprot database⁶⁵ for YUltq099_LFQ, YUltq819_LFQ and YUltq006_LFQ datasets. More details about the databases used for database search in our benchmarking are listed in **supp9. Tab1 of Supplementary file 9**. MSbooster-1.1.11⁶⁶, Percolator v3.06⁶⁷ and ProteinProphet⁶⁸ were used for peptide identification and protein inference. For quantification, IonQuant-1.9.8⁶⁹ with match between runs (ion FDR of 0.01)⁷⁰, normalize intensity across runs and MaxLFQ (minimum ions of 2)⁵¹ were used. For label-free DDA data, we ran the quantification twice with the parameter of “Top N ions” set as 0 and 3 respectively to obtain the top0 (all precursors are considered) and top3 (only top3 most intense precursors are considered) quantification intensities. Details of the parameters are listed in **Supplementary results**. The quantification results are stored in files containing identified peptides or proteins with their spectral counts or intensities and other annotation information. We extracted expression levels of identified proteins in the form of spectral counts or intensities and organized them as a matrix by setting proteins as rows and samples as columns. The cells hold the expression levels. This data structure is referred to as an expression matrix, and it holds varied expression information. In fact, four types of expression matrixes are obtainable from FragPipe including spectral counts, top0 intensities, top3 intensities and MaxLFQ intensities calculated by MaxLFQ algorithm⁵¹. In addition, the directLFQ intensities were extracted with the directLFQ algorithm¹⁹ (with the published directlfq tool: <https://github.com/MannLabs/directlfq>).

MaxQuant v2.1.0.0 was used as an alternative quantification platform for label-free DDA data¹⁷. Andromeda⁷¹ was applied for database search. We kept most parameters default (see **Supplementary results** for more details) and used the same reference library as FragPipe. Similarly, match-between runs and MaxLFQ label-free quantification algorithm was used. Again, spectral count, protein top0 and top3 (set the Top N peptides as 0 and check the Top3 option of “Advanced site intensities”) intensities, and MaxLFQ intensities can be acquired directly from Maxquant’s “proteinGroups.txt” file for DEA. The directLFQ intensities were also extracted with the directLFQ algorithm¹⁹. We named the quantification analysis of DDA data with FragPipe the quantification setting of “FG_DDA” and quantification analysis of DDA data with Maxquant the quantification setting of “MQ_DDA”.

DIA data quantification. For DIA data, DIA-NN v1.8.1 was used for quantification under its default parameters (see details in **Supplementary results**) and libraries were predicted from corresponding databases¹⁴. Match between runs was checked and we ran DIA-NN under “Optimal results” mode. The top1 (most intense precursor-based quantification), top3 and MaxLFQ intensities were extracted by the “iq” R package⁷² (<https://github.com/tvpham/iq>) and the directLFQ intensities were extracted by the directLFQ algorithm from the report file of DIA-NN.

Spectronaut 18³⁸ was used as an alternative platform for DIA data quantification. We used the same sequence databases as DIA-NN used and predicted libraries by Spectronaut 18 (used the directDIA³⁸ workflow for quantification). Similar to DIA-NN, we extracted top1, top3, MaxLFQ intensities with the “iq” package and extracted directLFQ intensities with the directlfq tool from the report file of Spectronaut 18. The detailed parameters are described in the **Supplementary results** file. Similarly, the quantifications of DIA data with DIA-NN and Spectronaut were named quantification settings “DIANN_DIA” and “spt_DIA” for convenience.

TMT data quantification. Both FragPipe (v20.0) and Maxquant (v2.4.4.0) were adopted for TMT quantification. For FragPipe, similar to the DDA data quantification, we enabled the MSBooster, and percolator and ProteinProphet were used for peptide identification and protein inference. The TMT-Integrator and the Philosopher were used for quantification with most of the parameters as default⁴⁹. Detail descriptions of the parameters are available in our **Supplementary results**. We extracted the TMT-Integrator abundance and TMT-Integrator ratio from the TMT-Integrator reports (in the tmt-report folder, <https://github.com/Nesvilab/TMT-Integrator>), and the Philosopher intensities were extracted from the “protein.tsv” file of the specified result folder. For Maxquant, we selected the quantification type according to the quantification level specified in the publication of the dataset, e.g., the “Reporter ion MS2” with “10plex TMT” were selected for analyzing dataset HEqe277_TMT10. Most of the other parameters are set as default (see **Supplementary results** for detailed description). Only the “Reporter intensity corrected” values were extracted from the “proteinGroups.txt” file. We used “FG_TMT” and “MQ_TMT” to indicate the quantification analysis of TMT data with FragPipe and Maxquant respectively.

Reviewer-1-Comment-7: Moreover, the methods tested in this study should be described in more details (In Methods or SI) for better understanding of their major principles and fundamental differences. For example, in the first section of supplementary notes, the difference of two “quantiles” methods used for normalization is not clearly described. And descriptions mainly used in array data processing should be replaced with proteomics-related terms, such as “protein” or “peptide” instead of “probe”, “MS run” or “injection” instead of “array”. Further, in both MVI and DEA sections, each selected method should be briefly described. The leave-one-project-out cross validation is well explained in the text, yet 10-fold cross validation is not clearly

defined.

Response to Reviewer-1-Comment-7: We thank the reviewer’s suggestion of introducing the methods compared in our workflows in more details. We have updated our “Supplementary result.docx” file to include more descriptions of methods for normalization, imputation, and the DEA statistical tools (page 17-21 in the “Supplementary result.docx” file). We now added the detailed description of our 10-fold cross-validation for checking performance level prediction accuracy. We quote the descriptions as follows:

Revised in section “**CatBoost classification and the linear model**” at page 26 of main_R1_clean.docx (the description of 10-fold cross-validation has been highlighted in bold and with underline):

CatBoost classification and the linear model

The CatBoost is a new gradient boosting toolkit supporting solving categorical features-based machine learning tasks⁴³. To confirm that the performance level of a workflow is predictable with the information of its option in each step, 10-fold cross validation was applied to validate the classification accuracy of a CatBoost classifier (with python package of CatBoost-1.2.2) in classifying workflows. We first labeled the workflows as “H”, “RH”, “RL” and “L” if their ranking positions are located at the top 5%, 5%~25%, 25%~50% and 50%~100% respectively. Then, the workflows are encoded as feature vectors with their options in each step as categorical features. **At last, the workflows were split into 10-folds randomly and in each of the 10 rounds, 9 out of the 10 folds data were used to train the CatBoost classifier and predict the performance levels of the workflows in the remaining fold.** The hyperparameters were mostly left as default e.g., “iterations” of 1000, “depth” of 6, “l2_leaf_reg”, etc., while the “learning rate” was set as 0.3 for speeding up the training. The metrics MCC and F1-score were used as performance indicators and the average performance of the 10-round was recorded. The feature importance was evaluated by fitting the CatBoost classifier with all the workflows and their labels. The F1-score is calculated by:

$$Precision = \frac{TP}{TP+FP} \quad (7)$$

$$F1 = \frac{2*Recall*Precision}{Recall+Precision} \quad (8)$$

Reviewer-1-Comment-8: Table 1 should specify the instrument type (which Orbitrap model), and benchmark sample type (species mixture or standard protein spike-in). LTQ Orbitrap data acquired from out-of-date instruments should be avoided due to much lower sensitivity and accuracy than more current models.

Response to Reviewer-1-Comment-8: We thank the reviewer’s suggestion of adding instrument type information and removing the data generated by older instruments. We have updated **Table 1** to include instrument and sample information. For LTQ Orbitrap data, we didn’t remove all of them but select part of them for our benchmarking. We conducted the Kruskal-Wallis test to check whether there exists significant difference in median rankings of workflows based on datasets generated by different instruments, i.e., LTQ, and the newer ones, e.g., TimsTOF pro. Our results show that for DDA data (LTQ was only used for generating DDA data), when rank the workflows by each single dataset, the rankings of those top ranked workflows (through our benchmarking, we tested top30 workflows for both FragPipe and Maxquant-based quantifications) nearly have no significant median difference between instrument groups generating the datasets (only the ranked 28th FG_DDA workflow obtained $p\text{-value} < 0.05$, means significant difference exists, see our **Figure 2D**). Thus, keeping these datasets generated by older instruments won’t affect the optimal workflow recommendation. We quote the description of the Kruskal-Wallis test and the results as follows:

Revised in subsection of “**Kruskal-Wallis test**” at page 26 of main_R1_clean.docx:

Kruskal-Wallis test

The biggest contrast number of the 24 gold standard datasets is 3, thus it is impossible for us to check whether the benchmarking is sensitive to the dataset used for performance evaluation with statistical methods (the group size is no bigger than 3). The DDA, DIA and TMT data were generated by different types of instruments (see **Table 1**), e.g., newer ones such as Tims TOF or older ones such as LTQ Orbitrap Velos. For the 20 contrasts (SCIEX Triple TOF5600 and SCIEX Triple TOF6600 were excluded) of the 12 DDA datasets, we group them into 3 categories including Tims TOF pro (N=8), LTQ-Orbitrap/Velos (N=7) and other Orbitrap machines (N=5). Each of the 20 contrasts were used to rank the workflows based on the 5 indicators (above benchmarking method). Then, we conducted the Kruskal-Wallis test⁴² to check whether a workflow’s rankings based on the three groups of contrasts have the same median at threshold of $p\text{-value} \leq 0.05$. Similarly, for the 17 contrasts of the DIA data, 2 instrument groups are available such as the Tims TOF pro (N=7) and the other Orbitrap machines (N=10). For the TMT data, we cannot split them into groups with the minimum size requirement of $N \geq 5$, so no Kruskal-Wallis test was conducted.

The Kruskal-Wallis test results are discussed in subsection of “**Optimal workflows are predictable and are settings-specific**” at page 7 of main_R1_clean.docx:

The mean ranks may be unstable if performance metrics are sensitive to the choice of instrument. To assay this, we used the Kruskal-Wallis test⁴² (see **Methods**). We plot 30 top performing workflows ranked by mean performance under different settings against the log transformed p values of the Kruskal-Wallis tests (**Figure 2D**). We conclude that under most circumstances, the rankings of top workflows are not sensitive to instrument type (shown as gray markers in **Figure 2D**). However, some workflows involving

DIANN_DDA do show some sensitivity to instrument type, with 4 out of 30 workflows reporting p-values < 0.05 , indicating significant ranking position differences between instrument types (**Supplementary file 2**).

Reviewer-1-Comment-9: The author use 5 metrics for workflow evaluation. While pAUC is commonly used for DEP sensitivity/specificity assessment, what are the additional benefits of using two other metrics nMCC and G-mean? This can be clarified in the main text.

Response to Reviewer-1-Comment-9: We thank the reviewer's question. We now added the description of why using 5 metrics instead of only the commonly used pAUC in our **Methods** part at page 25 of main_R1_clean.docx. The main reason is that the pAUC is a rank-based metric which is not sensitive to the exact values of the data. In an extreme example, all the proteins have small fold changes, and their p-values are bigger than 0.05 while the true differentially expressed proteins are still ranked better than those non-differentially expressed proteins. In this case, a good pAUC score can still be obtained though all true positives will be wrongly labelled with the traditional threshold, e.g., $FC \geq 1.5$ and $q\text{-value} \leq 0.05$. In addition, pAUC is an overall performance indicator by ranging the FPR threshold. We cannot interpret the performance at a specific threshold, e.g., $FC \geq 1.5$ and $q\text{-value} \leq 0.05$. However, in a traditional differential expression analysis task, we are more interested in the detected differentially expressed proteins at the specific threshold but not a range of threshold, e.g., $FPR < 0.01$. As we used spike-in datasets for benchmarking where the true labels are known, thus we can use the popular classification performance indicators, e.g., normalized MCC (nMCC) and G-mean for fair performance comparisons at interested thresholds. We expect to use the two groups of metrics (pAUC-based and the confusion matrix-based) for a more accurate performance evaluation. We quote the revised contents in our manuscript below.

In the subsection of “**Performance evaluation metrics**” of the revised version manuscript at page 25 of main_R1_clean.docx:

The metric pAUC is a rank-based global performance indicator where its value is calculated by specifying a FPR range e.g., ≤ 0.01 . However, in a DEA task, we may always be interested in the reliability of detected differentially expressed proteins at a given logFC and q-value threshold but not a FPR range, which cannot be deciphered from the pAUC score. Thus, we combined the confusion matrix-based metrics nMCC and G-mean with the pAUC scores to conduct a more comprehensive performance evaluation.

Reviewer-1-Comment-10: In regard to Figures 2 and 3, apart from using these statistical metrics for comparison of different workflows, it would be more straightforward and informative to list the exact numbers of DEPs given by top1 and

top2 workflows based on a specific platform. In addition, numbers of true and false positives (TP and FP at a regular FDR of 0.01) can be listed so that one can clearly see the difference of performance between top1 and top2 workflows, or between two different methods at one key step.

Response to Reviewer-1-Comment-10: We thank the reviewer's suggestion for us to provide detailed TP and FP numbers for a clearer comparison of top-ranked workflows. As multiple datasets were used for benchmarking, it's impossible for us to plot all the TP and FP numbers for every dataset. Thus, in **Figure 2E** and **supp. Fig. 2 of Supplementary result** (page 3), we plot the mean pAUC(0.01) vs. mean G-mean score of the top 2 workflows for each specific expression matrix type, e.g., top1, top3, MaxLFQ, directLFQ, etc. used. In addition, in the cross-setting and cross-instrument comparisons (**Figure 5B-D**), we plot the detail TP, FP numbers detected by the top 1st single workflows and the best ensemble inference workflows.

We quote subfigures showing TP, FP numbers and the related descriptions as follows:

Description of plots showing top ranked workflows in subsection of “**Optimal workflows are predictable and are settings-specific**” at page 9 of main_R1_clean.docx:

We observed that top-ranked workflows for settings FG_DDA, MQ_DDA, FG_TMT, MQ_TMT, DIANN_DIA and spt_DIA can substantially change their rankings at the four remaining workflow steps (i.e., setting-wise rankings). **Figure 2E** presents the top 2 workflows for each matrix type under settings FG_DDA, MQ_DDA, DIANN_DIA and spt_DIA by their mean pAUC(0.01) and G-mean scores (The top-ranked workflows under FG_TMT and MQ_TMT are shown in **supp. Fig. 2 of Supplementary results**, see detailed descriptions of these top-ranked workflows in supp. Tab. 1 of **Supplementary results**). The pAUC(0.01) and the G-mean are used as the representative AUC-based metric and confusion matrix-based metric for the convenience of visualization, the other 3 types of metrics can be found in **supp. Tab. 1 of Supplementary results**. The label and its corresponding color denote the DEA tool, normalization method, MVI algorithm and expression matrix type used by the workflow. The overall workflow rank position is shown in brackets.

Below is the subfigure **Figure 2E** and the corresponding caption:

Figure 2E shows the top 2 workflows for each matrix type under FG_DDA, MQ_DDA, DIANN_DIA and spt_DIA settings. The color encodes the matrix type, and the labels encode selection details on DEA, normalization and MVI. The overall ranks are shown in brackets.

Description of plots showing TP, FP numbers detected by top ranked single workflows and top ranked ens_multi-quantif workflows from the PXD028735 project and PXD036134 project data in subsection of “Comparisons within quantification settings and within instrument types for differential expression analysis” at page 15-16 of main_R1_clean.docx:

Comparisons within quantification settings and within instrument types for differential expression analysis

Since we now have a means of predicting optimal pipelines across a variety of settings, we may ask when optimized to maximize performance, if some settings are superior to others when it comes to DEA. We compared workflows under labelled (FG_TMT and MQ_TMT) and label-free settings, e.g., label-free DDA (FG_DDA and MQ_DDA) and label-free DIA (DIANN_DIA and spt_DIA).

We chose datasets which are compatible with the evaluated settings (see **Table 1**). The dataset HYEtimes735_LFQ and HYEtimes735_DIA (generated with the same samples and instrument but acquired under DDA and DIA mode respectively, see **Table 1** and **Supplementary file 9**) were used to compare settings FG_DDA, MQ_DDA, DIANN_DIA and spt_DIA. To facilitate comparisons across settings where differential proteins are reportable, we pooled a reference protein list by merging proteins detected under different settings. Then, for each best workflow specific to a setting (prefixed top1), we calculated coverage of this reference protein list (see **Methods**).

From the top 4 bars of **Figure 5B**, we can see, the two DIA settings DIANN_DIA and spt_DIA work better than the DDA settings FG_DDA and MQ_DDA, with more true

positives (TPs) and less FPs. For example, DIANN_DIA detected 1271 more TPs (3734-2463) than FG_DDA but found less than half the number of FPs (54 vs 144). Compared against MQ_DDA, the TP number reported by DIANN_DIA is nearly tripled (3734 vs. 937), and the FP number is halved (54 vs 134). Compared against DIANN_DIA, spt_DIA detected 680 less TPs (3734-3054) and 12 less FPs (54-42). Based on another two pairs of datasets (HEqe408_LFQ, HEqe408_DIA) and (HYtims134_LFQ, HYtims134_DIA), we can observe similar performance that DIA works better than DDA, and FG_DDA works better than MQ_DDA. In addition, DIANN_DIA works better than spt_DIA in 2 out of the 3 pair-comparisons (see **Supplementary results** and **Supplementary file 10**).

Only the dataset pair (HYqfl683_LFQ, HYqfl683_TMT11) is applicable to compare with DDA settings and TMT settings. As seen from the top 4 bars of **Figure 5C**, the TP numbers found by the TMT settings (1435 TPs for FG_TMT and 1015 TPs for MQ_TMT) are nearly doubled that of each DDA setting (685 TPs for FG_DDA and 519 TPs for MQ_DDA). Again, much less FPs were found by the TMT settings than the DDA settings (19 and 13 for FG_TMT and MQ_TMT vs 144 and 134 for FG_DDA and MQ_DDA).

Below are the subfigures **Figure 5B** and **Figure 5C** showing the TP, FP numbers detected by top ranked workflows (shown as setting names, e.g., spt_DIA means the top 1st workflow under setting spt_DIA) and top rank 1st ens_multi-quant workflows (shown as setting name plus the suffix “_ens”). The detail labels of the bar plots can be found in **Supplementary file 10**.

Figure 5B

Figure 5C

Figure 5B compares the best workflows of DDA and DIA settings and investigates the ensemble inference’s improvement in DEP detection. The label of the bar gives the setting type, e.g., FG_DDA, or whether ens_multi-quant is used or not. Similarly, **Figure 5C** compares the DDA and TMT and ensemble inference-enabled DDA and TMT best workflows.

Reviewer-1-Comment-11: From a user point of view, one would most like to see the differences in overall performances for specific workflows in processing DDA and DIA data, yet such results are not clearly presented. For DDA data, the distribution of pAUC(0.01) for top3 workflows based on Fragpipe vs MaxQuant platforms can be graphed so as to reveal which platform performs better with the same datasets. Likewise, for DIA data, Spectronaut (v17) which is a widely used tool, should be included in this benchmark study to be compared with DIA-NN in DEP analysis, and the same pAUC graphs can be prepared for top3 workflows based on two platforms.

Response to Reviewer-1-Comment-11: We thank the reviewer’s suggestion for us to add cross-platform comparisons. We added the cross-setting (**Figure 5B** and **Figure 5C**, comparing top ranked workflows with datasets quantified by different settings, here the setting refers to the combination of a quantification platform and a data type) and cross-instrument comparison (shown in **Figure 5D** comparing top ranked workflows with dataset generated by different instruments) section where we used the bar plots to show TP, FP detected with top ranked single and ens_multi-quant workflows. The bar plots showing cross-setting comparisons have been quoted in above “**Response to Reviewer-1-Comment-10**” (together with **Figure 5B** and **Figure 5C**). The bar plot showing the detected TP, FP numbers of the cross-instrument comparisons is shown in below quoted **Figure 5D** (labels of the bar plot can be found in **Supplementary file 11**):

Figure 5D

We quote the discussion of the cross-instrument comparison result in subsection of “Comparisons within quantification settings and within instrument types for differential expression analysis” at page 16 of main_R1_clean.docx as follows:

Given the multiple choices for the instruments, e.g., SCIEX Triple TOF machines, Orbitrap QE-HFX and TimsTOF pro, we ask which one can maximize DEA performance. We adopted the 5 datasets from the same project (ID PXD028735, see **Table 1**) to conduct such a cross-instrument-cross-setting comparisons (settings are indicated by the prefixes, and instruments are indicated by the suffixes of the labels for the top bars in **Figure 5D**). The TimsTOF pro (suffix as `_tims`) worked the best, discovering more proteins (higher proteome coverage) and more TPs no matter under DDA or DIA acquisition modes. SCIEX Triple TOF5600 (suffix as `_st5600`) detected smaller numbers of TPs under both `FG_DDA` (588) and `MQ_DDA` (600) quantification settings, which account for only about 20% of the numbers of TimsTOF pro with DIA and 25% of the numbers of TimsTOF pro with `FG_DDA`. However, this machine also detected less FPs (16 with `MQ_DDA` and 24 with `FG_DDA`). SCIEX Triple TOF6600 (suffix as `_st6600`) worked better than SCIEX Triple TOF5600 with about 200 more TPs and also more FPs (21 increased by `MQ_DDA` and 53 by `FG_DDA`). Orbitrap QE-HFX (suffix as `_qe`) doubled the sizes of detected TP numbers compared to SCIEX Triple TOF6600 and kept the same or increased a little in FP numbers. It even detected more TPs than TimsTOF pro under the quantification setting `MQ_DDA` (1625 vs 927). In conclusion, if possible, TimsTOF pro is recommended for proteomics data analysis, otherwise Orbitrap QE-HFX is suggested.

We also plot the ROC curves for the above cross-setting comparisons in **supp. Fig. 7** of our **Supplementary result** (page 13) showing the changes of the true positive rates under different false positive rates. Below are the two groups of ROC curves corresponding to above **Figure 5B** and **Figure 5C**:

(a)

(b)

supp. Fig. 7 The Receiver Operating Characteristic (ROC) curves showing the performances of the best single workflows and best ens_multi-quant (for MQ_TMT, the ens_topk was tested instead) workflows under different settings. (a) shows the ROC curves of settings FG_DDA, MQ_DDA, DIANN_DIA and spt_DIA. The partial area under ROC curves with $FPR \leq 0.05$ (pAUC(0.05)s) are shown in the legend of the figure. (b) shows the ROC curves of settings FG_DDA, MQ_DDA, FG_TMT and MQ_TMT. The pAUC(0.05) scores can be found from the legend.

The worse performances of top ranked single and ensemble inference workflows of label-free DDA settings, e.g., MQ_DDA and FG_DDA, may be due to that lower proteome coverages were achieved compared to the DIA and TMT quantification where their missed true positives were padded as non-differentially expressed proteins.

Reviewer-1-Comment-12: In Figure 4, when comparing the top-ranking workflows with ensemble inferences, it seems the differences are more subtle if using G-mean for evaluation than using pAUC (0.01), highlighting the impact of choosing different metrics. So it would be also helpful to show the numbers of DEPs (including TP and FP) given by top-ranking workflows and ensemble inferences. Since the ensemble inference method is more time-consuming especially when processing large datasets, the recommended ens_TK (K=20, min p) method for DIA data analysis lacks feasibility as one would need to run data analysis with 20 different workflows. The time consumption for the ensemble inference method need to be discussed and proved valid (i.e. whether significantly more TPs and fewer FPs can be obtained than single workflows).

Response to Reviewer-1-Comment-12: We thank the reviewer's suggestion for us to present the TP, FP numbers of ensemble inference results as well. Again, in our **Figure 5B-D** (see above quoted figures), we present the detailed TP, FP numbers detected by ensemble inference workflows compared to the top 1st single workflows (see above two point-to-point Responses). In our newer version, the ens_multi-quant method works better than the ens_topk method. The top ranked ens_multi-quant workflows always integrate no more than 3 workflows thus the time cost won't be quite high. On the other hand, DEA is a downstream analysis process. Thus, the time cost is not that important if a slight time increase can result in more accurate detection of DEPs. In **Figure 4C-4E**, we added the running time of each method used in the DEA workflows. In our **supp. Tab. 5** (page 14 of our **Supplementary results**, see below quoted table), we also showed the required running time of our ensemble inference. We discuss the results of TP, FP numbers comparisons related to ensemble inference as below:

In subsection of “**Comparisons within quantification settings and within instrument types for differential expression analysis**” at page 16-17 of main_R1_clean.docx as follows:

We are also interested in whether the use of ensemble inference can narrow the performance gaps caused by different settings or instruments. The bottom bars in **Figure 5B-D** display the TP and FP numbers achieved by the best ensemble inference workflows (suffix as _ens) and these numbers are compared with the performances of the single best workflows (see **Methods**). We can see that the ensemble inference recovered more TPs than the single workflows except for the TMT data analysis (FG_TMT_ens and MQ_TMT_ens only find 1 and 15 more TPs than FG_TMT and MQ_TMT respectively, see **Figure 5C**). For example, 150 (3884-3734) and 175 (3229-3054) more TPs were reported by DIANN_DIA_ens and spt_DIA_ens respectively

than by DIANN_DIA and spt_DIA (**Figure 5B**). Impressively, the TP number gap due to using Orbitrap QE-HFX and TimsTOF pro with setting MQ_DDA can be completely closed, where MQ_DDA_ens reached the TP number from 927 (MQ_DDA) to 1606 (**Figure 5B**), which is almost the same as MQ_DDA_qe and FG_DDA_qe's 1625 and 1682 (Orbitrap QE-HFX, **Figure 5C**). Similarly, ensemble inference has let Orbitrap QE-HFX (FG_DDA_qe_ens) to catch up with the performance of TimsTOF pro (FG_DDA_tims) in detecting TPs (2376 vs. 2463, **Figure 5D**). In addition, FG_DDA_ens (3282) detected more TPs than spt_DIA (3054) and spt_DIA_ens (3229). However, ensemble inference increases the false positive rate at the same time. For instance, although FG_DDA_ens found 819 (3282-2463) more TPs than FG_DDA, 1218 (1362-144) more FPs were also recommended (**Figure 5B**). FG_DDA_st6600_ens found 372 (1214-842) more TPs than FG_DDA_st6600 with a sacrifice of detecting 923 (1000-77) more FPs (**Figure 5C**). We can see more successes in closing performance gap (setting-specific) albeit with smaller false positive rate increases when leveraging on ensemble inference, e.g., MQ_DDA_ens recovered 125 more TPs but with only 46 more FPs comparing to MQ_DDA testing on dataset HEqe408_LFQ (see **Supplementary results**). These evidences suggest that integrating different quantification method is a promising approach for improving DEA performance.

supp. Tab. 5 Top-ranked workflow running time comparison

Setting	Method	Dataset	Workflows involved	Running time
DIANN_DIA	TOP1	HEqe408_DIA	1	27.94s
DIANN_DIA	ens_multi-quant	HEqe408_DIA	3	207.03s
spt_DIA	TOP1	HEqe408_DIA	1	103.4s
spt_DIA	ens_multi-quant	HEqe408_DIA	3	254.48s

Reviewer-1-Comment-13: In the section of ‘Guidelines for selecting optimal workflows’, the authors can discuss more about whether their recommendations depend on the MS instrument type, sample type (total cell lysate vs IP samples) or the acquisition or quantification method (DDA, DIA, label-free vs isobaric labeling). Moreover, the main conclusion in Abstract that ‘These rules include avoiding normalization, utilizing MinProb for missing value imputation’ seems contradictory with general recommendations by DIA-NN and FragPipe which prefer normalization for complex biological datasets and no imputation applied. The major difference in guidelines from different studies should be mentioned and discussed.

Response to Reviewer-1-Comment-13: We thank the reviewer’s suggestion for us to have more discussions about the optimal workflow recommendations. We have included more detailed discussions about the optimal workflow recommendation in our **Supplementary results** (page 14, section **Optimal workflow recommendations**). Our recommendations depend on the setting type, i.e., the proteomics acquisition or quantification method such as label-free DDA data quantified by a specified platform,

e.g., FragPipe. We have proved that the ranking of the top-ranked workflows is not sensitive to instrument types, so we didn't make instrument type related recommendations. And because our benchmarking datasets are from artificial spike-in data, we don't have enough sample type information to support our sample type related recommendations.

We quote the discussion about our recommendations as follows:

In subsection of “**Optimal workflow recommendations**” at page 14 of Supplementary results.docx:

In the first subsection (“**Recommendations and resources for selecting optimal workflows**”) of the **Discussion** section in our main text, we listed some recommendations of optimal workflows for each setting. Our recommendation is mainly based on the benchmarking results as we found that our benchmarking has good generalizability (LODOCV with average Spearman correlation coefficients exceed 0.58, **Figure 2C**) and the optimality is predictable where higher than 0.84 of average F1 score and average MCC scores were achieved in the workflow performance level classification. These predictions are stable, exhibiting low sensitivity to instrument types (only 6 out of 120 top ranked workflows show instrument sensitivities (Kruskal-Wallis test²⁸, **Figure 2D**). We also consider the frequent patterns extracted from high-performing workflows where the inclusive of these frequent patterns possibly will lead to better performance (**Figure 3D**). In addition, the workflow step option comparison results (**Figure 4B-E**) were also used as clues. We explain our recommendation as below:

We have revised our conclusion in **Abstract** and the general recommendations in **Discussion** according to our new results obtained from new benchmarking data.

We quote the revision as follows:

Our revised conclusion in **Abstract** at page 2 of main_R1_clean.docx:

Some high-performing rules include using directLFQ for quantification, avoiding normalization or applying “lossf”, and deploying SeqKNN, Impseq, or MinProb for MVI. Finally, for differential expression analysis, ROTS and limma are good choices.

Our conclusion of the preference of no normalization and with imputation is based on the evaluations of workflows (including DIA-NN related and FragPipe related workflows) with various spike-in datasets but not complex biological datasets. No normalization is preferred may be due to two main reasons. Firstly, the spike-in datasets used in this work mainly simulate technological variance but not biological variance, where small data variances exist comparing to complex biological datasets. Secondly, the quantification methods directLFQ and MaxLFQ for label-free data or the TMT-

Integrator for TMT data have already implemented inner peptide level or protein level normalization. Thus, for the spike-in dataset, data variances in the expression matrices extracted from these quantification results are further reduced. In our **Discussion**, we suggest checking the data variance in advance to determine whether normalization is required.

We discussed the selection of normalization in our **Results** and **Discussion** sections, and are quoted as follows:

In subsection “**General usefulness of options across different workflow settings**” at page 12-13 of main_R1_clean.docx:

Normalization: it is surprising to find the “no normalization” (or “none”) option works consistently well on every setting except MQ_DDA (**Figure 4C**). Does this suggest normalization methods are useless? We advise caution on this direct interpretation since the datasets used are artificial spike-in datasets. Moreover, certain quantification methods such as dlq¹⁹, LFQ⁵¹ and TMT-Integrator⁵² have built-in peptide-level or protein-level normalization steps. Thus, no additional normalization is usually required when these quantification methods have been used. The regression-based normalization method "lossf" and the simple approach "center.median" demonstrate superior performance compared to other methods. Thus, when normalization becomes necessary, especially in cases where substantial variances are observed among the samples within the same class, we recommend employing these methods.

In subsection “**The ranked workflows provide a means for identification of algorithms associated with high performance**” at page 19 of main_R1_clean.docx:

- Normalization performed surprisingly badly and may even worsen the DEA performance. Some advanced quantification methods (directLFQ and MaxLFQ) have built-in peptide-level or protein level normalizations, thus usually no additional normalization is required. We recommend checking the distributions of the expression matrix in advance to determine whether normalization is required (and not just do it routinely, especially when it is not needed). If needed, regression-based normalization methods, e.g., lossf and Rlr, and center.median or center.mean are good options.

In subsection “**Spike-in data**” at page 20 of main_R1_clean.docx:

Spike-in data

We used spike-in data to benchmark workflows. Spike-in datasets mainly simulate technical variation by producing technique replicates for each sample. In real-world proteomics data, biological variation always exists and may manifestly quite differently from technical variance. The inability to simulate biological variation exactly may introduce some degree of bias to our benchmarking results. However, there is currently no universally accepted way of simulating biological replicates. Attempting to do so at

this time may invite scepticism. There are some promising new approaches based on deep generative models⁶⁰ for in silico simulation of biological variation, we will look at this closely for inclusion in our next round of benchmarking efforts.

For large complex biological datasets especially the DIA datasets with many biological and technical replicates and quantified by advanced platforms e.g., DIA-NN and Spectronaut, the missing rate is quite small. In addition, to avoid bias introduced by missing value imputation algorithms, it's safer to delete those replicates with higher missing rates or ignore the missing values during the statistical analysis than to apply a MVI algorithm. However, in this benchmarking work, the data sizes are small and the missing rates of DDA data is high (average missing rate is about 17% see **Supplementary file 9**), deletion of high missing rate replicates or ignoring missing values will result in lower proteome coverages and the loss of true positives. Thus, MVI is preferred in our conclusion.

We discussed the selection of MVI algorithm in our **Discussion** section and are quoted as follows:

In subsection “**The ranked workflows provide a means for identification of algorithms associated with high performance**” at page 19 of main_R1_clean.docx:

- Simple MVI algorithms such as MinProb and MinDet work well with DIA and TMT data. This is probably because most missingness in proteomics data are attributable to missing-not-at-random^{53,56}. However, for bigger projects with many biological and technical replicates, removing high-missing-rate replicates or ignoring the missing values during the statistical analysis may be safer for avoiding bias introduced by MVI algorithms^{57,58}.

Reviewer-1-Comment-14: A supplementary table can be provided to list the file names for different benchmark data sets to facilitate easy access.

Response to Reviewer-1-Comment-14: We thank the reviewer's suggestion for us to provide information of file names for facilitating data accessibility. We provide the **Supplementary file 9** to list more detailed information about our benchmarking data including the download link of the expression matrices and the detailed metric values from our benchmarking results, etc.

Comments from Reviewer 2

The article describes how the parameters of a label-free quantification and differential expression analysis workflow influence its performance.

The authors examined how data quantification normalization, imputation, and DEA influence performance measures such as ROC, pAUC, or nMMC, given benchmarking

datasets where the ground truth is known.

The authors also included a Shiny application to examine the benchmarking results. They furthermore made the source code to run the benchmarks available on Git Hub.

Reviewer-2-Comment-1:

The authors provide the source code to run the benchmarks https://github.com/PennHui2016/OpDEA/tree/master/source_codes_for_benchmarking. Although this is very welcome, the aim must be to allow readers and reviewers to replicate the analysis. However, providing detailed instructions in README.md and making the input protein, peptide, or design_files available is necessary. The links to the data in the Rshiny application are all broken (access is denied), for instance: https://drive.google.com/drive/folders/1qv-P_0Jhpe1ZevSVL84-ORLkG4B_vcuB?usp=sharing

Please make the data available at a public repository, not on google drive.

Response to Reviewer-2-Comment-1: We thank the reviewer's suggestion about the improvement of our data and tool accessibility. We have updated our github site (<https://github.com/PennHui2016/OpDEA/>) to give more details about the reproducibility of our work, the usage of our tools etc. We now updated our website (<http://www.ai4pro.tech:3838>) to provide data download links.

Reviewer-2-Comment-2:

MIV - the Author examines various imputation methods. Although the option "None" (no imputation) is listed in Figure 2, there are no results for the no imputation option. Please include the None imputation option in the benchmark.

Response to Reviewer-2-Comment-2: We thank the reviewer's comment about the comparison of workflows without imputation. In this revision manuscript, we discussed the results of workflows without MVI. In our **Figure 4D**, the method "none" refers to no imputation. It always works quite bad for label-free data analysis. However, for TMT data it works well as lower missing rates exist (the average missing rates of label-free DDA data, label-free DIA data and TMT data are 17%, 3% and 0.2% respectively, see our **Supplementary file 9**). We quote the discussion about the workflows applying no imputation as follows:

In subsection of "**General usefulness of options across different workflow settings**" at page 14 of main_R1_clean.docx:

No imputation ("none" in **Figure 4D**) works the best with FG_TMT and is the 3rd top-ranked workflow with MQ_TMT though it works quite bad for the other 4 settings (ranked lower than 12th among the 16 algorithms). This may be due to the low missing rates of TMT data (average missing rate = 0.2%) compared to DIA data (average missing rate of 3%) and DDA data (average missing rate of 17%, see **Supplementary**

file 9).

Reviewer-2-Comment-3:

The R packages proDA and msqrob2-hurdle model do not require imputation before modeling but can handle missing data directly. However, the Authors did use these packages with MVI only, which disagrees with their intended usage scenario.

Response to Reviewer-2-Comment-3: We thank the reviewer's comments about the usage of DEA tool proDA and msqrob2. Though these two tools can work without missing value imputation, we'd like to check whether imputation can still improve their performances. We have removed the peptide intensity-based workflows including those workflows using the msqrob2 as the DEA tool per the suggestion of reviewer 1. Those proDA related workflows with MVI of "none" are exactly testing the no-MVI usage scenario of it. We use the proDA without MVI as baseline workflows for checking MVI's effects on proDA related workflows' performances. Below figure shows the performance distributions of workflows using proDA and with or without MVI under setting of using FragPipe to quantify DDA data:

In above figure, we show the distributions of performances (measured by metrics of pAUC(0.01) etc., shown in x-axis) of workflows using proDA for DEA and applying different imputation algorithms (shown in y axis). The red points show the mean performances, and the black vertical lines show the baseline mean performance, i.e., proDA without imputation ("none"). We can see, even though proDA can work without imputation (shown as "none" in above figure), using some imputation methods, e.g., missForest, MinProb and MinDet can still improve the performance, e.g., get higher mean pAUC(0.05), pAUC(0.1), nMCC and G-mean scores.

Reviewer-2-Comment-4:

The study's problem is that the dataset selection could be more balanced. All the datasets are spiked in datasets capturing technical variability only. Results should be interpreted accordingly. These biases must be sufficiently discussed in the Manuscript, which must be addressed in the Section "Limitations of current work". Furthermore, conclusions about normalization or the modeling methods should be adjusted to accommodate information about typical biological datasets where variances and systematic differences among samples are more significant.

Response to Reviewer-2-Comment-4: We thank the reviewer's suggestion for us to add discussions about the bias related to the spike-in data used for benchmarking. We have added the discussion about the dataset bias in our **Discussion** section. It's true that the spike-in datasets are mostly simulating the technique variance. However, simulation of biological variance is quite hard and currently we don't have such type of gold-standard data for workflow evaluation. Our conclusion that no normalization works better may be due to two reasons. One is the spike-in dataset are mainly simulating technique variance where the data are more stable with less variance. Secondly, the advanced quantification methods such as directLFQ and MaxLFQ for label-free data and the TMT-integrator for TMT data have built-in peptide-level or protein-level normalization. We also suggested that for real-life data, it's better for us to check the data variance in advance, e.g., using histograms to help determine whether data normalization is necessary. If so, we suggest using the regression-based normalization method such as lossf and Rlr or the center.median, as these methods work better through our benchmarking. We quote the discussion contents as follows:

In subsection "**Ranked workflows provide a means for identification of algorithms associated with high performance**" at page 19 of main_R1_clean.docx:

- Normalization performed surprisingly badly and may even worsen the DEA performance. Some advanced quantification methods (directLFQ and MaxLFQ) have built-in peptide-level or protein level normalizations, thus usually no additional normalization is required. We recommend checking the distributions of the expression matrix in advance to determine whether normalization is required (and not just do it routinely, especially when it is not needed). If needed, regression-based normalization methods, e.g., lossf and Rlr, and center.median or center.mean are good options.

The additional discussion about the limitation of no biological variance simulation in subsection of "**Limitations**" at page 20 of main_R1_clean.docx:

Spike-in data

We used spike-in data to benchmark workflows. Spike-in datasets mainly simulate technical variation by producing technique replicates for each sample. In real-world

proteomics data, biological variation always exists and may manifestly quite differently from technical variance. The inability to simulate biological variation exactly may introduce some degree of bias to our benchmarking results. However, there is currently no universally accepted way of simulating biological replicates. Attempting to do so at this time may invite scepticism. There are some promising new approaches based on deep generative models⁶⁰ for in silico simulation of biological variation, we will look at this closely for inclusion in our next round of benchmarking efforts.

Fidelity and applicability to the real world

Although we try to test as many datasets and methods as possible, it does not mean the predictions and recommendations built on these will work on any data. We cannot guarantee it. Real world data is complex and covers many situations (scenarios or sources or variation) that cannot exist in our relatively simpler simulations. Still, we think that the availability of OpDEA as a recommendation resource provides a useful starting point. Researchers will no longer need to simply copy a workflow from some other publication, with no knowledge of whether the workflow really works well for their own data. In this regard, using OpDEA can positively enhance proteomics research.

Reviewer-2-Comment-5:

What the publication attempts is a sensitivity analysis of the DEA workflows. The studied parameters (explanatory variables) are imputation, normalization, etc. The pACU MCC are the observed response variables. The Authors write: "Our findings indicate that the selection of a good MVI algorithm, normalization method and DEA tool is more crucial for DEA performance compared to the choice of expression matrix type.". However, the authors need to explain how they arrived at these results.

A sensitivity analysis must summarize and quantify the variance of the responses to parameter changes and consistently consider the sample sizes.

The results obtained by the parameter scans can be better summarized using an ANOVA analysis or a linear model, where the parameters of the model (explanatory variables) are the dataset, imputation, or DEA model, and the response is the performance score. ANOVA and Linear models would also permit the authors to reveal and assess interactions among the input variables.

Response to Reviewer-2-Comment-5: We thank the reviewer's suggestion for us to use ANOVA and linear models to assess the interactions among different variables and workflow performance.

Our previous conclusion about the importance of variables to the response namely the workflow performance is based on the variable importance evaluated in the classification of workflow's performance levels, e.g., top 5% ("H"), top 5% to top 25% ("RH"), top 25% to top 50% ("RL") and remaining ("R"). The variables e.g., matrix type and normalization methods, were encoded as categorical features. The CatBoost classifier was used for workflow performance level classification. The classification performance was evaluated by a 10-fold cross-validation test, where we split the

workflows into 10 folds randomly and in each of the 10 rounds, we use 9 out of the 10 folds to training the classifier and predict the performance level of the remaining fold. Quite good average classification performance was achieved (MCC and F1 are higher than 0.84, **Figure 3A**). We at last analyzed the variable importance in the classification, which can reflect the importance of each variable in determining the performance level of a workflow. In this revised version, our results show that the variables DEA tool and the normalization method show higher importance in determining the performance levels (**Figure 3B**) of DDA and TMT workflows. For DIA data, the matrix type also got the high importance similar to DEA tool and the normalization method. We quote our classification-based variable importance evaluation method description and results as follows:

We describe our method of variable importance evaluation in subsection “**CatBoost classification and the linear model**” at page 26 main_R1_clean.docx:

CatBoost classification and the linear model

The CatBoost is a new gradient boosting toolkit supporting solving categorical features-based machine learning tasks⁴³. To confirm that the performance level of a workflow is predictable with the information of its option in each step, 10-fold cross validation was applied to validate the classification accuracy of a CatBoost classifier (with python package of CatBoost-1.2.2) in classifying workflows. We first labeled the workflows as “H”, “RH”, “RL” and “L” if their ranking positions are located at the top 5%, 5%~25%, 25%~50% and 50%~100% respectively. Then, the workflows are encoded as feature vectors with their options in each step as categorical features. At last, the workflows were split into 10-folds randomly and in each of the 10 rounds, 9 out of the 10 folds data were used to train the CatBoost classifier and predict the performance levels of the workflows in the remaining fold. The hyperparameters were mostly left as default e.g., “iterations” of 1000, “depth” of 6, “l2_leaf_reg”, etc., while the “learning rate” was set as 0.3 for speeding up the training. The metrics MCC and F1-score were used as performance indicators and the average performance of the 10-round was recorded. The feature importance was evaluated by fitting the CatBoost classifier with all the workflows and their labels. The F1-score is calculated by:

$$Precision = \frac{TP}{TP+FP} \quad (7)$$

$$F1 = \frac{2*Recall*Precision}{Recall+Precision}$$

(8)

The result of variable importance evaluation in subsection “**Frequent patterns extracted from high-performing workflows**” at page 9 of main_R1_clean.docx:

To discover decision rules enriched in top-ranked workflows, we used machine learning.

Under each setting, we encoded every workflow as a feature vector where every option in a step is considered as a categorical feature value. We assigned each workflow a performance level such as high (“H”), relatively high (“RH”), relatively low (“RL”) or low (“L”), if its rank falls within top 5%, between top 5% and top 25%, between top 25% and 50%, and in the remaining 50% respectively (see **Supplementary file 3**). We used these workflow feature vectors and their labels to train a CatBoost classifier⁴³ followed by 10-fold cross validation (we randomly split the workflows into 10 folds and each fold is evaluated against a trained classifier from the other 9 folds of workflows, see **Methods**) for performance evaluation (**Figure 3A**).

Workflow performance levels are predictable with average F1 and nMCC scores above 0.84. To understand which features are important for good prediction performance, **we examined model feature importance and found that model performance depends more on the choice of normalization and DEA tool than on expression matrix type and MVI algorithms for label-free DDA and TMT data (Figure 3B; see Supplementary file 3 for more details). For label-free DIA data, normalization and DEA are again important but the matrix type also appears to be important.**

Per the suggestion, we fitted linear models to check the response of workflow performance to parameter change, i.e., the options of a workflow step. As we use 5 different performance metrics to evaluate the workflow performance, we didn't use the exact metric values as response value, the ranking score of the workflow is used instead. The ranking score is calculated by $N - \text{rank positive}$, where N means the total number of workflows, rank position of the workflow is determined by the average ranks of a workflow based on mean metric values of the 5 metrics (see our **Methods**). As shown in **Figure 3C**, most DEA tools show positive affection of performance. Some of the normalization and imputation methods also show positive affections while most of the normalization methods affect the ranking score negatively. In **supp. Tab. 2** of the **Supplementary result**, we extracted the ANOVA table from the linear model for checking the variances among variable values. The results show that significant differences exist among different variable values for all four variables where $p\text{-value} < 0.05$ were obtained under label-free data related four settings. For the settings of FG_TMT and MQ_TMT, the results show that there is no significant difference between imputation methods to the response values. The F values show ratio of the between-group variance to the within-group variance of the response values, which can be used to indicate the importance of variable values to the change of the response value. We can see similar trends of the F-values and our feature importance values of the four variables obtained by above classification.

We quote the linear model method and related results as follows:

Description of the linear model in subsection “**CatBoost classification and the linear model**” at page 26-27 of main_R1_clean.docx:

We also fitted linear models to check the interactions and synergies of predictor variables, i.e., the options in each step (with dummy coding where the categorical variables, e.g., the normalization, are recoded into a set of separate binary variables), and the response variable, i.e., the ranking score of a workflow. The ranking score is calculated by:

$$\text{ranking score} = N - \text{ranking position} \quad (9)$$

where N means the number of workflows considered, e.g., $N=7852$ for workflows accepting DDA data analyzed by FragPipe. *Ranking position* means the number indicating the order of the performance of a workflow among all the workflows. Ranking position of 1 means the best and smaller the better. The “Estimate” values (also known as coefficients) are used to indicate the average increase in the response variable associated with a one unit increase in the predictor variable, assuming all other predictor variables are held constant. Bigger “Estimate” absolute value means higher impact of the predictor variable on response variable. Positive “Estimate” value means positive impact otherwise negative impact (see **Supplementary results**). The $p\text{-value} < 0.05$ indicates the predictor variable is significant to the response variable. The interactions between categorical variables are discussed in **Supplementary results**.

We also extracted the classic ANOVA (analysis of variance) ²⁴ table from the above linear model to check the affections of categorical variables (i.e., expression matrix type, normalization, imputation and DEA tool) on the response variable. The F-value (the ratio of the variation between sample means to the variation within the samples) calculated from the F-test ⁸⁹ was used as the indicator of the impact of category variable changes on the response variable. The $p\text{-value} < 0.05$ indicates the sample means are significantly different (see **Supplementary results**).

The analysis results of the linear model in subsection of “Frequent patterns extracted from high-performing workflows” at page 9-10 main_R1_clean.docx:

As independent measures, feature importance (of each feature) is not always linked directly to workflow performance. Interactions between features can result in synergies or conflicts depending on their compatibility. We fitted linear models to investigate the interactions between features and workflow ranks. In our linear model, the options in each step of a workflow are set as predictor features while the performance ranks of the workflows are set as response features (see **Methods**). **Figure 3C** displays the log transformed p-value against the log transformed estimated increase in ranking scores (calculated by N minus ranking position, N means the total number of workflows) for FG_DDA workflows. The choices of DEA tools such as limma, DEP ⁴⁴ etc., and concomitant selection of MVI algorithms such as SeqKNN, missForest ¹⁶, no-normalization or center.median normalization etc., can improve workflow ranks. In contrast, most normalization methods coupled with MVI algorithms such as GMS ⁴⁵ and no-imputation negatively impact rankings (linear model fitting results for other

settings are discussed in **Supplementary results** and are listed in **Supplementary file 4**).

To deep dive into the extent of the rank shifts, we extracted the ANOVA²⁴ tables from the linear models of the six settings to check the ranking score differentiations induced by changing the workflow step options (see **Methods**). Under setting FG_DDA, we observed that the statistical testing based on the mean differences of workflow groups formed from the options in each workflow step are significant (p-value < 0.05). The F values (calculated as the ratio of the between-group variance to the within-group variance, which is used to indicate the impacts of option changes on workflow rankings) are consistent with the feature importance scores obtained earlier (see **supp. Tab. 1**). We observed similar conclusions in other settings (**supp. Fig. 4** and **supp. Tab. 2 in Supplementary results**).

We quote the **supp. Tab. 2** (page 8 of **Supplementary results**) as follows:

supp. Tab. 2 The comparison of CatBoost-based and the ANOVA-based variable importance evaluations

		DEA	Imputation	normalization	Matrix
FG_DDA	F value	754.1432	368.73213	679.0099	90.72208
	Pr(>F)	0	0	0	1.65E-75
	importance	30.91	21.88	33.7	13.51
MQ_DDA	F value	873.5715	407.36016	690.8741	90.97371
	Pr(>F)	0	0	0	1.02E-75
	importance	34.1	21.3	32.3	12.3
FG_TMT	F value	261.4199	1.030828	217.0357	23.30576
	Pr(>F)	0	0.4193933	0	3.36E-23
	importance	28.46	9.49	38.31	23.74
MQ_TMT	F value	321.8335	1.2653863	165.1316	NA
	Pr(>F)	2.1E-296	0.2111009	1.5E-294	NA
	importance	40.99	13.8	45.22	NA
DIANN_DIA	F value	707.4383	238.53104	311.671	887.5088
	Pr(>F)	0	0	0	0
	importance	25.68	20.35	28.62	25.36
spt_DIA	F value	850.1747	48.953274	390.2877	1339.511
	Pr(>F)	0	7.5E-147	0	0
	importance	26.57	14.1	29.95	29.37

Reviewer-2-Comment-6:

The authors use leave-on-out cross-validation to predict workflow performance to out-of-bag datasets, which suggests that such a prediction is possible. Although technically, it can be implemented, since the selection of datasets used for benchmarking is necessarily highly biased, it is not viable to build a prediction model.

Therefore, remove the leave-one-out cross-validation and instead use linear models and

ANOVA to make inferences about the obtained results.

Response to Reviewer-2-Comment-6: We thank the reviewer's suggestion for us to use ANOVA or linear model instead of LOOCV for checking dataset bias in benchmarking. In our leave-one-dataset-out cross-validation (LODOCV, in previous version, we called it the leave-one-project-out cross-validation, namely LOPOCV), we tried to investigate whether the ranks of workflows benchmarked by gold standard datasets can be generalized to the ranks of the same workflows benchmarked by a newcoming dataset. So, we are in fact checking the ranking consistency of the same workflows evaluated by different datasets. Here, we use the rank positions as response value, e.g., workflow A is ranked at top 1st (position 1) and workflow B is ranked at position 2, but not the exact performance metric values. If high consistency is achieved, it's safe to recommend optimal workflows, i.e., the workflows with higher rank positions, for analyzing the new dataset to assure better performance. We used the Spearman correlation to evaluate the consistency, where higher correlation means better consistency. We think the LODOCV can help support such kind of consistency and can prove the safety of using benchmarking results to suggest better workflows for analyzing a new dataset. In **Figure 2B**, we show an example round of the LODOCV, where the dataset HYEtimes735_LFQ (see **Table 1**) was regarded as a newcoming dataset. We first rank the available label-free DDA workflows (N=7852) based on their performances evaluated with HYEtimes735_LFQ. Then, we ranked the same workflows by their mean performances evaluated on remaining gold standard label-free DDA datasets in Table1 (11 DDA datasets). Finally, we check the ranking position consistency by calculating the Spearman correlation between the two groups of ranking positions. In our **Figure 2C**, we show that high Spearman correlations were achieved (mean value is about 0.6 or higher for different quantification settings, see **Figure 2C**) in LODOCVs under different quantification settings, which can prove the good generalizability of our benchmarking results. So, we keep this part instead of removing it totally.

We quote the method of our leave-one-dataset-out cross-validation, the **Figure 2B**, **Figure 2C** and related discussions as follows:

We described the leave-one-dataset-out cross-validation in subsection of "**Leave-one-dataset-out cross-validation**" at page 25 of main_R1_clean.docx:

To confirm whether our benchmarking results can be used for recommending optimal workflows for newcoming datasets, we conduct the leave-one-dataset-out cross-validation (LODOCV). Taking the label-free DDA data as an example, there are 12 datasets for benchmarking, see **Table 1** for more details. In a LODOCV, each time, we use 11 out of 12 datasets to rank workflows (benchmarking), the contrasts in the remaining dataset are regarded as newcoming datasets. We calculated the Spearman correlation coefficient between the workflow ranks based on benchmarking with the 11-datasets and the true workflow ranks of the newcoming data. The higher the

correlation is, the more accurate recommendations could be made with our benchmarking.

The **Figure 2B**, **Figure 2C** and related discussions are in subsection “**Optimal workflows are predictable and are settings-specific**” at page 7 of main_R1_clean.docx:

To test whether our benchmarking results can support workflow recommendations for new datasets, we designed a leave-one-dataset-out cross-validation (LODOCV) procedure to take advantage of our assemblage of datasets in **Table 1**. LODOCV is a form of multiple validation, where in each test round, one dataset is reserved for performance testing while the remaining datasets are used for model training (see **Methods**). Although LODOCV takes the form of a typical cross-validation procedure, there is an important distinction: In cross-validation, a dataset of single origin is split into multiple components for the purpose of model tuning. In LODOCV, the datasets are of multiple origins. In this regard, the LODOCV procedure is akin to performing several rounds of independent validation and is more robust than typical procedures where only one independent validation is performed for a trained model. To evaluate consistency, we use the Spearman correlation coefficient (R) to compare the workflow ranks obtained from the training dataset against the validation dataset.

Figure 2B illustrates results from one round of LODOCV. We plot the ranks of workflows obtained from the validation dataset HYEtimes735_LFQ²⁷ (under the setting FG_DDA, **Table 1**) against the corresponding ranks from the remaining DDA datasets (training datasets). In this comparison, a mean R of 0.71 was achieved, suggesting conservation of information. To summarize across all our datasets, we present the overall LODOCV results tested under different settings in **Figure 2C**.

The workflow ranks on TMT are quite stable where we obtained ~0.8 mean R regardless of whether FragPipe or Maxquant (FG_TMT or MQ_TMT) was used for quantification. A mean R ~ 0.66 was achieved on DEA workflows for FG_DDA and MQ_DDA (label-free DDA data quantified by Maxquant). For workflows based on DIANN_DIA and spt_DIA (label-free DIA data quantified by DIA-NN and Spectronaut respectively), a mean R of ~0.57 was obtained. We report the means as it was slightly superior to the median. For corresponding results based on the median, please refer to **supp. Fig. 1 in Supplementary results**).

Figure 2B

Figure 2C

Figure 2B presents an example to demonstrate the process of the leave-one-dataset-out cross-validation, where the x-axis shows the averaged ranks of FG_DDA workflows (across five metrics) obtained from dataset HYEtimes735_LFQ; the y-axis shows corresponding ranks obtained via benchmarking with mean performance of the remaining datasets. A Spearman correlation of 0.71 is obtained (N=7852). **C** gives the distributions of LODOCV results under different quantification settings. Mean Spearman correlations are marked by red triangles.

Per the reviewer's suggestion, we now added the Kruskal-Wallis test to check the variance of ranks grouped by instrument type generating the benchmarking datasets. For example, the DDA datasets were generated by three groups of datasets including Tims TOF pro, older instrument LTQ-Orbitrap and newer Orbitrap machines such as Q Extractive and Orbitrap Fusion Lumos. We have at least 5 samples in each of the three groups, where the Kruskal-Wallis test can be used to check whether the rankings of a workflow obtained by different datasets (We ranked workflows by every available benchmarking dataset independently) from the three instrument groups have no median difference at $p\text{-value} \leq 0.05$.

The reason why we didn't conduct ANOVA to check the ranking variance among datasets directly is that if we regard each dataset as a group, the group sizes are always smaller than 3. The ANOVA test or other variance checking statistical methods always require bigger group size, e.g., ANOVA requires 15 at least. In our **Figure 2D**, we show the Kruskal-Wallis tests of the top-30 workflows under different label-free quantification settings, e.g., FG_DDA (refers to label-free DDA data quantified by platform FragPipe). Most of the top-ranked workflows are insensitive to the instrument type generating datasets used for performance evaluation. We quote the method of the Kruskal-Wallis test, **Figure 2D** and related discussions as follows:

Figure 2D displays the Kruskal-Wallis (KW) test results checking if workflow ranks are sensitive to instrument types. The x-axis lists the ranks of the top 30 workflows ranked by mean performances. The y-axis shows the log transformed p-value of the KW tests. Most comparisons are non-significant, suggesting workflows are not sensitive to instrument types.

The Kruskal-Wallis test method is described in subsection of “**Kruskal-Wallis test**” at page 26 of main_R1_clean.docx:

Kruskal-Wallis test

The biggest contrast number of the 24 gold standard datasets is 3, thus it is impossible for us to check whether the benchmarking is sensitive to the dataset used for performance evaluation with statistical methods (the group size is no bigger than 3). The DDA, DIA and TMT data were generated by different types of instruments (see **Table 1**), e.g., newer ones such as Tims TOF or older ones such as LTQ Orbitrap Velos. For the 20 contrasts (SCIEX Triple TOF5600 and SCIEX Triple TOF6600 were excluded) of the 12 DDA datasets, we group them into 3 categories including Tims TOF pro (N=8), LTQ-Orbitrap/Velos (N=7) and other Orbitrap machines (N=5). Each of the 20 contrasts were used to rank the workflows based on the 5 indicators (above benchmarking method). Then, we conducted the Kruskal-Wallis test⁴² to check whether a workflow’s rankings based on the three groups of contrasts have the same median at threshold of p-value≤0.05. Similarly, for the 17 contrasts of the DIA data, 2 instrument groups are available such as the Tims TOF pro (N=7) and the other Orbitrap machines (N=10). For the TMT data, we cannot split them into groups with the minimum size requirement of N≥5, so no Kruskal-Wallis test was conducted.

The Kruskal-Wallis test results are discussed in subsection of “**Optimal workflows are predictable and are settings-specific**” at page 7 of main_R1_clean.docx:

The mean ranks may be unstable if performance metrics are sensitive to the choice of instrument. To assay this, we used the Kruskal-Wallis test⁴² (see **Methods**). We plot 30 top performing workflows ranked by mean performance under different settings against the log transformed p values of the Kruskal-Wallis tests (**Figure 2D**). We conclude that under most circumstances, the rankings of top workflows are not sensitive to instrument type (shown as gray markers in **Figure 2D**). However, some workflows involving DIANN_DIA do show some sensitivity to instrument type, with 4 out of 30 workflows reporting p-values < 0.05, indicating significant ranking position differences between instrument types (**Supplementary file 2**).

Reviewer-2-Comment-7:

The Figure captions need to be revised, and the figure description extended. Please provide reports of the y and x axis.

Specifically, Figure 3, caption

"Panel A: shows the pairwise comparison of expression matrix types of FragPipe and maxquant." unclear

Furthermore, Panel B states, "B compares five expression matrix types from DIA-NN." But what is the mean-pairwise difference on the Y-axis?

Response to Reviewer-2-Comment-7: We thank the reviewer's suggestion about the improvement of our figure legends. We have updated our figures and their captions accordingly. We compared the options in each step of a workflow by the way of pairwise comparison. We take the comparison of a pair options of the DEA tool as an example, e.g., limma and ROTS. For the workflow **A** using limma, where **A**: expression matrix MaxLFQ + no normalization + knn for MVI + limma for DEA, we can find the corresponding workflow **B**: MaxLFQ + no normalization + knn + ROTS. **A** and **B** have the only difference of DEA tool option, so their performance difference must be caused by the DEA tool option they choose, i.e., limma and ROTS. We calculate the performance difference of workflows **A** and **B** tested on the same dataset to indicate the performance of **A** and **B**. Thus, we can find all such workflow pairs with only difference of the DEA tool and calculate their performance differences to conduct a comprehensive evaluation of DEA tools limma and ROTS using different datasets and accepting different preprocessed expression matrices.

In our previous version manuscript, the y-axis label of "mean pairwise difference" of Panel B refers to the mean value of differences between workflow pairs containing the two compared matrices shown in x-axis, e.g., A-E (performance of workflow containing "MaxLFQ" minus performance of workflow contain "raw").

In **Figure 4A** of this revised version manuscript, we show an example of using pairwise comparison method to compare the expression matrix types available for settings of DIANN_DIA (means DIA data quantified by DIA-NN) and spt_DIA based on the pAUC(0.01) scores. We quote the method description about the pairwise comparison

for option comparison, the **Figure 4A** and related results as follows:

We describe the method of pairwise comparison in subsection “**Comparisons among choices in a single step of a workflow**” at page 27-28 of main_R1_clean.docx:

After obtaining quantification results from a quantification setting, such as analysis of label-free DDA data with FragPipe (setting of FG_DDA), a comprehensive DEA workflow integrates several key selection steps, including:

- a) An expression matrix that contains the expression levels of identified proteins;
- b) A normalization method to reduce bias or noise;
- c) An algorithm for imputing missing values in the selected expression matrix;
- d) A DEA tool for conducting the final differential expression analysis.

Each step plays a crucial role. To examine the impact of a particular step, we simply maintained the options for other steps while varying the options of the step under investigation. To compare any two options for a given step, e.g., protein top1 intensity and protein MaxLFQ intensity in step a, we calculated performance differences of workflow pairs where they are alike in every other way except the choice of option. Different options in each step can be ranked by their pairwise comparisons. For a given step, we first count the frequencies of the options winning in pairwise mean performance comparisons. Then, the option with bigger frequency will be ranked higher. If two options have the same win frequencies, then their median performances will be compared for ranking. All five performance indicators were used to ranking the options separately, the average rank of the 5 independent ranks was used for as final ranks (similar to above workflow ranking).

The discussions of the example of pairwise comparison for expression matrix comparison are in subsection of “**General usefulness of options across different workflow settings**” at page 12 of main_R1_clean.docx:

We opted to compare options in each workflow step via a pairwise comparison method (see **Methods**). The options are compared across 5 performance metrics. **Figure 4A** shows an example of pairwise comparison of expression matrix types available for DIANN_DIA and spt_DIA based on pAUC(0.01) scores. The pairwise difference of pAUC(0.01) for “dlfq-LFQ” is calculated by subtracting the pAUC(0.01) score of a workflow using the expression matrix LFQ from the pAUC(0.01) score of the corresponding workflow that has replaced LFQ with dlq. We can infer that incorporating the dlq option is better than using LFQ under both DIANN_DIA and spt_DIA since the mean pairwise difference is higher than 0 (red points in **Figure 4A**). Similarly, both dlq and LFQ are superior to top1 and top3. In addition, top3 is better than top1. Based on these reciprocal comparisons, we can derive the following rank order: 1:dlq, 2:LFQ, 3:top3 and 4:top1. These matrix types can also be ranked by the remaining 4 performance metrics in the same way. Finally, the five performance metrics

are averaged to finalize the order of options in each step (see **Methods**).

Figure 4A shows the distributions of pAUC(0.01) differences between a pair of matrices showing in x-axis. The “dlfq-LFQ” in x-axis indicates subtracting the pAUC(0.01) value of a LFQ related workflow by the corresponding dlfq related workflow where only matrix types are different between them. The colors of the boxplots show the settings of the compared matrix types. The red points indicate mean differences.

Comments from Reviewer 3

This is a very technical manuscript describing the process of evaluating over 10,000 various ‘workflows’ (combinations of tools and parameter settings) for three commonly used pipelines, FragPipe, MaxQuant, and DIA-NN. The author have clearly put a lot of effort in this work.

Reviewer-3-Comment-1:

Unfortunately, this reviewer had a very hard time following the manuscript and understanding the significance of this work. The authors consider multiple different datasets, DIA and DDA data, tools, evaluation metrics etc. The plots are highly technical, the manuscript is full of terminology and numbers. The conclusions they arrive at are somewhat reasonable and expected, that is, that the results vary widely depending on various decisions made at each step in the data analysis process. However, after reading the manuscript, I do not have a clear ‘take home’ message.

Response to Reviewer-3-Comment-1: We thank the reviewer's affirmation of our efforts in the recommendation of optimal workflows for proteomics data differential expression analysis. We have summarized our findings in 3 key take-home messages. These are raised in the abstract, end of the introduction section and elaborated in the discussion section. We attach these 3 points below (summarized in **Introduction** at page 4-5 of main_R1_clean.docx):

First, we report that optimality is predictable. Via cross validation, we confirm that a workflow's performance level (be it high-performing or low-performing) is predictable (average F1 score or MCC score > 0.84). To further confirm this, we performed leave-one-dataset-out cross-validations (using one dataset as test data while the remaining datasets are used for benchmarking to rank the workflow performances on the test data) and find good generalization performance (mean spearman correlation coefficients > 0.58).

Next, some steps in a workflow are more important in determining outcomes in a setting specific manner. Through feature importance analysis and ANOVA tests, we find that normalization and DEA statistical method exert greater influence (e.g., achieving 30% feature importance and with higher F-value by the ANOVA tests indicating wider group variances) than other steps for label-free DDA data and TMT data. Whereas for label-free DIA data, the matrix type is also important (in addition to normalization and DEA statistical method). From frequent pattern mining of high-performing workflows (ranked at top 5%), we further report that high-performing workflows of label-free data are enriched for the directLFQ intensity¹⁹, no normalization and incline SeqKNN^{20,21}, Impseq^{21,22} or MinProb²³ (probabilistic minimum) for imputation while eschewing simple statistical tools (e.g., ANOVA²⁴, SAM²⁵ and t-test²⁶ are enriched in low-performing workflows).

Finally, we report that integration of multiple workflows is beneficial for expanding differential proteome coverage but requires more development. Obviously, given the multitude of workflows possible with a given data, there is the potential for complementary and expansion of insight if these workflows are integrated or merged. There are several ways of doing this. However, given the increased attention to machine learning approaches currently, we decided to go with an ensemble inference approach that integrates DEA results from individual top-performing workflows. This ensemble approach can increase true positives, leading to improvements of mean pAUC(0.01) by 1-4% and improvements of the mean G-mean scores by as high as 11% across 6 different quantification settings. In particular, the integration of top 1st workflows using spectral counts and intensities extracted with directLFQ and top0 (which incorporates all precursors quantification methods) improved the DEA performance more than any of the best single workflows did, gaining a pAUC(0.01) of 4% under the FG_DDA setting (using FragPipe to quantify label-free DDA data). This suggests that while spectral count may not work as well as intensity in DEA workflows, combining these multiple workflows provides complementary information that enhances DEA outcomes.

However, the increase in true positives also comes with the risk of false positives. To mitigate such risks, further development on workflow integration approaches is needed.

Reviewer-3-Comment-2:

First, I think the challenge here is that each biological study is different, and it is simply unrealistic to come up with universal guidelines based on the analysis of somewhat artificial benchmark datasets. For example, biological samples such as affinity purified protein complexes require completely different analysis strategies compared to say organellar protein profiling, and different again for large scale plasma protein profiling experiments. In certain cases, normalization of the data is absolutely necessary, in other cases it is not advisable. I do not think what we learn from spike-in benchmarking datasets can be easily generalized to most of the biological studies.

Response to Reviewer-3-Comment-2: We appreciate the reviewer's comments regarding the generalizability of our benchmarking. We agree that benchmarking, primarily based on spike-in datasets where technical variance is the predominant factor, may not be directly applicable to projects characterized by substantial biological variances. Thus, following this line of logic, our top-ranked workflows may not guarantee the best performance could be achieved in real life data analysis (even in our benchmarking, the top-ranked workflows achieve higher average performance but not always the best for every dataset). However, rules assuring better performance (given specific settings) can be found (as in our results). This allows us to avoid getting bad results at the get-go, without even considering the issue of biological variance.

While imperfect, working with spike-in datasets allows us to find trends for guiding workflow selection. For example, we suggest applying directLFQ and MaxLFQ, which may result in better performance. In addition, directLFQ works better than MaxLFQ through our benchmarking. Both directLFQ and MaxLFQ have built-in normalization during quantification, this may be another reason why we observed that no normalization is preferred (besides no biological variance was simulated). Though we don't know which workflow works the best in real life data analysis, using directLFQ and without additional data normalization should assure relatively good performance when we don't have true labels of the data (this is true in analyzing real life data).

Once we have a recommended workflow based on the pre-determined settings, we can explore further fine-tuning. It is outside the scope of the paper, but the reviewer is astute in pointing out that in some cases, normalization is necessary as a pre-condition before differential analysis. This can be platform, experiment or even biological comparison specific.

In addition, biological variance, which as you say, is idiosyncratic to every study, is therefore impossible to simulate artificially and convincingly. Using spike-in data to evaluate the effectiveness of tools is a popular approach in various research fields and has proven effective in most situations. This commonality allows us to produce our

assemblage of data, allowing us to execute our brute-force benchmarking experiments.

We quote the discussion about the limitation of lacking simulation of biological variance during our benchmarking as follows:

We discussed the normalization method recommendation in subsection “**The ranked workflows provide a means for identification of algorithms associated with high performance**” at page 19 of main_R1_clean.docx:

- Normalization performed surprisingly badly and may even worsen the DEA performance. Some advanced quantification methods (directLFQ and MaxLFQ) have built-in peptide-level or protein level normalizations, thus usually no additional normalization is required. We recommend checking the distributions of the expression matrix in advance to determine whether normalization is required (and not just do it routinely, especially when it is not needed). If needed, regression-based normalization methods, e.g., `lossf` and `Rlr`, and `center.median` or `center.mean` are good options.

The additional discussion about the limitation of no biological variance simulation in subsections of “**Spike-in data**” and “**Fidelity and applicability to the real world**” at page 20 of main_R1_clean.docx:

Spike-in data

We used spike-in data to benchmark workflows. Spike-in datasets mainly simulate technical variation by producing technique replicates for each sample. In real-world proteomics data, biological variation always exists and may manifestly quite differently from technical variance. The inability to simulate biological variation exactly may introduce some degree of bias to our benchmarking results. However, there is currently no universally accepted way of simulating biological replicates. Attempting to do so at this time may invite scepticism. There are some promising new approaches based on deep generative models⁶⁰ for in silico simulation of biological variation, we will look at this closely for inclusion in our next round of benchmarking efforts.

Fidelity and applicability to the real world

Although we try to test as many datasets and methods as possible, it does not mean the predictions and recommendations built on these will work on any data. We cannot guarantee it. Real world data is complex and covers many situations (scenarios or sources or variation) that cannot exist in our relatively simpler simulations. Still, we think that the availability of OpDEA as a recommendation resource provides a useful starting point. Researchers will no longer need to simply copy a workflow from some other publication, with no knowledge of whether the workflow really works well for their own data. In this regard, using OpDEA can positively enhance proteomics research.

Reviewer-3-Comment-3:

Second, most users of pipelines such as FragPipe, DIA-NN, or MaxQuant will have a hard time connecting this manuscript with how they use these pipelines in practice. This is so in part because these pipelines generate quantification matrices that are then used as input into other downstream tools. Most commonly, the users would use Perseus, or MSstats, or one of the several python or R-based pipelines (e.g., LFQ-Analysis) etc. The authors, however, are not evaluating various settings and parameters available in those popular downstream data analysis pipelines. So the reader is left wondering what they should actually do if they are using say MSstats or Perseus. I believe to be useful, the downstream evaluation performed in this work should closely link to what options are actually available in the commonly used downstream tools for proteomics data.

Response to Reviewer-3-Comment-3: We appreciate the reviewer's comment regarding the connection of our work with users' tasks. Perseus, being a non-open resource software for proteomics data analysis, lacks compatibility for interaction via R or Python scripts. The LFQ-Analysis is a web tool designed for conducting differential expression analysis, enabling the examination of proteomics data with a selected imputation algorithm. However, both tools are designed for accepting quantification results from Maxquant. In LFQ-Analysis, only a limited set of imputation algorithms is available, and neither normalization methods nor differential expression analysis (DEA) tools are tunable. MSstats is an R package for statistical analysis of quantitative mass spectrometry-based proteomic experiments. We already compared MSstats-based DEA workflows with other workflows (see **Figure 1F**, we also listed the MSstats-related workflows in our **Supplementary file 2** where MSstats was specified to be the DEA statistical tool). However, MSstats has limitations, as it only supports a limited number of quantification approaches, normalization methods, and MVI (Missing Value Imputation) algorithms, rendering it less flexible. Additionally, comparing MSstats-related workflows with those generated by other popular statistical tools, such as limma, is not straightforward for users.

We now enable our webserver (<http://www.ai4pro.tech:3838/>), standalone toolkit (can be downloaded for our webserver) and R package (<https://github.com/PennHui2016/OpDEA>) to conduct differential expression analysis directly by selecting a workflow or accepting our suggested workflow. Users can upload their quantification reports, such as FragPipe's 'combined_protein.tsv' file, to the server. Alternatively, they can utilize the offline standalone toolkit (recommended, as there is no need to upload data) to test various differential expression analysis workflows. Results from the differential expression analysis will be reported with specified thresholds. Our server and toolkit can be easily extended with additional functions in future versions, enhancing their power and utility for the proteomics community.

Reviewer-3-Comment-4:

Third, a significant part of the manuscript is devoted to ensemble approaches, that is, application of multiple workflows followed by some integration of the results. Unfortunately, for an average user such a strategy is not practical as it requires

significant bioinformatics expertise. Perhaps even more importantly, it complicates the analysis and thus makes it harder for people to describe it properly and get their papers through peer review. I would personally not advise anyone to employ more complicated, ensemble-based analysis strategies, especially given very modest improvement reported by the authors (just a few percent in most cases).

Response to Reviewer-3-Comment-4: We thank the reviewer's comment about ensemble inference. We agree at this point of writing, ensemble approaches towards proteomic analysis are uncommon. But it did strike us during our research that many high performing workflows did not yield common proteins, combining the workflows and evaluating the outcomes was tempting. Given the increasing availability of computational resources and democratization of advanced AI and data analytics approaches, we felt it was good to get an early sense on the viability of such ensemble approaches. It is too early, and we are not trying to pass this as a new paradigm that may disadvantage our fellow researchers who may not have access to advanced computational resources/tools.

Our finding is that while there seems to be promise, we also advise caution, as concomitant coverage of true positives can also be accompanied by false positives. More advanced development in this area is therefore needed.

Furthermore, we believe there will be readers keen to explore such methods, and whom can perhaps, help drive future developments in this area. Thus, to make the ensemble inference easier to use, we extended our webserver, standalone toolkit and R package with the function of conducting differential expression analysis with the suggested ensemble inference workflows. The user can try it by providing their raw quantification outputs from FragPipe, Maxquant, Spectronaut, or DIA-NN.

As we have reperformed many of our experiments with the expanded datasets, we would also like to update the reviewer on our latest findings. In this revised version, we showed that the ensemble inference approach `ens_mutli-quant`, i.e., integration of top-rank 1st workflows that using different quantification results (expression matrix types) as input, can improve the DEA performance comparing to single workflows. It improves the pAUC (FPR \leq 0.01) scores ranging from 1%-4% while the improvement of the G-mean scores range from 6%-11%. Especially, more true positives can be detected by `ens_multi-quant` comparing to single workflows. At most 3 top-ranked workflows are combined, so the running time won't increase a lot (see below quoted supp. Tab. 5 showing the running speed comparison). We quote the method and related results as follows:

We described the `ens_multi-quant` method in subsection of "**Ensemble Inference**" at page 28 of `main_R1_clean.docx`:

We proposed two ensemble inference strategies: (1) integration of no less than 2 of top

1st workflows using spectral counts, protein top0, top3, MaxLFQ and directLFQ intensities for label-free DDA data (we name it ens_multi-quant), and (2) integration of top K workflows in our overall rankings (ens_topk). For label-free DIA data, ens_multi-quant integrates no less than 2 of the top 1st workflows using top1, top3, MaxLFQ and directLFQ intensities. For TMT data, ens_multi-quant is only applicable when FragPipe is used for quantification where TMT-Integrator abundance, TMT-Integrator ratio and Philosopher intensity are available for integration.

The comparison of ensemble inference and single workflows are discussed in subsection of “Ensemble inference through integration of top-ranked workflows improves DEA coverage” at page 15 of main_R1_clean.docx:

Figure 5A compares the performance by ensemble inferences and those by top1 single workflows under different settings. The red triangle shows the mean performance. TOP1 refers to the best workflow specific to a given setting. Ens_multi-quant is an ensemble approach combining top1 workflows using different expression matrices. Ens_topk is an ensemble approach integrating the top-ranked k workflows.

In general, ens_multi-quant improves performance (**Figure 5A**). The performance gains based on mean pAUC(0.01) ranges from 1.17% to 4.31% across different settings (notably, there is a 3.95% decrease observed under the spt_DIA setting). For mean pAUC(0.05) and mean pAUC(0.1), gains range from 0.9% to 3.4% and from 0.9% to 3.8% were obtained (again, 2.4% and 1% decreases were achieved under spt_DIA). We observed gains in mean nMCC ranging from 1% to 1.7% under settings FG_TMT, DIANN_DIA and spt_DIA while decreases of 0.1% and 3.4% were seen under MQ_DDA and FG_DDA respectively. The performance gain in terms of the mean G-mean score ranges from 6% to 11%, and is even more appreciable at 11.14% (or 12.68% if in terms of the median performance) under DIANN_DIA. Although ens_topk improves DEA performances under most settings, it is inferior to ens_multi-quant. The improvement of mean pAUC(0.01) score ranges from 0.04% to 3% with the exception of decreasing by 1.73% under MQ_DDA (see more details in **Supplementary results** and **Supplementary file 8**). The hurdle model⁵⁴ (which transforms p-values to z-values and then combines the z-values into a χ^2 statistic) for p-value integration is more frequently applied than the fisher method (see **Supplementary file 8**).

Below quoted **supp. Tab. 5** (in page 14 of our Supplementary results.docx file) shows the comparison between running speeds of single top-ranked workflows and the ens_multi-quant workflows.

supp. Tab. 5 Top-ranked workflow running time comparison

Setting	Method	Dataset	Workflows involved	Running time
DIANN_DIA	TOP1	HEqe408_DIA	1	27.94s
DIANN_DIA	ens_multi-quant	HEqe408_DIA	3	207.03s
spt_DIA	TOP1	HEqe408_DIA	1	103.4s
spt_DIA	ens_multi-quant	HEqe408_DIA	3	254.48s

Yours sincerely,

Hui Peng, Jinyan Li and Wilson Goh

REVIEWERS' COMMENTS

Reviewer #1 (Remarks to the Author):

The authors have completed a substantial amount of work in revision which led to significantly improved quality of the current manuscript. Particularly, the reviewer very much appreciated the discussion part which concisely summarizes the major findings, offers useful guidelines and also discusses limitations honestly. Only some minor points are suggested here to further strengthen this systematic work.

1. Summary of findings at the end of Introduction can be shortened or removed as it overlaps with Discussion.

The authors have done substantial amount of work in revision which led to significantly improved quality of the current manuscript. Particularly, the reviewer appreciated the discussion part which concisely summarizes the major findings, offers useful guidelines and also discusses limitations honestly. Some additional points are provided here to further strengthen this work.

1. In Abstract, a few abbreviations such as "lossf", SeqKNN, Impseq, MinProb for MVI, and certain technical details can be deleted. Summary of findings at the end of Introduction can be also shortened or removed as it mainly overlaps with Discussion.

2. Normalization

The authors have repeatedly stated that normalization is unnecessary, as seen in the following statement: "Normalization performed surprisingly badly and may even worsen the DEA performance." Yet two important issues are noteworthy here. First, as mentioned by the authors, normalization of the quantification matrices is enabled in built-in modules implemented in MaxQuant, FragPipe (IonQuant), DIA-NN, Spectronaut, or directLFQ. Second, the necessity of normalization is largely dependent on the dataset, given that real-world MS-based proteomic data (usually with higher variations than the benchmark data) published in high-impact journals typically apply normalization via software built-in methods or manual processing. Thus, it is suggested to tone down this possibly misleading statement.

3. Imputation

There exists an undisclosed consensus in the community that imputation would yield artefacts in MS-based quantification and should generally be avoided. Alternatively, many other methods have been developed to mitigate missing values, such as MBR in MaxQuant and IonQuant, and the second search in DIA-NN. Given that the benchmark datasets differ from real-world data in many aspects, it would be risky to draw the conclusion that missing value imputation can enhance the performance of DEP analysis. In particular, most imputation methods rely on local or global patterns of quantification present in multiple runs, thereby making imputation simpler for benchmark datasets (easier to discern the pattern) than for real-world data (more unpredictable pattern and larger variations between runs). Thus, the conclusion drawn from this study may not be universally applicable, which could be explicitly mentioned in Discussion.

4. To thoroughly assess the effectiveness in integrating multiple workflows for DEP analysis is a highlight of this work and it is less investigated previously. The authors have shown the ens_multi-quant approach improves performance by increasing the mean pAUC(0.01) by < 5% for multiple datasets, which is in fact a minor gain. However, in reality, we often see an increased number of DEPs when searching the same data with different platforms (so many are available now), which is a common practice employed in quite a few studies. It would be interesting to test combining DEP lists given by two platforms using these well-designed benchmark data. For example, for datasets shown in Fig. 5D, would it be possible for the authors to try pooling DEPs from spt-DIA-tims and DIANN-DIA-tims, and pooling DEPs from MQ-DDA-tims and FG-DDA-tims to see how pAUC and the number of TP/FP vary with this type of ensemble method? This evaluation would offer critical information for users to decide whether it is worthwhile to run multiple platforms if they want to maximize the DEP coverage. Thanks for the additional effort.

Reviewer #2 (Remarks to the Author):

Dear Editor and Authors,

The authors addressed many reviewer comments, which helped improve the manuscript's readability.

I suggested using ANOVA to study the responses obtained from this computer experiment instead of the CatBoost. However, I am sorry for making this suggestion.

This analysis, now added to the CatBoost results, makes the manuscript more challenging to follow. When I first read the text, the article was hard to read and misinformed readers by overinterpreting the data.

The analysis should focus on describing the results rather than building prediction models or reporting p-values, because of the biased benchmark datasets, which exclusively capture technical variability.

I am pleased that now there are Figures that show:

- ranks of the various methods (Figure 4),
 - show boxplots of the pAUCs or the number of TP and FP (Figure 5),
 - or Figure 2E, where the pAUC and G-mean are correlated.
- , which is how the results should have been presented and discussed in the first place and exclusively.

However, many authors think they must report as many significant p-values as possible and use machine learning and AI whenever possible. And they are not wrong in thinking this. It is what many journals, specifically Nature editors and reviewers, expect to see.

The editor must focus on ensuring FAIR principles:

https://en.wikipedia.org/wiki/FAIR_data

It happens that the websites maintained by authors go offline directly after publication. Therefore, I would ask the editor to ensure that all the data is deposited in a data repository maintained by a third party.

The data deposited must include the outputs of the DIA-NN, Spectronaut, MQ, etc, software used as input to workflows examined.

Furthermore, the authors should generate and publish a Docker image with all the scripts that automatically download the data from the third-party repository and replicate the analysis (7k + workflows).

Kind regards

Reviewer #2 (Remarks on code availability):

The editor must focus on ensuring FAIR principles:

https://en.wikipedia.org/wiki/FAIR_data

It happens that the websites maintained by authors go offline directly after publication. Therefore, I would ask the editor to ensure that all the data is deposited in a data repository maintained by a third party.

The data deposited must include the outputs of the DIA-NN, Spectronaut, MQ, etc, software used as input to workflows examined.

Furthermore, the authors should generate and publish a Docker image with all the scripts that automatically download the data from the third-party repository and replicate the analysis (7k + workflows).

Reviewer #3 (Remarks to the Author):

The authors performed a substantial revision and addressed many of the questions raised by the reviewers. While I still think the paper is too technical, the analysis is solid and may be useful for users of proteomics pipelines to optimize the parameters. In light of the efforts that authors put into the revision I support the publication.

Author point-by-point response letter

Manuscript: NCOMMS-23-27297A (Optimizing Differential Expression Analysis for Proteomics Data via High-Performing Rules and Ensemble Inference)

April 10, 2024

We would like to thank the reviewers for their valuable comments. We have revised our manuscript accordingly. Our point-by-point responses are as follows.

Below, we have highlighted changes made in the main revised text (in red) in the main_R2_with_revision_highlighted.docx file. Please note the page numbers cited here corresponds to the main_R2_clean.docx file without revision highlights. To help reviewers locate key information, we copy these changes here (colored in red).

Comments from Reviewer 1

Reviewer #1 (Remarks to the Author):

The authors have completed a substantial amount of work in revision which led to significantly improved quality of the current manuscript. Particularly, the reviewer very much appreciated the discussion part which concisely summarizes the major findings, offers useful guidelines and also discusses limitations honestly. Only some minor points are suggested here to further strengthen this systematic work.

Reviewer-1-Comment-1: Summary of findings at the end of Introduction can be shortened or removed as it overlaps with Discussion.

The authors have done substantial amount of work in revision which led to significantly improved quality of the current manuscript. Particularly, the reviewer appreciated the discussion part which concisely summarizes the major findings, offers useful guidelines and also discusses limitations honestly. Some additional points are provided here to further strengthen this work.

In Abstract, a few abbreviations such as “lossf”, SeqKNN, Impseq, MinProb for MVI, and certain technical details can be deleted. Summary of findings at the end of Introduction can be also shortened or removed as it mainly overlaps with Discussion.

Response to Reviewer-1-Comment-1: We appreciate the reviewer’s acknowledging our efforts in improving our manuscript per the valuable suggestions by the reviewers. We shortened our Abstract and removed the abbreviations and technical details in the Abstract. We quote the revised Abstract as follows:

Revised Abstract at page 2 of main_R2_clean.docx:

Identification of differentially expressed proteins in a proteomics workflow typically encompasses five key steps: raw data quantification, expression matrix construction, matrix normalization, missing value imputation (MVI), and differential expression analysis. The plethora of options in each step makes it challenging to identify optimal workflows that maximize the identification of differentially expressed proteins. To identify optimal workflows and their common properties, we conduct an extensive study involving 34,576 combinatoric experiments on 24 gold standard spike-in datasets. Applying frequent pattern mining techniques to top-ranked workflows, we uncover high-performing rules that demonstrate optimality has conserved properties. Via machine learning, we confirm optimal workflows are indeed predictable, with average cross-validation F1 scores and Matthew's correlation coefficients surpassing 0.84. We introduce an ensemble inference to integrate results from individual top-performing workflows for expanding differential proteome coverage and resolve inconsistencies. Ensemble inference provides gains in pAUC (up to 4.61%) and G-mean (up to 11.14%) and facilitates effective aggregation of information across varied quantification approaches such as topN, directLFQ, MaxLFQ intensities, and spectral counts. However, further development and evaluation are needed to establish acceptable frameworks for conducting ensemble inference on multiple proteomics workflows.

We thank the reviewer for the kind suggestion of creating better separation between the introduction and discussion. We reviewed the content in both sections, but we just wanted to highlight that the introduction only contains a list of key findings/achievements. This helps the reader quickly evaluate if this paper is of value to them. We do not cover the key findings in the discussion but rather, emphasize detailed recommendations and limitations of our study. We do acknowledge this is a rather verbose manuscript and we should do more to streamline content. Hence, we shortened the lists of our findings and quote the revised part as follows:

The revised content at the end of the Introduction at page 3-4 (rows 93-122) of main_R2_clean.docx:

Here, we list our key findings:

First, we report that optimality is predictable, where workflow performance levels can be classified accurately (average F1 score or MCC score > 0.84) and workflow ranks generalize well to unseen datasets (mean spearman correlation coefficients > 0.56 between ranks based on unseen datasets against benchmark datasets).

Next, some steps in a workflow are more important in determining outcomes in a setting specific manner. Normalization and DEA statistical method exert greater influence than other steps for label-free DDA data and TMT data. Whereas for label-free DIA data, the matrix type is also important. We find that high-performing workflows of label-free data are enriched for the directLFQ intensity¹⁵, no normalization (referring only to

distribution correction methods that are not embedded with any particular settings) and incline SeqKNN^{16,17}, Impseq^{17,18} or MinProb¹⁹ (probabilistic minimum) for imputation while eschewing simple statistical tools (e.g., ANOVA²⁰, SAM²¹ and t-test²² are enriched in low-performing workflows).

Finally, we report that workflow integration is beneficial for expanding differential proteome coverage but is also a double-edged sword. Given the increased attention to machine learning approaches, we design an ensemble inference approach that integrates DEA results from individual top-performing workflows. The ensemble approach can increase true positives, leading to improvements of mean pAUC(0.01) by 1.17-4.61% and improvements of the mean G-mean scores by as high as 11.14% across 6 different quantification settings. In particular, the integration of top 1st workflows using top0 intensities (which incorporates all precursors quantification methods) and intensities extracted with directLFQ and MaxLFQ²³ improved the DEA performance more than any of the best single workflows did, gaining a pAUC(0.01) of 4.61% under the MQ_DDA setting (using Maxquant to quantify label-free DDA data). This suggests that while top0 may not work as well as directLFQ intensity in DEA workflows, combining these multiple workflows provides complementary information that enhances DEA outcomes. However, the increase in true positives also comes with the risk of false positives. To mitigate such risks, further development on workflow integration approaches is needed.

Reviewer-1-Comment-2: Normalization

The authors have repeatedly stated that normalization is unnecessary, as seen in the following statement: "Normalization performed surprisingly badly and may even worsen the DEA performance." Yet two important issues are noteworthy here. First, as mentioned by the authors, normalization of the quantification matrices is enabled in built-in modules implemented in MaxQuant, FragPipe (IonQuant), DIA-NN, Spectronaut, or directLFQ. Second, the necessity of normalization is largely dependent on the dataset, given that real-world MS-based proteomic data (usually with higher variations than the benchmark data) published in high-impact journals typically apply normalization via software built-in methods or manual processing. Thus, it is suggested to tone down this possibly misleading statement.

Response to Reviewer-1-Comment-2: We thank the reviewer's suggestion to tone down the possible misleading statement of optimal normalization method suggestion. During the proteomics data quantification process with the four quantification platforms, we left most of the parameters as default including those that are normalization related. Thus, if these quantification platforms always normalize the quantification results by default, then during the workflow evaluation where we conduct data normalization after the expression matrix extraction, the normalization step may be an additional normalization to the built-in normalization. To reduce the misleading, we modified the normalization suggestion related statements.

In **Abstract** at page 2 (rows 30-31) of main_R2_clean.docx, we removed the detailed high-performing rules we found, please refer to the tracked amendments below:

Applying frequent pattern mining techniques to top-ranked workflows, we uncovered high-performing rules that demonstrate optimality has conserved properties. ~~Some high-performing rules include using directLFQ for quantification, avoiding normalization or applying “lossf”, and deploying SeqKNN, Impseq, or MinProb for MVI. Finally, for differential expression analysis, ROTS and limma are good choices.~~
Via machine learning,

In **Introduction** at page 3-4 (rows 104-105) of main_R2_clean.docx, we added the description of no normalization:

We found that high-performing workflows of label-free data are enriched for the directLFQ intensity ¹⁵, no normalization (referring only to distribution correction methods that are not embedded with any particular settings) and incline SeqKNN ^{16,17}, Impseq ^{17,18} or MinProb ¹⁹ (probabilistic minimum) for imputation while eschewing simple statistical tools (e.g., ANOVA ²⁰, SAM ²¹ and t-test ²² are enriched in low-performing workflows).

In the section **Frequent patterns extracted from high-performing workflows in Results** at page 8 (rows 293-295) of main_R2_clean.docx, we added description of no normalization:

Normalization methods “no_norm” (no additional normalization to data extracted by quantification platforms with default settings, built-in normalization may have been conducted, e.g., FragPipe enables “Normalize intensity across runs” by default, more details see our Supplementary Note 2) and “lossf” ^{37,38}, and DEA tool “limma” are frequently selected for all the six settings with support ratio bigger than 0.10 (see Supplementary Data 5).

In section **Recommendations and resources for selecting optimal workflows in Discussion** at page 13 (rows 514-515) of main_R2_clean.docx, we add description of no normalization:

- For label-free DDA data quantified by FragPipe (i.e., setting FG_DDA), we recommend a workflow combining protein directLFQ intensity, no normalization (refers to no additional normalization to data extracted with default quantification settings), SeqKNN for MVI, and ROTS (or limma, if running time is a concern) for DEA.

In section **Ranked workflows provide a means for identification of algorithms associated with high performance in Discussion** at page 14 (rows 580-581) of

main_R2_clean.docx, we removed previous statement about the normalization and rephrased our suggestion of choosing normalization methods:

- ~~Check data variance in advance to determine the need for data normalization. Normalization performed surprisingly badly and may even worsen the DEA performance.~~ Some advanced quantification methods (directLFQ and MaxLFQ) have built-in peptide-level or protein level normalizations, thus usually no additional normalization is required. We recommend checking the distributions of the expression matrix in advance to determine whether normalization is required (and not just do it routinely, especially when it is not needed). If needed, regression-based normalization methods, e.g., lossf and Rlr, and center.median or center.mean are good options.

Reviewer-1-Comment-3: Imputation

There exists an undisclosed consensus in the community that imputation would yield artefacts in MS-based quantification and should generally be avoided. Alternatively, many other methods have been developed to mitigate missing values, such as MBR in MaxQuant and IonQuant, and the second search in DIA-NN. Given that the benchmark datasets differ from real-world data in many aspects, it would be risky to draw the conclusion that missing value imputation can enhance the performance of DEP analysis. In particular, most imputation methods rely on local or global patterns of quantification present in multiple runs, thereby making imputation simpler for benchmark datasets (easier to discern the pattern) than for real-world data (more unpredictable pattern and larger variations between runs). Thus, the conclusion drawn from this study may not be universally applicable, which could be explicitly mentioned in Discussion.

Response to Reviewer-1-Comment-3: We thank the reviewer's astute observation. Indeed, we also found similar outcomes in our other work involving imputation algorithms. In a personal sense, I feel there is a parallel in education --- we train students to prepare them for the outside world. But those A+ students who go out may not necessarily be star performers in real life!

Hence, we agree that the suggestion of imputation algorithms based on the conclusion observed from the analysis of benchmarking data may not work as the real-world data are more complex and with bigger variances. But this critique can also be extended to the rest of our workflow prediction work. While we cannot fully incorporate the full complexity of real-world data, we consider that our platform at least provides a viable starting point from which researchers can then decide for themselves, if the suggested workflows are valuable for their needs.

Coming back to the topic of imputation, we want to make it more clear therefore that our suggestions do come with caveats. We added the discussion of real-world data imputation in addition to our previous suggestion in the following:

In section **Fidelity and applicability to the real world** in **Discussion** at page 15-16 (rows 626-634) of main_R2_clean.docx:

Although we try to test as many datasets and methods as possible, it does not mean the predictions and recommendations built on these will work on any data. We cannot guarantee it. Real world data is complex (e.g., with more complex missing patterns, e.g., mixed by MNAR and MCAR, and bigger cross-run variances) and covers many situations (scenarios or sources or variation) that cannot exist in our relatively simpler simulations. **Though, the conclusions drawn here may not work universally, still, we think that OpDEA provides a useful starting point as a recommendation resource.** Researchers will no longer need to simply copy a workflow from some other publication, with no knowledge of whether the workflow really works well for their own data. **In this regard, OpDEA has the potential to** positively enhance proteomics research.

Reviewer-1-Comment-4: To thoroughly assess the effectiveness in integrating multiple workflows for DEP analysis is a highlight of this work and it is less investigated previously. The authors have shown the ens_multi-quant approach improves performance by increasing the mean pAUC(0.01) by < 5% for multiple datasets, which is in fact a minor gain. However, in reality, we often see an increased number of DEPs when searching the same data with different platforms (so many are available now), which is a common practice employed in quite a few studies. It would be interesting to test combining DEP lists given by two platforms using these well-designed benchmark data. For example, for datasets shown in Fig. 5D, would it be possible for the authors to try pooling DEPs from spt-DIA-tims and DIANN-DIA-tims, and pooling DEPs from MQ-DDA-tims and FG-DDA-tims to see how pAUC and the number of TP/FP vary with this type of ensemble method? This evaluation would offer critical information for users to decide whether it is worthwhile to run multiple platforms if they want to maximize the DEP coverage. Thanks for the additional effort.

Response to Reviewer-1-Comment-4: We thank the reviewer's good suggestion of evaluating whether combining DEA results based on different quantification platforms may further improve the performance. We used the HYEtimes735_LFQ and HYEtimes735_DIA datasets to test the pooling of DEPs detected by top 1st workflows from different quantification platforms, i.e., pooling DEPs from top 1st workflow of FG_DDA (FG_DDA in Fig. 5b) and top 1st workflow of MQ_DDA (MQ_DDA in Fig. 5b) (we call this method "pool_DDA"), pooling DEPs from top 1st workflow of DIANN_DIA (DIANN_DIA in Fig. 5b) and top 1st workflow of spt_DIA (spt_DIA in Fig. 5b) ("pool_DIA") and pooling all of the DEPs detected by all these four top 1st workflows (FG_DDA + MQ_DDA + DIANN_DIA + spt_DIA, named as "pool_DDA_DIA"). The pooling is implemented by a simple voting method, where the pooled q-value equals the minimum q-value of the candidates and the pooled logFC value equals the one has the biggest absolute value among the candidates. The comparison results are shown in below bar plots and table (also see our Supplementary

Note 1).

Method	pAUC(0.01)	pAUC(0.05)	pAUC(0.1)	TP	TN	FP	FN	Recall	Precision	Specificity	G-mean	nMCC
FG_DDA	0.69	0.75	0.76	2556	7474	137	2199	0.54	0.95	0.98	0.73	0.81
MQ_DDA	0.50	0.51	0.51	927	7477	134	3828	0.19	0.87	0.98	0.44	0.65
DIANN_DIA	0.82	0.87	0.88	3734	7557	54	1021	0.79	0.99	0.99	0.88	0.91
spt_DIA	0.77	0.85	0.86	3102	7564	47	1653	0.65	0.99	0.99	0.81	0.86
FG_DDA_ens	0.71	0.76	0.77	3096	6989	622	1659	0.65	0.83	0.92	0.77	0.80
MQ_DDA_ens	0.50	0.51	0.51	1607	7016	595	3148	0.34	0.73	0.92	0.56	0.67
DIANN_DIA_ens	0.60	0.79	0.84	3884	7328	283	871	0.82	0.93	0.96	0.89	0.90
spt_DIA_ens	0.78	0.85	0.86	3252	7459	152	1503	0.68	0.96	0.98	0.82	0.86
pool_DDA	0.66	0.73	0.75	2644	7326	285	2111	0.56	0.90	0.96	0.73	0.80
pool_DIA	0.78	0.88	0.90	3978	7513	98	777	0.84	0.98	0.99	0.91	0.93
pool_DDA_DIA	0.78	0.88	0.90	4178	7205	406	577	0.88	0.91	0.95	0.91	0.92

The results show that the pooling strategy really can improve the performance comparing to single workflows. The pooling methods can even work better than the ens_multi-quant (with postfix of ‘_ens’). In this test, we simply pooled DEPs from top 1st workflows under different settings. More TPs can be found with small increases in FP numbers. The results indicate that the integration of outputs from multi-quantification platforms has the potential to maximize the DEP coverage. However, for large proteomics projects, the running time will increase hugely when multiple quantifications are conducted.

Unfortunately, this is not the best way to systematically explore this topic. But we are now inspired. In our next paper, we will explore the design of efficient approaches for heterogeneous quantification data analysis to maximize the DEP coverages or other downstream applications. This can help users exploit OpDEA more effectively.

Comments from Reviewer 2

Reviewer #2 (Remarks to the Author):

Dear Editor and Authors,

The authors addressed many reviewer comments, which helped improve the manuscript's readability.

I suggested using ANOVA to study the responses obtained from this computer experiment instead of the CatBoost. However, I am sorry for making this suggestion. This analysis, now added to the CatBoost results, makes the manuscript more challenging to follow. When I first read the text, the article was hard to read and misinformed readers by overinterpreting the data.

The analysis should focus on describing the results rather than building prediction models or reporting p-values, because of the biased benchmark datasets, which exclusively capture technical variability.

I am pleased that now there are Figures that show:

- ranks of the various methods (Figure 4),
 - show boxplots of the pAUCs or the number of TP and FP (Figure 5),
 - or Figure 2E, where the pAUC and G-mean are correlated.
- , which is how the results should have been presented and discussed in the first place and exclusively.

However, many authors think they must report as many significant p-values as possible and use machine learning and AI whenever possible. And they are not wrong in thinking this.

It is what many journals, specifically Nature editors and reviewers, expect to see.

The editor must focus on ensuring FAIR principles:

https://en.wikipedia.org/wiki/FAIR_data

It happens that the websites maintained by authors go offline directly after publication. Therefore, I would ask the editor to ensure that all the data is deposited in a data repository maintained by a third party.

The data deposited must include the outputs of the DIA-NN, Spectronaut, MQ, etc, software used as input to workflows examined.

Furthermore, the authors should generate and publish a Docker image with all the scripts that automatically download the data from the third-party repository and replicate the analysis (7k + workflows).

Kind regards

Reviewer #2 (Remarks on code availability):

The editor must focus on ensuring FAIR principles:

https://en.wikipedia.org/wiki/FAIR_data

It happens that the websites maintained by authors go offline directly after publication. Therefore, I would ask the editor to ensure that all the data is deposited in a data repository maintained by a third party.

The data deposited must include the outputs of the DIA-NN, Spectronaut, MQ, etc, software used as input to workflows examined.

Furthermore, the authors should generate and publish a Docker image with all the scripts that automatically download the data from the third-party repository and replicate the analysis (7k + workflows).

Response to Reviewer-2: We thank the reviewer again for providing so many valuable suggestions which help improve this manuscript a lot.

To ensure the FAIR principles, the following efforts are made (see **Data Availability** and **Code Availability** at page 25-26):

1. We bought a long-term cloud service for supporting the running of our webserver (<http://www.ai4pro.tech:3838/>). We will maintain it well and extend it to implement more functions in our future work.
2. We stored our codes in the popular code database GitHub (<https://github.com/PennHui2016/OpDEA>) and we linked it with the Zenodo (<https://doi.org/10.5281/zenodo.10867031>). The offline tool of our OpDEA, which has the same function as our website but without uploading data has also been stored in the third-party repository Zenodo (<https://doi.org/10.5281/zenodo.10958381>).
3. The quantification outputs have been stored in Zenodo and can be obtained via <https://doi.org/10.5281/zenodo.10482353>. The expression matrices extracted from the quantification outputs can be obtained via <https://doi.org/10.5281/zenodo.10953347> and the benchmarking results can be downloaded through <https://doi.org/10.5281/zenodo.10953480>.
4. We submit supplementary data files (Supplementary Data 1-11 and Source Data) to the journal for supporting our findings reported in the manuscript and our figures.

To be honest, currently we are not familiar with the publishing of a Docker image. We may publish a Docker image after we have enough skills in preparing one or in our future version of OpDEA.

Comments from Reviewer 3

Reviewer #3 (Remarks to the Author):

The authors performed a substantial revision and addressed many of the questions raised by the reviewers. While I still think the paper is too technical, the analysis is solid and may be useful for users of proteomics pipelines to optimize the parameters. In light of the efforts that authors put into the revision I support the publication.

Response to Reviewer-3: We appreciate the reviewer's recognition of our efforts to improve our manuscript, and the proposal of the OpDEA to benefit the proteomics community. We have made substantial amendments to improve readability while also ensuring the manuscript is technically robust and correct.

To help readers of diverse backgrounds:

In the introduction, we listed down the key achievements and contributions of the work.

The results section is very technical but allows interested researchers to replicate our work and to understand how OpDEA was constructed and evaluated.

For lay readers or biologists who simply want to know what the key recommendations are, they can simply refer to the end of the discussions section for this. We have also included a resource on how to use the web and software resources.

Again, we thank the reviewer for acknowledging the extent of our efforts. But we also hope that our desire to communicate and engage with the scientific community shows as well.

Yours sincerely,

Hui Peng, Jinyan Li and Wilson Goh